



# Thermal regime, energy budget and lake evaporation at Paiku Co, a deep alpine lake in the central Himalayas

Yanbin Lei[1,2], Tandong Yao[1,2], Kun Yang[1,2,3], Lazhu[1], Yaoming Ma[1,2,4], Broxton W. Bird[5]

[1] Key Laboratory of Tibetan Environment Changes and Land Surface Processes, Institute of Tibetan Plateau Research, Chinese Academy of Sciences, Beijing 100101, China
[2] CAS Center for Excellence in Tibetan Plateau Earth System, Beijing, 100101, China
[3] Department of Earth System Science, Tsinghua University, Beijing 10084, China
[4] University of Chinese Academy of Sciences, Beijing, China
[5] Department of Earth Sciences, Indiana University-Purdue University Indianapolis (IUPUI), Indianapolis, IN 46202, USA.

*Correspondence to*: Yanbin Lei (leiyb@itpcas.ac.cn)

**Abstract.** Endorheic lakes on the Tibetan Plateau (TP) experienced dramatic changes in area and volume during the past decades. However, the hydrological processes associated with lake dynamics are still less understood. In this study, lake

evaporation and its impact on seasonal lake level changes at Paiku Co, central Himalayas, were investigated based on three years of in-situ observations of lake thermal structure and hydrometeorology (2015-2018). The results show that Paiku Co is a dimictic lake with thermal stratification at the water depth of 15-30 m between July and October. As a deep alpine lake, the large heat storage significantly influenced the seasonal pattern of heat flux over lake surface. Between April and July, when the lake gradually warmed, about 66.5% of the net radiation was consumed to heat lake water. Between October and January,

when the lake cooled, heat released from lake water was about 3 times larger than the net radiation. There was ~5 month lag between the maximum lake evaporation and maximum net radiation at Paiku Co. Lake evaporation was estimated to be 975±82 mm between May and December, with low values in spring and early summer, and high values in autumn and early winter. The seasonal pattern of lake evaporation at Paiku Co significantly affected lake level seasonality, that is, a significant lake level decrease of 3.8 mm/day during the post-monsoon season while a slight decrease of 1.3 mm/day during the pre-

monsoon season. This study may have implications for the different amplitudes of seasonal lake level variations between deep and shallow lakes.



## 1 Introduction

Lake not only plays an important role in global hydrological and biogeochemical cycle, but also serves as important water resources including drinking water supply, agricultural production, recreation and fisheries (e.g. Lehner and Döll, 2004). Lake is also of vital importance in sustaining regional ecological balance as being home for organism such as macrofauna, vegetation and microbe (e.g. Hoverman et al., 2012). Lakes additionally influences regional climate. For example, due to different heat properties between lake water and the land, lake water plays a cooling role in summer and warming role in winter (Scott and Huff, 1996). As such, investigations of lake water and energy budgets, which provide basic information regarding the loss and gain of water and energy, are important in ascertaining the role of lakes in local to regional scale communities (Sugita et al., 2019).

The Tibetan Plateau (TP) hosts the greatest concentration of high-altitude inland lakes in the world. More than 1200 lakes (>1 km$^2$) are distributed on the TP, with a total lake area of ~45000 km$^2$ (Ma et al., 2011; G. Zhang et al., 2014). During the past decades, lakes on the TP experienced significant changes. Most lakes on the interior TP expanded dramatically since the late 1990s, in contrast with lake shrinkage on the southern TP (e.g. Lei et al., 2014). Lake water temperature increased dramatically in response to climate warming (e.g. Su et al., 2019). Lake ice duration shortened across the TP during the past decades (Ke et al., 2013; Cai et al., 2017).

Evaporation is one of the largest components of lake water budget (Li et al., 2001; Morrill, 2004; Xu et al., 2009; Yu et al., 2011). Direct measurements of lake evaporation were usually conducted by using the eddy covariance system or energy budget method (e.g. Blanken et al., 2000; Winter et al., 2003; Rouse et al., 2003, 2008; Rosenberry et al., 2007; Giannoiu and Antonopoulos, 2007; Zhang et al., 2014). Eddy covariance is a more direct means of measuring lake evaporation, but it is difficult to perform and requires specialized and expensive instrument, which makes it more suitable for short-term observations (Giannoiu and Antonopoulos, 2007; Rouse et al., 2003, 2008). Although the energy budget method also needs costly instrumentation and significant personnel commitment for fieldwork, it is more suitable for accurate, long-term monitoring (Winter et al., 2003).

On the TP, there are several studies regarding lake evaporation using the eddy covariance system, e.g. Nogring Lake (Li et al., 2015), Qinghai Lake (Li et al., 2016), Nam Co (Wang et al., 2017), Siling Co (Guo et al., 2016). Results show that the seasonal pattern of lake evaporation is significantly affected by the lake heat storage, especially for deep lakes. At Nam Co, for example, Haginoya et al. (2009) found that the sensible and latent heat fluxes were small during the spring and early summer, and increased considerably during the autumn and early winter due to the large heat storage. However, lake evaporation throughout the year is not typically investigated because it is difficult to install and maintain measurement platform due to the harsh natural conditions on the TP and the influence of lake ice during the late autumn and early winter. As a result, how lake evaporation affects seasonal lake level changes remains unclear due to lack of comprehensive observation of lake water budget.



To understand lake water budget and the associated lake level changes, we conducted comprehensive in situ observations at Paiku Co in the central Himalayas since 2013. In this study, we first address the thermal regime and changes in lake heat storage at Paiku Co based on three years' water temperature profile data, then investigate hydro-meteorology and energy

budget over the lake, and finally analyse the seasonal pattern of lake evaporation and its effect on lake level changes.

## 2 Methodology

### 2.1 Site description

Paiku Co (85°35.12′ E, 28°53.52′ N, 4590m a.s.l) is located in the north slope of the central Himalayas. The lake has a surface area of 280 km$^2$ and watershed area of 2376 km$^2$. Bathymetry survey showed that Paiku Co has mean water depth of

41.1 m with the maximum water depth of 72.8 m (Lei et al., 2018). The lake is hydrologically closed and lake salinity is about 1.7 g/L. Glaciers are well developed to the south of the Paiku Co, with a total area of ~123 km$^2$. Dozens of paleo-shorelines are visible around Paiku Co. The highest shoreline is ~80 m above the modern lake level. Wünnemann et al. (2015) found that there was a close relationship between glacier dynamics and lake level changes since the Last Glacial Maximum (LGM). The lake has been shrinking since the 1970s (Nie et al., 2013). Between 1972 and 2015, lake levels at Paiku Co

decreased by 3.7 ±0.3 m and water storage reduced by 8.5 % (Lei et al., 2018).

### 2.2 Data acquisition

In situ observations, including lake water temperature profile and hydro-meteorology over lake surface, were carried out in Paiku Co basin. HOBO water temperature loggers (U22-001, Onset Corp., USA) were used to monitor water temperature profile with an accuracy of ±0.2 $^{o}$C. Two water temperature profiles were installed in Paiku Co's southern (0-42 m in depth)

and northern (0-72 m in depth) basins (Fig. 1). In the southern basin, water temperature was monitored at the depths of 0.4 m, 5m, 10 m, 15 m, 20 m, 30 m and 40 m. In the northern basin, water temperature was monitored at the depths of 0.4 m, 10 m, 20 m, 40 m, 50 m, 60 m and 70 m. Since lake level fluctuates seasonally, the depth of water temperature loggers may also have changed in a range of 0.4-0.8 m. Water temperatures were recorded at an interval of 1 hour and daily-averaged values were used in this study. Three years' observational data from June 2015 to May 2018 from the southern basin was acquired,

while only one year's data (June 2016 and May 2017) from the northern basin was acquired.

>>Fig. 1<<

To investigate the local hydro-meteorology at Paiku Co, air temperature and specific humidity over the lake were monitored since June 2015 using HOBO air temperature and humidity loggers (U12-012, Onset Corp., USA). The instrument has an accuracy of 0.35 $^{o}$C for air temperature and 2.5% of relative humidity. Two loggers were installed in an outcrop ~2 m above

the lake surface (Fig. 2). One is located in the north shoreline of Paiku Co, the other is located in the central shoreline of Paiku Co (Fig. 1). The instruments were under large rock where there was a hole facing the lake. The site was ventilated and





therefore the meteorological condition over the lake surface can be recorded. There was no data available between February and May 2017 because the instrument battery was too low.

>>Fig. 2<<

Radiation, including downward shortwave radiation and longwave radiation to lake, was measured by Automatic Weather Station (AWS) at Qomolangma station for Atmospheric Environmental Observation and Research, Chinese Academy of Sciences (CAS). This station is located at the northern slope of Mount Everest, about 150 km east of Paiku Co (87°1.22′E, 28°25.23′N, 4276 m a.s.l). The 2 m air temperature, relative humidity, wind speed, radiation were recorded at an interval of 10 min. In this study, downward shortwave radiation and longwave radiation at this station were used because the climate

conditions between Paiku Co and Qomolangma station were very similar, including topography, altitude, cloud cover etc. Nonetheless, weekly averaged radiation was used to calculate lake evaporation in order to reduce the error caused by regional difference. The related information about hydro-meteorology observations at Paiku Co basin are listed in Table 1.

**2.3 Energy budget derived lake evaporation**

Lake evaporation was calculated using the energy budget (Bowen-ratio) method as described by Winter et al. (2003) and
Rosenberry et al. (2007). The energy budget of a lake can be mathematically expressed as:

$$R = H + lE + S + G + A_v \qquad (1)$$

where R is the net radiation on the lake, H is the sensible heat flux from lake surface, lE is the latent heat utilized for evaporation, S is the change in lake water energy, G is the heat transfer between lake water and bottom sediment, and $A_V$ is the energy advected into lake water. The units used for the terms of Eq (1) are $W \cdot m^{-2}$. As a deep lake, the influence of river
discharge on the total lake heat storage at Paiku Co is very small and can be neglected. Therefore, we do not consider the influence of G and $A_V$ on the lake energy budget in this study.

The net radiation on the lake can be expressed as the following:

$$R = R_s - R_{sr} + R_a - R_{ar} - R_w \quad (2)$$

where $R_s$ is downward shortwave radiation, $R_{sr}$ is the reflection of solar radiation from lake surface, which is taken as 0.07 $R_s$
in this study (Gianniou and Antonopouls, 2007), $R_a$ is downward longwave radiation to lake, $R_{ar}$ is the reflected longwave radiation from the lake surface, which is taken as 0.03 $R_a$, and $R_w$ is the upward longwave radiation from the lake surface. The units of the items in Eq (2) are $W \cdot m^{-2}$.

The longwave radiation from lake is approached by the equation:

$$R_a = \varepsilon_a \times \sigma \times (T_w + 273.15)^4 \quad (3)$$

where $R_a$ is the longwave radiation from lake, σ is the Stefan-Boltzmann constant (=5.67×10$^{-8}$ $W \cdot m^{-2} \cdot K^{-4}$), $\varepsilon_a$ is the water emissivity (0.97 for water surface) and $T_w$ is surface water temperature of the lake (°C). In this study, the water temperature at the depth of 0.4-0.8 m was use to represent the surface water temperature because the surface water temperature was not





monitored. Although the water temperature at the depth of 0.4-0.8 m does not represent 'skin' temperature (Prats et al., 2018), the daily average between them is very similar during most time of a year because surface water can be mixed quickly

by water convection or strong wind in the afternoon.

The sensible heat flux is related to the evaporative heat flux through the Bowen ratio (Henderson-Sellers, 1984):

$$\beta = \frac{H}{lE} = \gamma \times P \times \frac{T_s - T_a}{e_{sw} - e_d} \qquad (4)$$

where $\beta$ is Bowen ratio, $T_s$ is the surface water temperature ($^{o}$C), $T_a$ is air temperature at 2m high above the water surface ($^{o}$C), $e_{sw}$ and $e_d$ are the saturated vapor pressure at the temperature of the water surface and the air vapor pressure above the

water surface (kPa), respectively, P is air pressure (kPa), and $\gamma$ is the psychrometric constant, $6.5 \times 10^{-4}$ $^{o}$C$^{-1}$. In this study, air temperature, air pressure and specific humidity were monitored at the lake's shore. Saturated vapor pressure at the lake surface was calculated according to surface water temperature in the southern center of the lake. To match the radiation, all the input data were averaged at weekly interval before lake evaporation was calculated.

Changes in lake heat storage (S) were calculated according to the detailed lake bathymetry and water temperature profile:

$$S = \frac{\sum_{i=0}^{72.8} c_w \times \rho_w \times \Delta V_i \times \Delta T_i}{A_l} \quad (5)$$

where $c_w$ is the specific heat of water (J·kg$^{-1}$·K$^{-1}$), $\rho_w$ is water density (=1000 kg·m$^{-3}$), $\Delta V_i$ is the lake volume at certain depth (m$^3$), and $\Delta T_i$ is water temperature change at the same depth, $A_l$ is lake area (m$^2$). S was calculated at an interval of 5 m and therefore there are 13 layers in vertical direction. $\Delta V_i$ was acquired according to the 5m isobath of Paiku Co (Lei et al., 2018). $\Delta T_i$ was calculated at 5 m interval as the average temperature of the top and bottom layer. Changes in lake heat storage for

the bottom water (>40 m) in 2015/2016 and 2016/2017 were calculated according to the data in 2016/2017 since there is no data in the other two years.

## 3 Results

### 3.1 Thermal structure of lake water

Water temperature profiles between 2015 and 2018 show that Paiku Co was thermally stratified between July and October,

and fully mixed between November and June in each year of the study period (Fig. 3). Lake water temperature increased rapidly from 2 to 7 $^{o}$C between April and June due to the strong solar radiation. During this period, temperature differences between the surface and bottom water was less than 1 $^{o}$C. The temperature gradient on vertical profile increased dramatically in late June with clear stratification occurring in July, which corresponded to a significant reduction in wind speed (data not shown). Strong lake surface heating and the reduction in wind speed together contributed to the development of thermal

stratification (Wetzel, 2001). During the summer stratification period, the surface water warmed rapidly from 7 to ~13 $^{o}$C between July and August, while the bottom water warmed slowly. As a result, the thermocline formed between 15 m and 30 m water depth, with the largest temperature difference of 5~6 $^{o}$C occurring in late August.



Lake surface temperature started to decrease gradually since September due to the decrease in solar radiation, however, the bottom water continued to warm slowly (Fig. 3). As a result, the water temperature gradient on vertical profile decreased,

which caused the lake stratification to break down in late October of each year. Notably, the timing of the stratification breaking down corresponded well to significantly increased wind speed during this time (data not shown). Unlike the rapid appearance of lake stratification in late June, the breakdown of stratification occurred more gradually, with the mixed layer deepening gradually throughout October (Fig. 4). The mixed layer reached to 40 m water depth on October 13th, 2016, and to 70 m water depth about two weeks later (October 30th). Following the complete breakdown of the water column's

stratification, the bottom water experienced rapid warming in several days due to its mixture with the warmer water from the upper layer. For example, the water temperature at 70 m water depth remained stable at ~6.9 $^o$C from July to October, but increased abruptly from 6.9 to 8.6 $^o$C in less than one weeks (October 25$^{th}$ to October 30$^{th}$).

Paiku Co's water column was fully mixed between November and May as indicated by the identical lake water temperature profiles at the two monitoring sites (Fig. 3, Fig. 4). Water temperature of the whole lake decreased gradually from 8.6 to 1 $^o$C

from November to January and remained stable at 1-2 $^o$C until March. Landsat satellite images show that Paiku Co did not completely freeze up in winter during the study period, therefore lake water stratification in winter did not appear as reported in other studies on the TP (Wang et al., 2019).

The thermal structure of Paiku Co indicates that it is a dimictic lake, which is similar to Bangong Co (Wang et al., 2014) and Nam Co (Wang et al., 2019), but different from Dagze Co (Wang et al., 2014). The water temperature gradients at Paiku Co

and other lakes on the TP are considerably lower than those in other parts of the world (Livingstone, 2003; Stainsby et al., 2011, Zhang et al., 2015), which is probably due to the lower air temperature in summer in this high elevation area. As a deep lake, Paiku Co stored a large amount of energy in spring and summer and released it in autumn and early winter. The identical lake water temperature profile between November and June at Paiku Co indicates that changes in lake heat storage are not only affected by surface water, but also the bottom water. For deep lakes like Paiku Co, changes in lake heat storage

can be significantly underestimated if only the surface water is considered.

>>Fig. 3<<

>>Fig. 4<<

## 3.2 Spatial difference of lake water temperature

Spatial difference of lake water temperature was investigated using in-situ observations at different sites in 2016/2017. First,

we compared water temperature difference between Paiku Co's southern and northern basins (Fig.5). Since the northern basin is much deeper than the southern basin, lake water in the northern basin warmed more slowly than that in the southern basin during the spring and early summer, and cooled more slowly during the autumn and early winter. The surface water temperature in the southern basin was about 0.85 $^o$C higher on average than that in the northern basin between April and September. The water temperature became spatially uniform in late October when the water column was fully mixed. In

November and December, when the water temperature decreased, the surface water temperature in the southern basin was





about 0.45 °C lower on average than that in the northern basin. Water temperature became spatially uniform at both basins again between January and March. Similar spatial difference can also be found at 10 m depth (Fig.5), indicating that this characteristics exists in the surface layer (the epilimnion).

Contrasting changes in water temperature occurred in the bottom water (the hyplimnion). Between the mid-August and mid-
September, water temperature at 20 m depth was about 0.81 °C lower in the southern basin than in the northern basin, which contrasts with that of the surface layer (Fig. 5c). Similar conditions occurred at 40 m depth, where water temperature was 0.75 °C lower in the southern basin relative to the northern basin between mid-September and mid-October (Fig. 5d). This contrasting pattern occurred in the late summer or early autumn when the vertical temperature gradient started to decrease. As shown in Fig.3, both the start and end of lake stratification were about half a month earlier in the southern basin relative
to the northern basin. However, water convection occurred earlier in the northern basin relative to the southern basin during this period due to the relatively lower vertical temperature gradient (Fig. 3). Lower temperature gradient caused stronger water convection in the northern basin compared with the southern basin during the late summer and early autumn.

We further compared the water temperature between the lake centre and shoreline. Water temperature along the northern and eastern shorelines of Paiku Co was recorded by HOBO water level loggers (Fig. 1). The results show that the water
temperature along the shoreline was very sensitive to air temperature and fluctuated with much larger amplitude than that in the lake centre, although both exhibited similar seasonal fluctuations (Fig. 5E, 5F). For example, shoreline water warmed more quickly to higher temperature during the spring and summer as compared with that in the lake centre, but conversely the shoreline water cooled more quickly in the autumn. The spatial difference of water temperature indicates that large errors can result if only water temperature data collected at the shoreline are used to calculate lake heat storage and energy budget.
>>Fig. 5<<

## 3.3 Lake hydrometeorology

Lake hydrometeorology was measured at the north and central shoreline of Paiku Co, respectively. Energy budget and lake evaporation at Paiku Co in this part are addressed according to the data at the north shoreline. The spatial difference between the north and central shoreline will be discussed in part 4.1.
Annual mean air temperature over Paiku Co was 4.7 °C in 2016 with the highest air temperature in July (11.2 °C) and the lowest in January (-2.4 °C). There was a ~1.5 month lag between lake surface temperature and air temperature. The highest lake surface temperature occurred in late August and the lowest in February (Fig. 6). The temperature difference between the lake surface and the overlying atmosphere exhibited a linear increasing trend from June to November, and a linear decreasing trend from January to June. Positive temperature difference mainly occurred during the autumn and winter with
the highest value of ~7 °C in late October and early November. Negative temperature difference occurred during the spring and early summer with the lowest value of -3 °C in June.

Atmospheric water vapor content at Paiku Co was elevated from June to September (Fig. 6), which is consistent with the occurrence of Indian summer monsoon precipitation. During the non-monsoon season (October to May), the atmospheric





water vapor content was generally low. The water vapor pressure difference between the lake surface and the overlying

atmosphere exhibited a linear increasing trend from June to September and then a linear decreasing trend from October to February. High water vapor difference occurred between September and December (0.76 kPa), while low difference was observed between March and June (0.39 kPa).

>>Fig. 6<<

Radiation, including downward shortwave radiation, downward longwave radiation to lake and upward longwave radiation

from the lake body, are the main drivers of lake's energy balance. Downward shortwave radiation at Paiku Co had an annual average of 251.8 W·m$^{-2}$ (Fig. 7), which is slightly higher than the TP average due to its lower latitude (Yang et al., 2009). Downward longwave radiation to the lake had an average of 235.8 W·m$^{-2}$. Upward longwave radiation from the lake body had an annual average of 336.8 W·m$^{-2}$. The total incoming radiation was always higher than the outgoing radiation. The net radiation over Paiku Co varied seasonally between 19.0 and 212.1 W·m$^{-2}$, with an average value of 125.8 W·m$^{-2}$. Relatively

high net radiation occurred from April to August (200.4 W·m$^{-2}$), with the highest value in June (212.1 W·m$^{-2}$). Relatively low net radiation occurred from October to February (52.2 W·m$^{-2}$), with the lowest value in December (19.7 W·m$^{-2}$).

>>Fig. 7<<

### 3.4 Impact of lake heat storage on the heat fluxes

Changes in lake heat storage at Paiku Co were quantified using in-situ observations of water temperature profile and detailed

lake bathymetry. This also makes it possible to evaluate the impact of lake heat storage on the heat flux at lake surface (Fig. 8). Between April and July when Paiku Co warmed gradually, the lake water absorbed energy at an average rate of 128.6 W·m$^{-2}$, accounting for 66.5% of the net radiation during the same period. The lake heat storage increased most rapidly in June, with an average rate of 191.6 W·m$^{-2}$, accounting for 91.6% of the net radiation during the same period. The lake heat storage reached its peak in late August, when the surface water temperature was in the highest. Between October and

January, when Paiku Co cooled, the lake heat storage decreased at an average rate of 137.5 W·m$^{-2}$, which was more than 3 times larger than the net radiation during the same period. The lake heat storage decreased most rapidly in November at an average rate of 193.6 W·m$^{-2}$, which was about 5 times larger than the net radiation during the same period.

The heat flux at lake surface was determined as the difference between the net radiation and changes in lake heat storage. The lowest heat flux occurred in June and the highest in November, which is consistent with the seasonal pattern of changes

in lake heat storage. The seasonal pattern of heat flux at Paiku Co is almost anti-phase with the net radiation (Fig 8B). There was a ~5 month lag between the maximum heat flux and maximum net radiation due to the large heat storage of lake water. Although net radiation was high in spring and summer, a large portion of energy was consumed to heat lake water, which resulted in low heat flux. In the autumn and early winter, although net radiation was relatively low, a large amount of heat stored in the lake was released into the overlying atmosphere, which resulted in high heat flux.

>>Fig. 8<<





The Bowen ratio determines the distribution of sensible and latent heat flux. At Paiku Co, the Bowen ratio varied in a range of -0.26~+0.37, with an annual average value of +0.08 (Fig. 9, Tab. 2). Negative value occurred between April and July, with an average value of -0.12, indicating the lake water absorbed energy from the overlying atmosphere. Positive value occurred between August and January, with an average value of 0.20, indicating the lake water released energy to the overlying atmosphere.

Latent heat flux is the main component of heat flux, with an average value of 112.3 W·m$^{-2}$ between May and December. The latent heat was in low value between May and June, with an average of 38.7 W·m$^{-2}$, and high value between October and December, with an average of 153.3 W·m$^{-2}$ (Tab. 2). Latent heat flux at Paiku Co is positively correlated with the water vapor pressure difference between the lake surface and the overlying atmosphere ($r^2$=0.41). Sensible heat flux has an annual average value of 13.3 W·m$^{-2}$, accounting for ~11% of latent heat flux. Sensible heat flux was negative between April and July with an average value of -5.6 W·m$^{-2}$ (Fig. 9b), and was in positive value between August and December with an average of 23.0 W·m$^{-2}$. There was a high correlation between sensible heat and the water temperature difference between surface water and the overlying atmosphere ($r^2$=0.86).

>>Fig. 9<<

## 3.5 Lake evaporation at Paiku Co

Lake evaporation at Paiku Co between May and December is shown in Fig. 10. Lake evaporation was generally low between May and June with an average value of 1.7 mm/day. In July and August, lake evaporation increased rapidly from 2.9 to 4.1 mm/day. High lake evaporation occurred between September and December, with an average value of 5.4 mm/day. The total lake evaporation was estimated to be 975 mm between May and December during the study period. Lake evaporation between middle January and April is not determined because the energy budget during this period is also affected by intermittent lake ice.

>>Fig. 10<<

Lake evaporation at Paiku Co lagged net radiation by ~5 months and exhibited a similar seasonal pattern with changes in lake heat storage. Regression analysis shows that lake evaporation at Paiku Co positively correlated with changes in lake heat storage ($r^2$=0.63, P<0.001), but negatively correlated with net radiation ($r^2$=0.22, P<0.001), which indicating that the seasonal pattern of lake evaporation is significantly altered by lake heat storage. When the net radiation was high between May and July, most of the energy is used to heat the lake water and only a small part of it is consumed as to the latent heat flux. When the net radiation was low between November and December, a large amount of heat was released from the lake water as latent heat to the overlying atmosphere. Lake evaporation exhibited similar patterns with the water vapor pressure difference between surface water and the overlying atmosphere ($r^2$=0.33).

Significant changes in lake ice phenology occurred at Paiku Co during the study period. Generally, Paiku Co was covered by lake ice between the mid-January and mid-April (e.g. the winter of 2013/2014). During ice covered period, lake level was





very stable because lake ice can effectively prohibit evaporation. However, lake surface of Paiku Co did not completely frozen up between 2015/2016 and 2017/2018 with only intermittent lake ice in the shoreline region. Contrasting with the ice

covered period, Paiku Co's water level decreased considerably by 199 mm on average between January and April. Assuming lake evaporation between January and April is equal to lake level decrease because there was almost no surface runoff during this period, annual lake evaporation at Paiku Co is estimated to be 1174 mm during the study period. This also indicates that that annual lake evaporation increased by ~20.4% in recent years due to the disappearance of lake ice.

## 4 Discussion

**4.1 The representativeness of lake hydrometeorology**

The components of energy budget are not uniform at large lake and the meteorological station near the lake centre is usually expected to produce more accurate estimation of heat flux (Sugita, 2019). To check the representativeness of hydrometeorology at the shoreline of Paiku Co, we first compare air temperature and relative humidity between shoreline and lake centre. We set up a platform in the southern centre of Paiku Co in September 2019 (water depth: 19 m; least

distance from shoreline: 2 km) and a simple AWS station (GMX600) was installed on the platform. Meteorological data between September 22$^{nd}$ and October 26$^{th}$ were acquired and compared with that from shoreline (Fig. 11). Result shows that both air temperature and relative humidity fluctuated very similarly between the shoreline and lake centre, indicating the meteorological data from the shoreline of Paiku Co can be used to represent the general condition of the whole lake at least during the observed period. Unfortunately, the platform was damaged by lake ice in winter 2019/ 2020, so there is no more

data available.

Then we compare air temperature and relative humidity at the north and central shoreline of Paiku Co (Fig. 6). Results show that air temperature from the north and central shoreline of Paiku Co varied similarly throughout a year, except the considerable difference in early summer and early winter (Fig. 6a). Air temperature is about 2.7 $^{o}$C lower in May and June in the north shoreline than that in the central shoreline, but about 4 $^{o}$C higher in November and December. Different from air

temperature, water vapour content at both sites varied very similarly throughout the year (Fig. 6b). Similar with air temperature, Bowen ratio at both sites also varied similarly throughout a year, except the considerable difference in early summer and early winter. Bowen ratio is 0.22 higher in May and June in the north shoreline than that in the central shoreline, but 0.19 lower in November and December.

Lake evaporation derived from the north and central shoreline of Paiku Co also exhibits very similar seasonal fluctuations

throughout a year (Fig. 10), with slight difference in early summer and early winter. Lake evaporation derived from the north shoreline is 0.47 mm/day lower in May and June than that from the central shoreline, but 0.66 mm/day higher in November and December. Total lake evaporation derived from the north shoreline is 22 mm lower between May and December than that from the central shoreline. Although there is some spatial difference, the similar seasonal patterns of energy budget and lake evaporation at different sites indicate that our results are reliable.



### 4.2 Uncertainty of lake evaporation estimation

There are several factors that can cause uncertainty of lake evaporation. The first one is the determination of solar radiation and atmospheric long wave radiation at Paiku Co. In this study, solar radiation and atmospheric long wave radiation at Qomolangma station, which is about 150 km away from Paiku Co, were used to represent values at Paiku Co. To evaluate the spatial difference, we made a comparison of solar radiation at Paiku Co and Qomolangma Station by using Hamawari-8 satellite data (Tang et al., 2019; Letu et al., 2020). The results show that daily solar radiation at the two sites exhibited very similar seasonal fluctuations ($R^2=0.55$, P<0.001), with standard deviation of 23.9 $W·m^{-2}$. Assuming approximately 70% of the net radiation was consumed by lake evaporation (Lazhu et al., 2016), the uncertainty of lake evaporation due to error in solar radiation was ~74.5 mm per year ($\Delta E_1$).

The second factor affecting the estimation of lake evaporation is lake water temperature. As we have shown in section 3.2, there is considerable spatial difference of lake surface temperature between the southern and northern basin of Paiku Co. In this study, lake water temperature profile in the southern basin was used to determine the lake heat storage and energy budget. Although there were similar seasonal fluctuations (Fig. 5), water temperature exhibited considerably spatial differences between the lake's southern and northern basins. To estimate the uncertainty of lake evaporation, we further calculated lake evaporation at Paiku Co by using the same lake hydro-meteorology data, but the water temperature profile in the northern basin in 2016. The difference of lake evaporation between the two sites can be roughly taken as one of the uncertainties of lake evaporation at Paiku Co.

Lake evaporation using water temperature from the northern basin was estimated to be 911 mm from June to December 2016, which was 20 mm larger compared with that estimated from the southern basin (891 mm). The largest difference in lake evaporation between these two sites was in June and November. The accumulated lake evaporation from the northern basin was 51 mm higher than that from the southern basin in June, but 41 mm lower in November. Different lake heat storage in the southern and northern basins determined the energy distribution that can be used to evaporate lake water. Assuming similar error of lake evaporation between May and June, the uncertainty of lake evaporation caused by water temperature difference was estimated to be ~34.6 mm ($\Delta E_2$). Thus, the total uncertainty of lake evaporation was estimated to be 82.1 mm ($=\sqrt{{\Delta E_1}^2 + {\Delta E_2}^2}$), accounting for 8.4% of total evaporation between May and December.

Uncertainty of lake evaporation in this study was also validated by comparing lake level changes during the post-monsoon season and pre-monsoon seasons when the runoff was still very low. Runoff measurements at the three large rivers feeding Paiku Co (Fig. 1) makes it possible to compare the lake evaporation with lake level decrease. During the pre-monsoon season (mid-April to mid-May), lake evaporation (1.7 mm/day) was quite similar with the decreasing rate of lake level (1.8 mm/day). The high consistency between lake evaporation and lake level decrease confirms the reliability of lake evaporation estimation. During the post-monsoon season (October to January), lake evaporation (5.4 mm/day) is considerably higher than the rate of lake level decrease (3.8 mm/day). This discrepancy may be due to the contribution of precipitation and surface runoff (Tab. 3). As shown in Table 3, runoff at the three large rivers can contribute to lake level increase by 0.7~1.6



mm/day in October, thereby partially offsetting lake level changes from lake evaporation. According to this difference of 0.9 mm/day during the post-monsoon season, the error of lake evaporation is estimated to be 82.8 mm/year.

### 4.3 Comparison of lake evaporation with other lakes on the TP

To further explore the impact of lake heat storage on the seasonal pattern of lake evaporation, we compared lake evaporation at Paiku Co with other lakes on the TP. We only selected lakes with direct measurements of lake evaporation, including the eddy covariance system or energy budget method. At Ngoring Lake (area, 610 km$^2$; mean depth, 17 m) on the eastern TP, Li Z. et al. (2015) investigated the lake's energy budget and evaporation in 2011-2012 using the eddy covariance system, and found that the latent heat at Nogring Lake was lowest in June, peaked in August and then decreased gradually from September to November. At Qinghai Lake (area, 4430 km$^2$; mean depth, 19 m) on the northeast TP, Li X. et al. (2016) conducted studies concerning the lake's energy budget and evaporation in 2013-2015 using the eddy covariance system, and found that there was a 2–3 month delay between the maximum net radiation and maximum heat flux. Compared with the two larger but shallower lakes, there was longer time lag between the heat flux and net radiation at Paiku Co. As we have shown, Paiku Co has the mean water depth of ~41 m and the water column is fully mixed between November and June. This means that the lake can store more energy in spring and early summer than shallow lakes, and can release more energy to the overlying atmosphere in the autumn and early winter.

At Nam Co, a large and deep lake on the central TP, there have been several studies regarding lake evaporation (Haginoya et al., 2009; Ma et al., 2016; Wang et al., 2016, 2019). Haginoya et al. (2009) found that lake evaporation at Nam Co was lowest in May and highest in October. The Bowen ratio-derived lake evaporation was estimated to be 916 mm in 2013 (Lazhu et al. 2016). Comparison with Paiku Co shows that both lakes exhibited similar seasonal pattern of lake evaporation, although lake evaporation at Paiku Co was slightly larger than that at Nam Co due to its higher solar radiation. In fact, although the maximum depth at Nam Co is greater than that at Paiku Co, the average water depth of the two lakes is similar(Wang et al., 2009; Lei et al., 2018), which resulted in similar seasonal pattern of lake evaporation. At Siling Co, another large and deep lake on the central TP, monthly lake evaporation was found to vary within a range of 2.4-3.3 mm/day between May and September, 2014, with a total amount of 417.0 mm during the study period (Guo et al., 2016). Although the accumulative evaporation between Paiku Co and Siling Co was similar between May and September, lake evaporation at both lakes between October and December can not be further compared because the energy flux at the lake was not measured at Siling Co.

### 4.4 Implications for the seasonal lake level variations on the TP

The quantification of lake evaporation is important for understanding lake water budget and associated lake level changes. Compared with the eddy covariance system that can only work until October/November when the lake surface begins to freeze (Li et al., 2015; Wang et al., 2017; Guo et al., 2016), our results give a full description of lake evaporation during the entire ice-free period. More importantly, our results indicate that for deep lakes on the TP, evaporation during the post-





monsoon season can be much higher than that during the pre-monsoon seasons due to the release of large amount of stored heat (Haginoya et al., 2009), despite both air temperature and net radiation are already much lower. In this sense, lake evaporation during the cold season (October to December) is of great importance to lake water budget and can significantly affect the amplitude of lake level changes, especially for deep lakes.

As shown in Fig. 11b, lake level at Paiku Co decreased considerably at a rate of 3.8 mm/day on average between October
and December, which is in contrast to the slight decreasing rate of 1.3 mm/day in mid-April and May. So, what is the main cause for the large difference of lake level decrease during the two dry seasons? Runoff measurements at the three main rivers feeding Paiku Co indicate that the surface runoff had a weak impact on lake level changes during the pre-monsoon and post-monsoon seasons (Tab. 3). The seasonal pattern of lake evaporation can explain this well. High lake evaporation rates during the post monsoon season led to the rapid lake level decrease, while low lake evaporation in pre-monsoon season led
to much lower lake level decrease. This suggests that lake evaporation can largely determine the amplitude of lake level changes in dry seasons.

In a larger sense, our result may have implication for the different patterns of lake level seasonality that have been observed on the TP. Phan et al. (2012) showed that seasonal lake level variations in the southern TP are much larger than those in the northern and western TP. Lei et al (2017) investigated the lake level seasonality across the TP and found that there were
different amplitudes of lake level fluctuations even in similar climate regimes. For example, lake level at Nam Co and Zhari Namco, two large and deep lakes on the central TP (Wang et al., 2009, 2010), decreased considerably by 0.3-0.5 m in cold season (October to December), while lake level at two nearby small lakes, Bam Co and Dawa Co, decreased slightly by 0.1-0.2 m during the same period. Different lake heat storage can play an important role in the amplitude of lake level seasonality. For deep lakes (e.g. Paiku Co, Nam Co and Zhari Namco), the latent heat flux (lake evaporation) over lake
surface may lag the solar radiation by several months due to the large heat storage of lake water. For this kind of lake, the lake level drop is most dramatic in the autumn and early winter when lake evaporation is high. For shallow lakes, the latent heat flux closely follows solar radiation, with high lake evaporation during the pre-monsoon and monsoon seasons, and low lake evaporation during the post monsoon season (Morrill et al., 2004). Meanwhile, shallow lakes freeze up 1-2 months earlier than deep lakes. When the lake surface is covered by ice, lake evaporation is effectively prohibited. Consequently,
lake level decreased more slowly in post monsoon season in shallow lakes than that in deep lakes. This phenomenon can also be seen in some thermokarst lakes on the northern TP (Luo et al., 2015; Pan et al., 2017).

## 5 Conclusion

Lake evaporation and its impact on seasonal lake level changes were investigated based on three years' in-situ observations of lake water temperature profile and hydrometeorology at Paiku Co, a deep alpine lake in the central Himalayas. The results
show that Paiku Co is a dimictic lake with clear lake stratification between July and October. The thermocline formed between 15 m and 30 m water depth, with the largest temperature difference (5~6 $^{\circ}$C) occurring in August. The lake is

completely mixed between November and June. Considerable spatial difference of lake water temperature was also investigated between the southern and northern basins of Paiku Co.

As a deep alpine lake, lake heat storage significantly affected the seasonal pattern of energy budget and lake evaporation.
The lake absorbed most of net radiation to heat the lake water in the spring and early summer and released it to the overlying atmosphere in autumn and early winter. Between April and July, about 66.5% of the net radiation was consumed to heat the lake water. Between October and January, heat released from lake water was about 3 times larger than the net radiation. As a result, there was about a 5 month lag between the maximum heat fluxes and the maximum net radiation due to the large heat storage of lake water. Lake evaporation was estimated to be 975±82 mm between May and December during the study
period, with low values between May and June (1.7 mm/day), and high values between October and December (5.4 mm/day). Our result also indicates that that annual lake evaporation increased by ~20.4% during the study period due to the disappearance of lake ice.

This study may have implications for explaining the different seasonal lake level changes between shallow and deep lakes. For deep lakes like Paiku Co, high lake evaporation during the post monsoon season may leads to the rapid decrease in lake
level. In contrast, low lake evaporation during the pre-monsoon season may lead to slight lake level decrease. For shallow lakes, the seasonal pattern of lake evaporation varies similarly with the net solar radiation, which results in slight lake decrease in post-monsoon season and less amplitude of lake level seasonality.

## Data availability

All original data presented in this paper are publicly available via National Tibetan Plateau Data Center
(http://data.tpdc.ac.cn/en/).

## Author contribution

LeiY.B. and Yao T.D. conceived and designed the experiments; Lei.Y., YaoT.D., Yang K., Lazhu, and Ma Y.M. analyzed the data; LeiY.B. performed the fieldwork and wrote the paper; Bird B.W. helped write the paper.

## Competing interests

The authors declare that they have no conflict of interest.

## Acknowledgement

This research has been supported by the Strategic Priority Research Program of Chinese Academy of Sciences (XDA2006020102), the Second Tibetan Plateau Scientific Expedition and Research Program (2019QZKK0201), the NSFC



project (41971097 and 21661132003) and Youth Innovation Promotion Association CAS (2017099). We thank
Qomolangma Atmospheric and Environmental Observation and Research Station CAS for providing radiation data, Dr. Husi
Letu and Wenjun Tang for providing Hamawari-8 satellite radiation data. We are also grateful to all the members who took
part in the fieldwork.

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



**Figure and Captions**

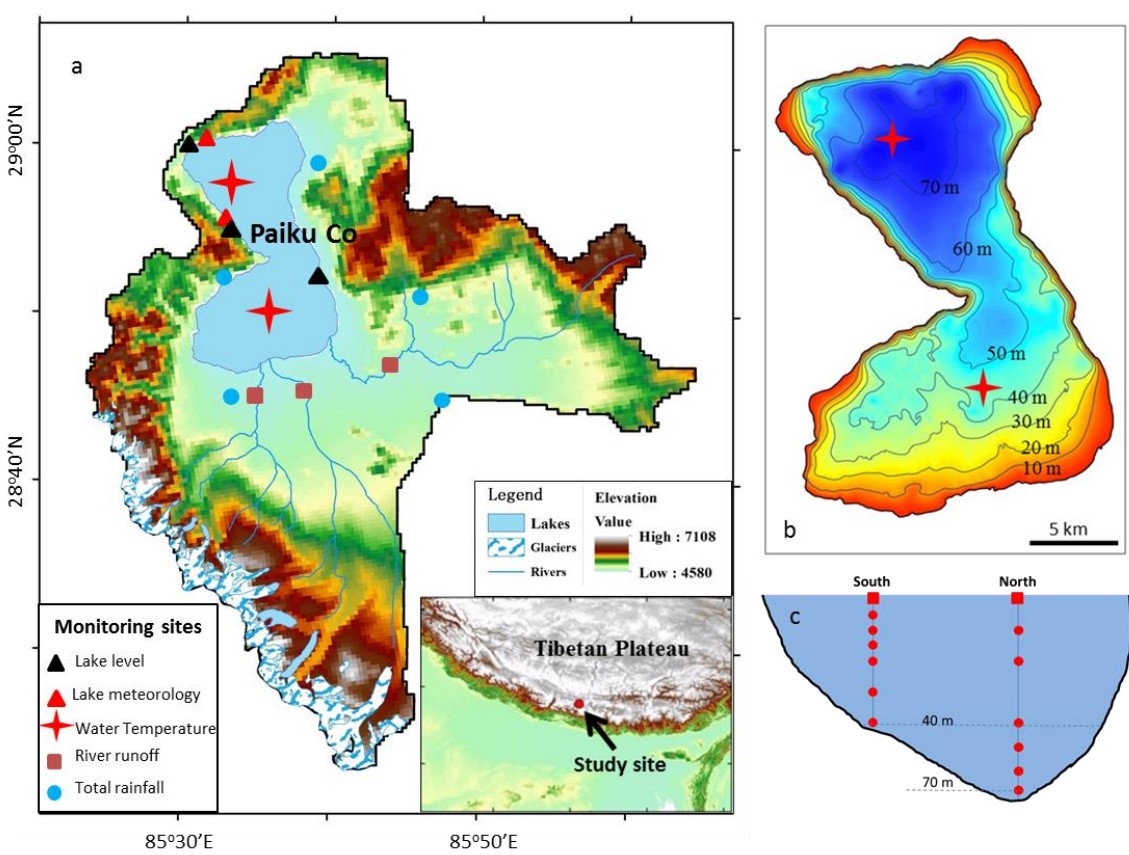

**Figure 1: Monitoring sites of lake level, hydro-meteorology, water temperature profile, runoff, and total rainfall at Paiku Co basin. a: Monitoring sites at Paiku Co basin. b: The isobath of Paiku Co and the two monitoring sites of water temperature profile. c: The water temperature monitoring at different water depth.**





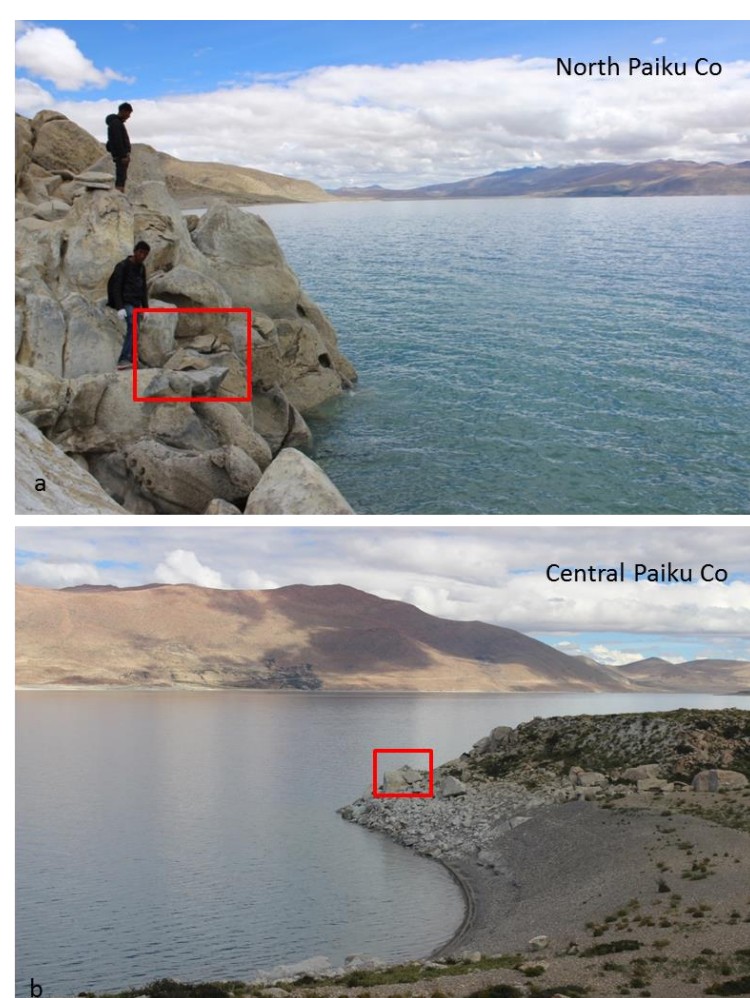

Figure 2: The monitoring site of air temperature and humidity at Paiku Co's shoreline of. a: The north shoreline.  b: The central shoreline.





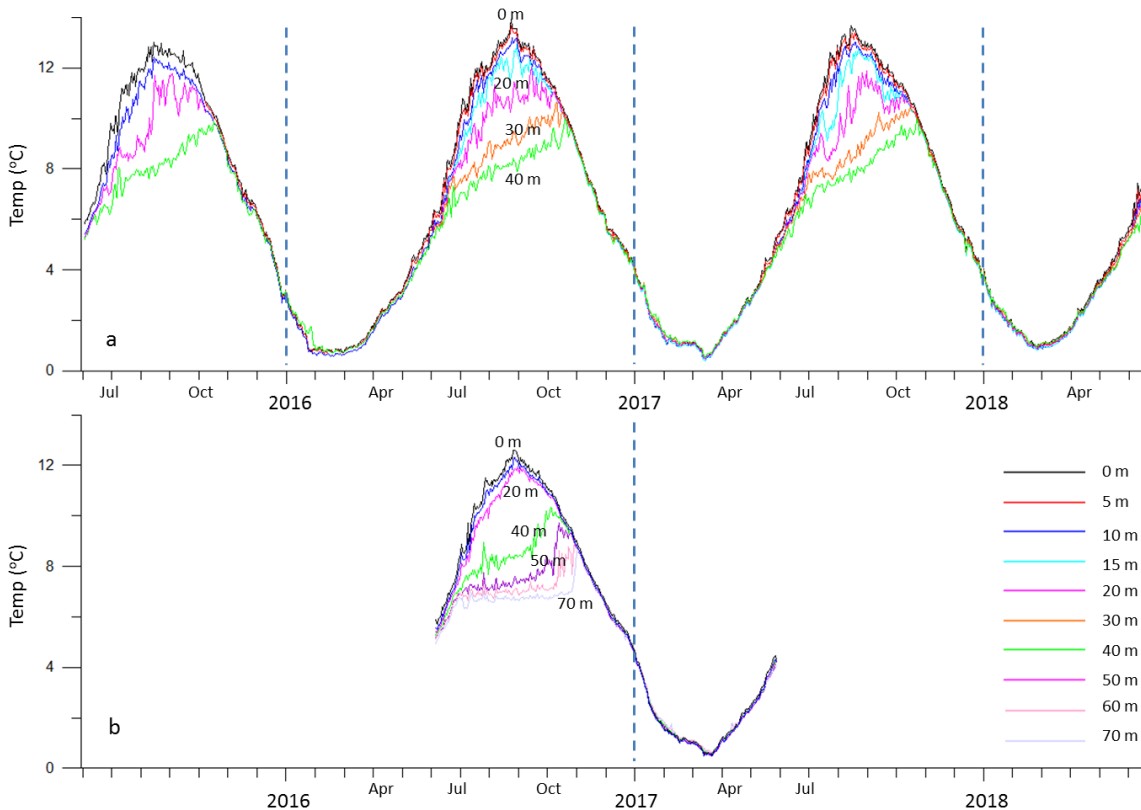


**Figure 3: Lake water temperature at different depth of Paiku Co. a: The southern Paiku Co. b: The northern Paiku Co.**

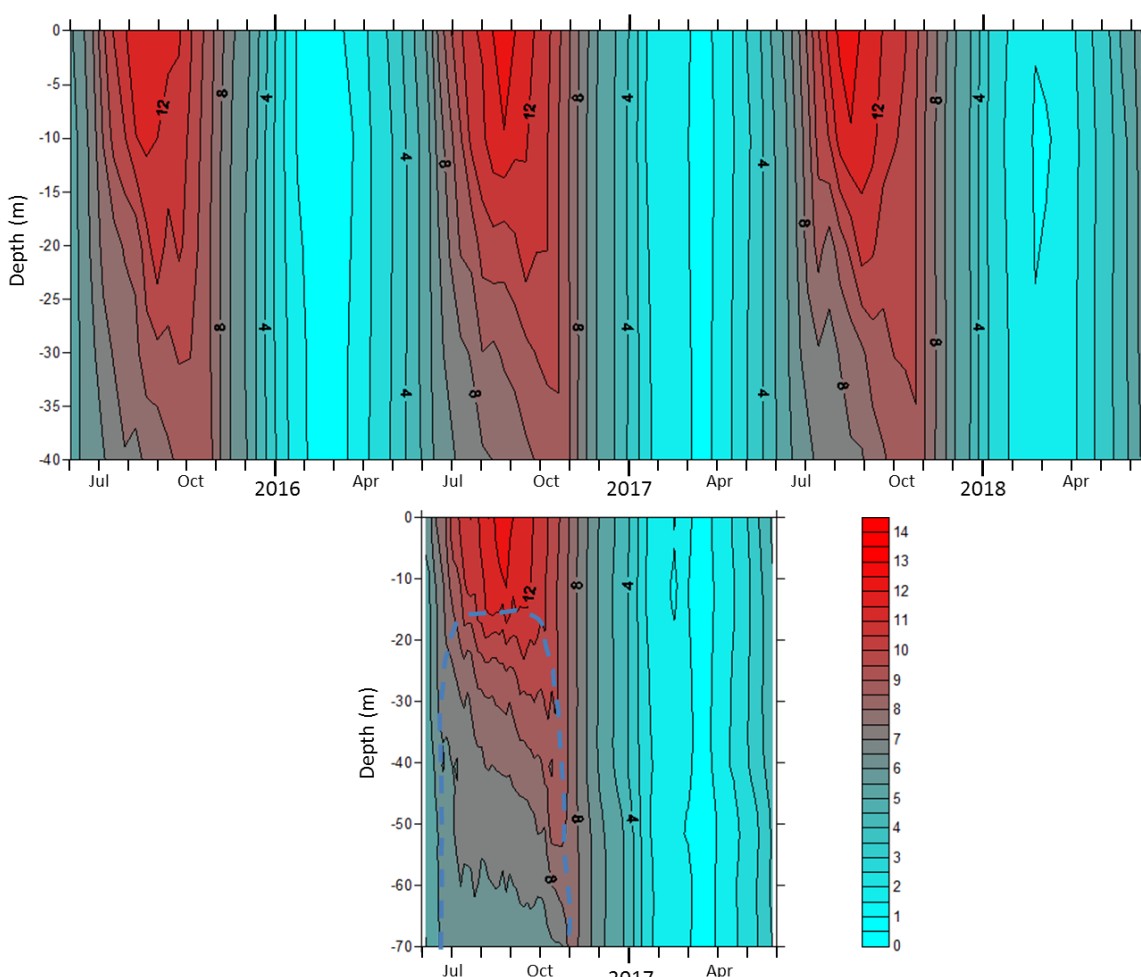

**Figure 4: Depth-time diagram of isotherm (°C) in Paiku Co's southern (upper, 42 m in depth) and northern (below, 72 m in depth) basins between June 2015 and May 2018.**

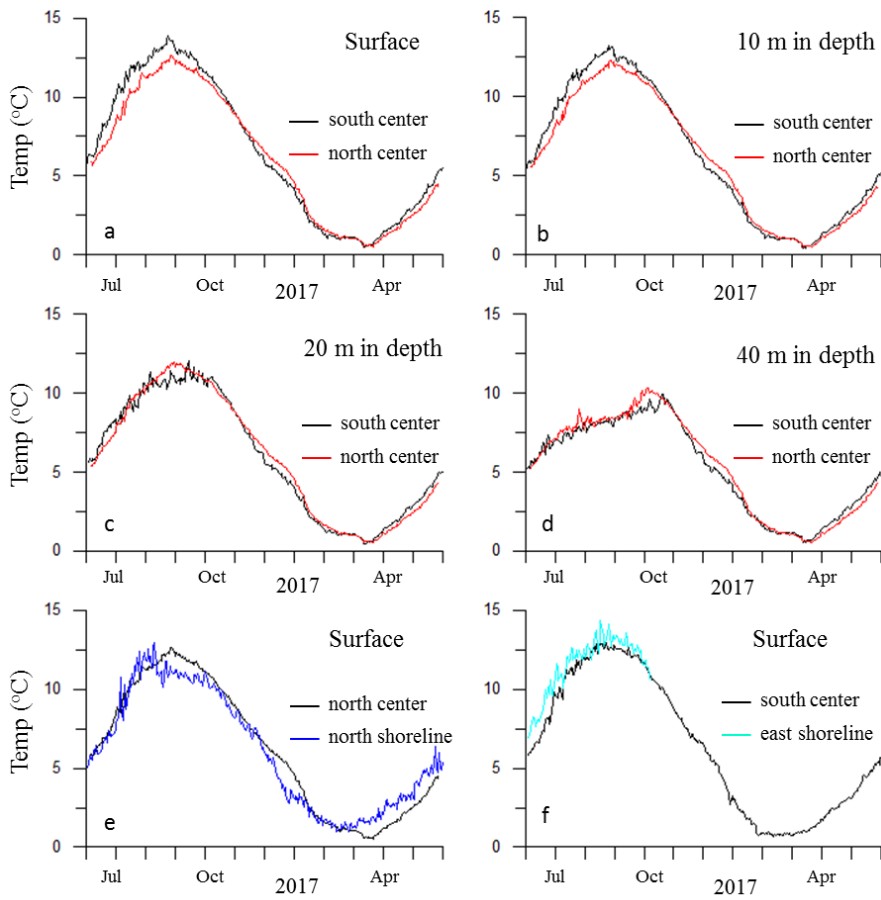

**Figure 5: A comparison of water temperature at different sites of Paiku Co. a-d: A comparison of water temperature at the depth of 0 m, 10 m, 20 m and 40 m between the southern and northern center of Paiku Co. e-f: A comparison of water temperature between lake center and shoreline.**

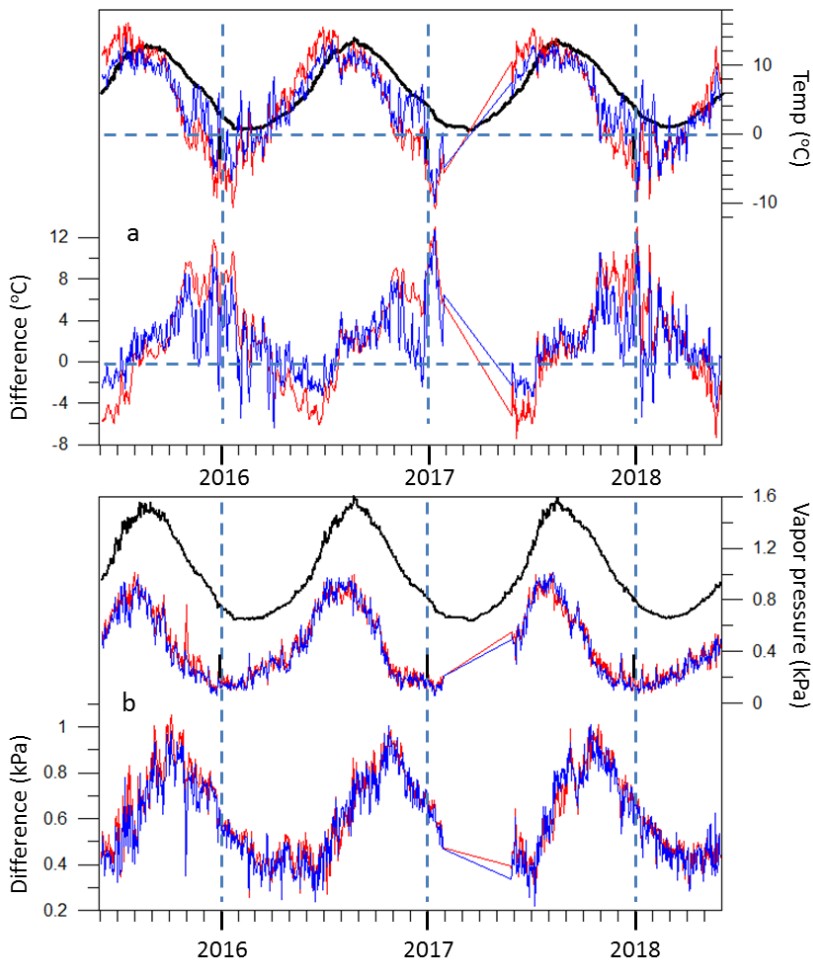


**Figure 6: Time series of hydro-meteorology at the north (blue lines) and central (red lines) shoreline of Paiku Co. a: Daily surface water temperature (black), atmosphere temperature, and their differences. b: Actual vapor pressure at lake surface (black) and the overlying atmosphere, and their differences.**

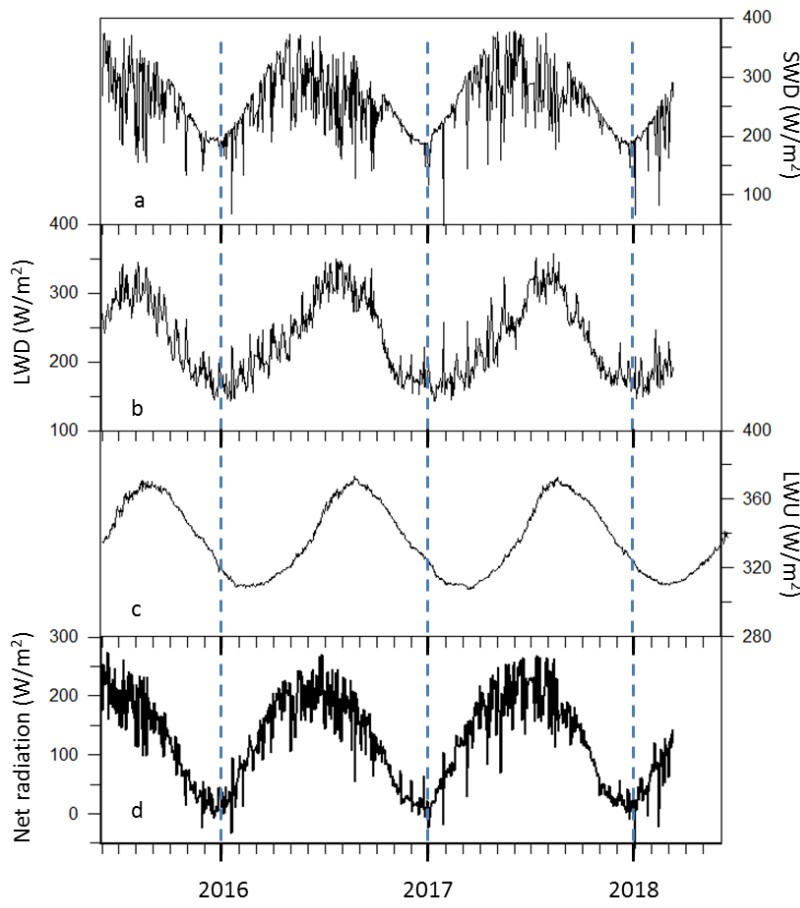

**Figure 7: Time series of daily radiation at the lake surface of Paiku Co. a: Downward shortwave radiation. b: Downward longwave radiation to lake. c: Upward longwave radiation emitted from lake surface. d: Net radiation.**

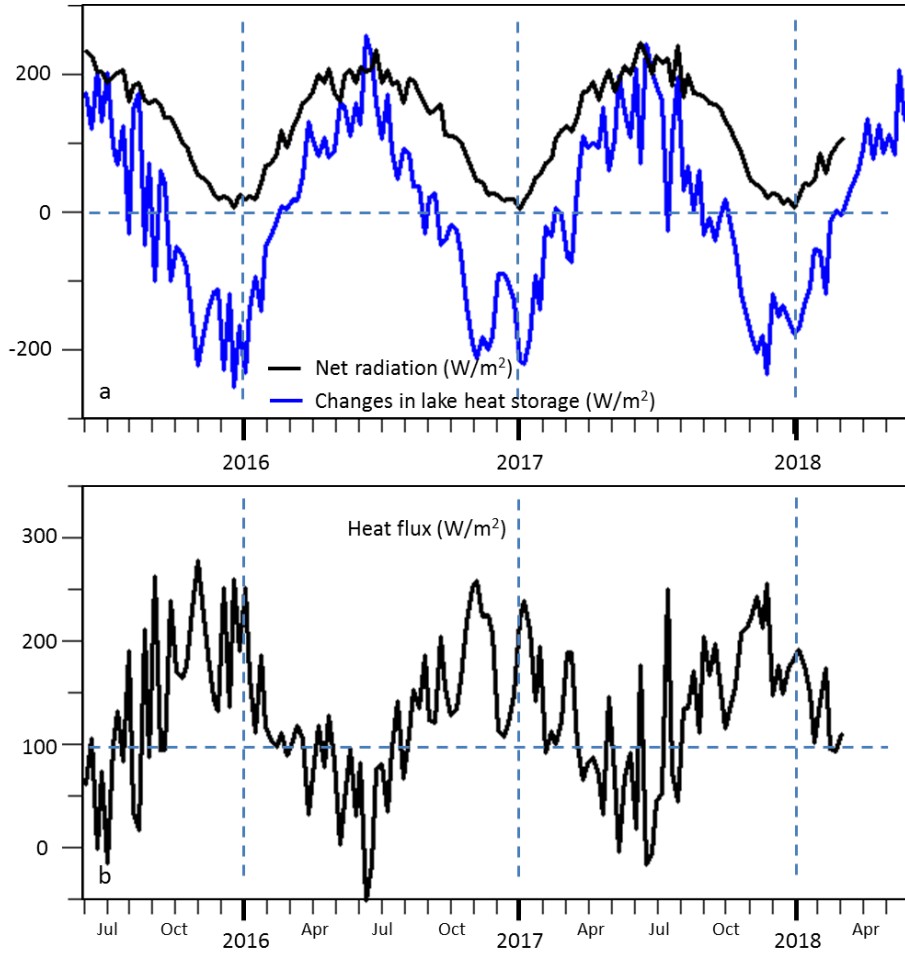

Figure 8: Time series of weekly net radiation and heat flux at the lake surface of Paiku Co. a: A comparison of weekly
net radiation with changes in lake heat storage. b: Weekly heat flux the lake surface.

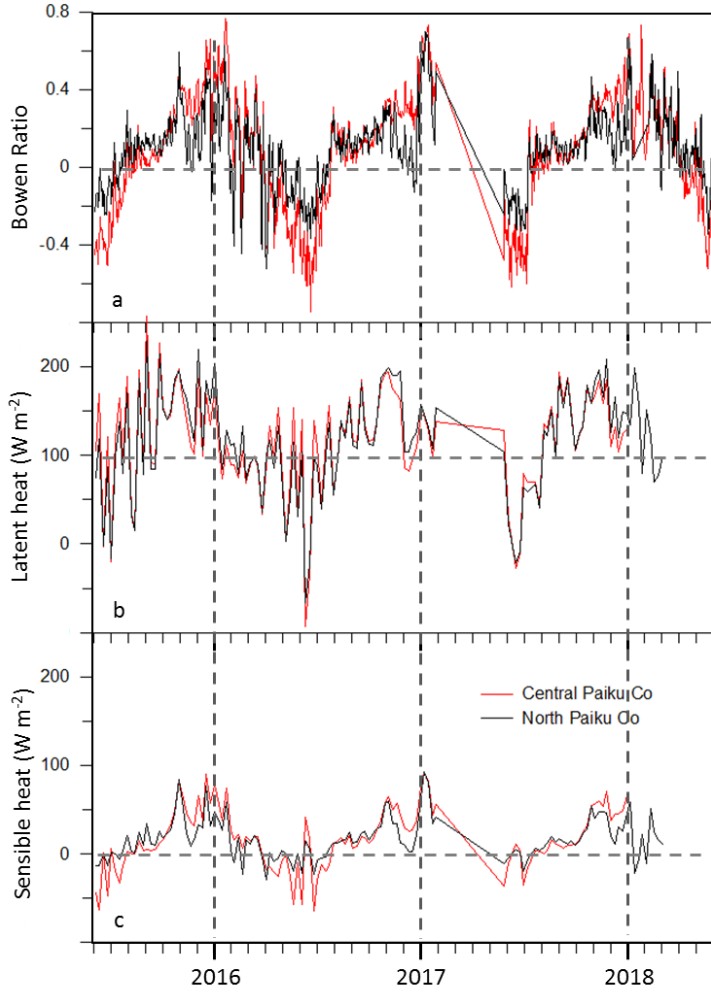

**Figure 9: Time series of Bowen ratio (a), weekly latent (b) and sensible (c) heat flux derived from the north and central shoreline of Paiku Co.**

620



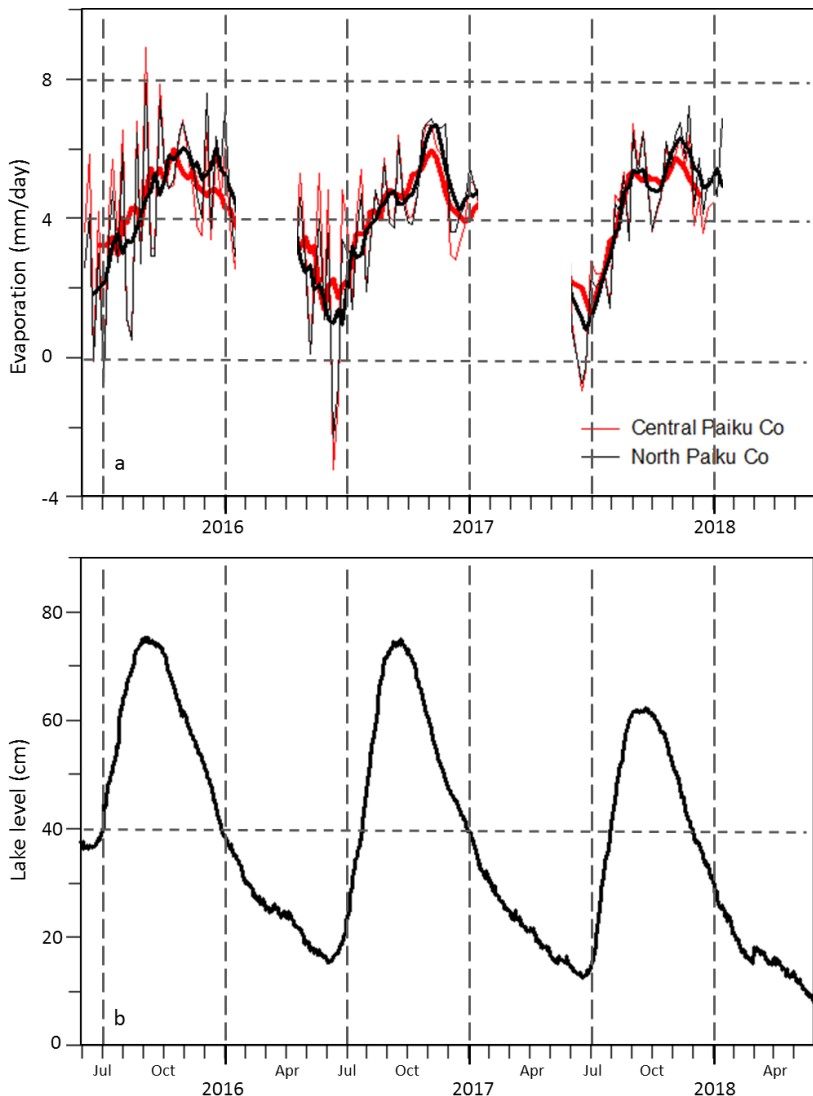

**Figure 10: Time series of weekly lake evaporation (a) and lake water level (c) at Paiku Co between June 2015 and May 2018. The thick lines (a) denote the 5-point running average. Lake evaporation derived from the north and central shoreline of Paiku Co are compared.**





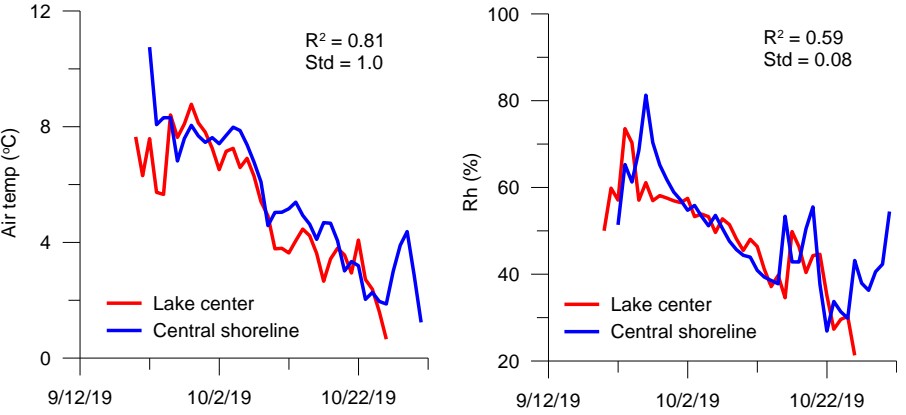

**Figure 11: A comparison of air temperature and relative humidity between shoreline and lake centre.**





**Table 1 The related information about hydro-meteorology observations**

| Parameter | Sensor | accuracy | Location | Duration |
|---|---|---|---|---|
| $T_w$ | HOBO U22-001 | 0.21 $^\circ$C | Southern basin | 2015.6-2018.5 |
| | | | Northern basin | 2016.6-2017.5 |
| $T_a$ and RH | HOBO U12-012 | 0.35 $^\circ$C<br>2.5% | North<br>Central | 2015.6-2017.1, 2017.6-2018.5 |
| $R_s$ and $R_a$ | Kipp & Zonen CNR4<br>net radiometer | 5% | Qomolangma<br>Station, CAS | 2015.6-2017.12 |

$T_w$=water temperature; $T_a$=air temperature; RH=relative humidity; $R_s$=solar radiation; $R_a$=downward long wave radiation










**Table 2 Monthly net radiation, total lake heat storage, Bowen ratio and lake evaporation between 2015 and 2017**

|     | Net energy (W·m⁻²) | | | Heat storage (W·m⁻²) | | | Bowen Ratio | | | Evaporation (mm/day) | | |
| --- | --- | --- | --- | --- | --- | --- | --- | --- | --- | --- | --- | --- |
|     | 2015 | 2016 | 2017 | 2015 | 2016 | 2017 | 2015 | 2016 | 2017 | 2015 | 2016 | 2017 |
| May |       | 188.5 | 194.8 |        | 145.2  | 138.6  |       | -0.10 |       |      | 1.72 |      |
| Jun | 217.2 | 214.3 | 224.8 | 157.3  | 191.6  | 181.8  | -0.15 | -0.24 | -0.20 | 2.40 | 0.98 | 1.81 |
| Jul | 198.0 | 185.2 | 218.1 | 123    | 101.0  | 93.4   | -0.02 | 0     | -0.04 | 2.6  | 2.89 | 3.28 |
| Aug | 170.4 | 178.6 | 177.2 | 62.3   | 32.4   | 39.3   | 0.11  | 0.13  | 0.11  | 3.33 | 4.47 | 4.31 |
| Sep | 148.4 | 140.2 | 154.1 | -24.6  | -10.7  | -15.4  | 0.13  | 0.14  | 0.08  | 5.29 | 4.57 | 5.40 |
| Oct | 89.1  | 91.4  | 92.4  | -115   | -87.1  | -86.4  | 0.23  | 0.20  | 0.20  | 5.67 | 5.12 | 5.15 |
| Nov | 34.7  | 34.9  | 34.3  | -140.6 | -193.7 | -199.5 | 0.17  | 0.18  | 0.24  | 5.12 | 6.69 | 6.51 |
| Dec | 17.7  | 16.6  | 19.7  | -192   | -125.3 | -148.5 | 0.26  | 0.14  | 0.20  | 5.78 | 4.22 | 4.88 |











**Table 3 Runoff (m$^3$/s) at the three main rivers at Paiku Co basin in spring and autumn between 2015 and 2017 and their total contribution to lake level increase (mm/day). The measuring dates of runoff are shown in brackets.**

| Rivers | Runoff-2015 | | Runoff-2016 | | Runoff-2017 | |
|---|---|---|---|---|---|---|
| | Spring | Autumn | Spring | Autumn | Spring | Autumn |
| | (6.1~6.2) | (10.6~10.7) | (6.2) | (10.11~10.13) | (5.25~5.28) | (10.14~10.16) |
| Bulaqu | 2.3 | 2.1 | 0.8 | 0.7 | 0.5 | 0.7 |
| Daqu | 0.4 | 2.8 | 1.1 | 1 | 0.5 | 1.2 |
| Barixiongqu | 0.2 | 0.4 | 0.1 | 0.5 | 0.1 | 0.5 |
| Total contribution | 0.89 | 1.64 | 0.62 | 0.71 | 0.62 | 0.74 |

Total contribution is calculated according to the total runoff of the three main rivers and lake area