# Peer review of "Contrasting hydrological and thermal intensities determine seasonal lake-level variations – A case study at Paiku Co on the southern Tibetan Plateau"

_Hydrology and Earth System Sciences, 2020_

## Referee Comment (RC1) · Anonymous Referee #1 · 20 Jul 2020

This paper reported the seasonal changes of lake water profile, lake levels, surface heat budget, and evaporations by three years in-situ observation data. They showed very interesting characteristics representing lake environment in southern edge of the TP, that gives us the hints to understand basic processes of heat/water budget of mountain lakes under Indian monsoon climate. As authors introduced in the introduction, lakes on the TP are changing. It is very important to reveal that how the global environment change could modify the lake environment through land-atmosphere interaction. The contents showed basic timelines of observed data with estimated heat budget and evaporation amount, and natures could be easily captured by figures. However, many key mechanisms are discussed by speculations without in-depth examination/comparisons to previous studies in the TP. This is because the study did not set clear objectives. Therefore, the title is also uncoordinated. For instance, do authors concern about the lake area (level) changes of Paiku Co ? Figure 10 shows that lake level show small seasonal variation (within 1m), but do you think this is critical? Or, authors investigated large evaporation rate instead of previous studies? Readers can not understand how the Fig. 10 differs from other lake or even from ground in the TP. If the HESS request level of paper as scientific article instead of "report", I would like to suggest that paper needs fundamental revisions with clear objectives and results based on additional in-depth analysis.

For the lake dynamics by means of hydrometeorology, following points need to be examined. 1) Water temperature profiles were almost homogeneity during Oct.- June (non-monsoon season), and author explained by "fully mixing" without any analysis. Please proof it physically using surface wind speed and variability conditions and water mixing theory. It is curious that such mixing occurred suddenly. In the central TP, large diurnal wind changes are found in winter due to the coupling of upper strong STJ and boundary layer development. Any relation to the seasonal change of atmospheric circulation ? 2) Seasonal change of water level should be explained by seasonal change of water budget, including precipitation, river runoff/inflow and surface water inflow. Even there are lack of areal in-situ measurements, some parameters could be estimated by previous studies or literature. This also links to Av calculation as mentioned in 3). I could not see precipitation records, but the Rn sequence demonstrated that rain season is not clear compare to southern Himalayas and central plateau. If the impact of monsoon is small with fair/non-freeze weather, location of the lake may represent local dry climate behind the Himalayas where lee-side subsidence prevails, and that would characterize evaporation rate at Paiku Co. 3) To consider the heat budget of the lake, especially for the condition of thermocline, advection of cold (snow/glacier-melt) water associated with river/surface inflow need to be considered. This paper only compares the heat budget at water surface, and conclude the evaporation as a ley parameter to affect lake level seasonality. Is there no effects of glacier melt water (they are illus-
trated in Fig. 1a) or monsoon precipitation inflow to establish lake temperature profile and lake level seasonality? Diurnal change of river level according to the glacier melt is observed by previous studies. There are some indication at the bottom temperature of northern point in Fig. 3b. At L115, please proof that Av can be ignored. Authors should not avoid those issues to analyze if they focus on the water cycle and environmental changes on the TP as introduced in Chapter. 1.

Minor comments are as follows. (sorry that order is not as in the paper)

> There are no previous studies in Paiku Co. ? Need reviews. > Water temperature sensors in the upper profile are shaded? Or, how deep the insolation can penetrate the water at the target lake? > L176ãĂĂSmall water temperature gradient is explained by cold air temperature. This is strange. Air temperature change is due to latent heat from the surface or advection. Enough radiation could increase the water temperature even the air temperature is cold with weak winds. > L175-180 Those are speculations, not results. > L138ãĂĂ"input data were averaged at weekly interval"ãĂĂDoes heat budget screened by the wind direction by instantaneous data then averaged? > Units in Fig. 10 are mm/d, cm and it is m3/s in Table 3. Please unify them to capture accurate water balance. > Title of 3.3 "Lake hydrometeorology" is vague. > L215ãĂĂ"There was a ∼1.5 month lag between lake surface temperature and air temperature." Is not clear. > I could not understand the meaning to show the Fig.6. > L230ãĂĂ"Downward shortwave radiation at Paiku Co had an annual average of 251.8 WÂům-2 (Fig. 7), which is slightly higher than the TP average due to its lower latitude (Yang et al., 2009)." What is the TP average? Effects of Indian monsoon is stronger in southern TP in general, and cloudy weather may reduce the insolation. Or, the observation represent local weather in the valley? > L230-237ãĂĂDiscussions are not clear due to mixture of seasonal change and annual average. Why the rainy season is not clear? > L17, L248" "a deep lake". Many discussion attribute the characteristics to the depth of lake without examination. Manly lakes over the TP are shallower than the target lake? Please review that how the depth of lake over the TP characterize the lake temperature condition.

---

## Author Comment (AC1) · 29 Jul 2020

This paper reported the seasonal changes of lake water profile, lake levels, surface heat budget, and evaporations by three years in-situ observation data. They showed very interesting characteristics representing lake environment in southern edge of the TP, that gives us the hints to understand basic processes of heat/water budget of mountain lakes under Indian monsoon climate. As authors introduced in the introduction, lakes on the TP are changing. It is very important to reveal that how the global environment change could modify the lake environment through land-atmosphere interaction. The contents showed basic timelines of observed data with estimated heat budget and

evaporation amount, and natures could be easily captured by figures.

Response: We are grateful to the reviewer's comments. We will consider these comments carefully. The main responses to these comments are shown as the following:

However, many key mechanisms are discussed by speculations without in-depth examination/comparisons to previous studies in the TP. This is because the study did not set clear objectives. Therefore, the title is also uncoordinated. For instance, do authors concern about the lake area (level) changes of Paiku Co? Figure 10 shows that lake level show small seasonal variation (within 1m), but do you think this is critical? Or, authors investigated large evaporation rate instead of previous studies? Readers can not understand how the Fig. 10 differs from other lake or even from ground in the TP. If the HESS request level of paper as scientific article instead of "report", I would like to suggest that paper needs fundamental revisions with clear objectives and results based on additional in-depth analysis.

Response: The main objective of this study is to quantify lake evaporation throughout the year based on energy budget method (Section 3) and its impact on seasonal lake level changes (Section 4.4). Until now, lake evaporation during the late autumn and early winter is not typically investigated on the TP because it is difficult to install and maintain measurement platform due to the harsh natural conditions and the influence of lake ice. As a result, how lake evaporation affects seasonal lake level changes remains unclear due to lack of comprehensive observation of lake water budget. We will add one paragraph in the introduction about endorheic lake level seasonality on the TP. 'Compared with numerous studies of inter-annual to decadal lake changes, seasonal lake level changes and the associated hydrological processes on the Tibetan Plateau (TP) are still less understood. Phan et al. (2012) showed that seasonal lake level variations in the southern TP are much larger than that in the northern and western TP. In-situ observations gave more details of seasonal lake level variations (Lei et al., 2017). One striking feature is the different amplitude of seasonal water level variations, that is, deep lakes usually exhibited considerably greater lake level variations than

shallow lakes. For example, Zhari Namco and Nam Co, two large and deep lakes on the central TP (Wang et al., 2009, 2010), exhibited significant water level increase by 0.3∼0.6 m during the summer monsoon season and a similar magnitude of lake level reduction by 0.3∼0.5 m during post-monsoon season between 2010 and 2014. For the two nearby small and shallow lakes, Dawa Co and Bam Co, although there was a similar pattern of lake seasonality, the amplitude of seasonal lake level variations was considerably smaller than the two large and deep lakes (Lei et al., 2017).' The main causes for the different amplitude of lake level changes are still not investigated in previous studies, which is the main topic of our study. We will change the title of the paper to: 'Contrasting hydrological and thermal intensities determine seasonal lake level variations Ì A case study at Paiku Co in the central Himalayas' Lake level changes of Paiku Co (Fig. 10) and other lakes will be shown in a new figure (Fig. 11) to show the different amplitude of seasonal lake level changes. Previous studies about lake evaporation on the TP have been reviewed and compared in Section 4.3.

For the lake dynamics by means of hydrometeorology, following points need to be examined. 1) Water temperature profiles were almost homogeneity during Oct.- June (non-monsoon season), and author explained by "fully mixing" without any analysis. Please proof it physically using surface wind speed and variability conditions and water mixing theory. It is curious that such mixing occurred suddenly. In the central TP, large diurnal wind changes are found in winter due to the coupling of upper strong STJ and boundary layer development. Any relation to the seasonal change of atmospheric circulation ?

Response: The thermal structure of Paiku Co has been addressed in the section 3.1. Lake water temperature profile can be taken as a proxy of lake water mixing. In summer, the lake water is stratified according to the dramatic temperature gradient between surface and bottom. In October, the lake stratification is weakened due to the decreased temperature gradient. The water temperature isobaths show that lake mixing was deepened gradually from early October to later October. Since the late October,

the lake water is totally mixed because the vertical temperature gradient suddenly disappeared. If the lake water can not completely mixed, the temperature gradient should always exist. As we have addressed in the main text, the lake mixing is mainly forced by wind disturbance and water convection. This is also the classic theory of lake water circulation, which has been addressed in many publications and books (i.e. Wetzel, 2001). Clear Lake stratification at Paiku Co occurred in late June or early July, which corresponded to a significant reduction in wind speed. The timing of the stratification breaking down occurred in late October, which corresponded well to significantly increased wind speed. The potential relationship with atmosphere circulation will be discussed.

2) Seasonal change of water level should be explained by seasonal change of water budget, including precipitation, river runoff/inflow and surface water inflow. Even there are lack of areal in-situ measurements, some parameters could be estimated by previous studies or literature. This also links to Av calculation as mentioned in 3). I could not see precipitation records, but the Rn sequence demonstrated that rain season is not clear compare to southern Himalayas and central plateau. If the impact of monsoon is small with fair/non-freeze weather, location of the lake may represent local dry climate behind the Himalayas where lee-side subsidence prevails, and that would characterize evaporation rate at Paiku Co.

Response: We do not focus on all components of lake water budget in this study. The main purpose of this study is to quantify lake evaporation and its impact on seasonal lake level changes. Therefore, precipitation, river runoff and other surface water inflow is not shown. We agree that the impact of monsoon precipitation is small in this dry area and location of lake represent local dry climate behind the Himalayas where subsidence prevails. We will add this point in the revision.

3) To consider the heat budget of the lake, especially for the condition of thermocline, advection of cold (snow/glacier-melt) water associated with river/surface inflow need to be considered. This paper only compares the heat budget at water surface, and

conclude the evaporation as a ley parameter to affect lake level seasonality. Is there no effects of glacier melt water (they are illustrated in Fig. 1a) or monsoon precipitation inflow to establish lake temperature profile and lake level seasonality? Diurnal change of river level according to the glacier melt is observed by previous studies. There are some indication at the bottom temperature of northern point in Fig. 3b. At L115, please proof that Av can be ignored. Authors should not avoid those issues to analyze if they focus on the water cycle and environmental changes on the TP as introduced in Chapter. 1.

Response: We will add one paragraph in Section 2.3 to evaluate the impact of Av on the lake energy budget. Av can be estimated according to total river discharge (Fig. 1) and the water temperature difference between river and lake. Lake water temperature was almost same to river water temperature between April and June, but 2-4oC higher between July and December. As a deep lake, total river discharge to Paiku Co was about 800-900 mm water equivalent to lake level and accounted for 2-2.5% of total lake water storage. The river discharge can accumulatively decrease lake water temperature by ∼0.1 oC in a year. Therefore, as a deep lake, the influence of river discharge on the total lake heat storage at Paiku Co is very small and can be neglected. Therefore, we do not consider the influence of Av on the lake energy budget in this study.

Minor comments: There are no previous studies in Paiku Co. ? Need reviews.

Response: We will review previous studies at Paiku Co. We did not do this because we have reviewed it in a previous publication about Paiku Co. Lei, Y., Yao, T., Yang, K., et al.: An integrated investigation of lake storage and water level changes in the Paiku Co basin, central Himalayas, J. Hydrol., 562, 599–608.

Water temperature sensors in the upper profile are shaded? Or, how deep the insolation can penetrate the water at the target lake?

Response: We do not think water temperature sensors in the upper profile are shaded.

[Figure]

As we have illustrated in section 2.2, the first water temperature sensors are fixed below buoy at the water depth of ∼0.5 m. The other sensors were fixed on the rope which was tied to an anchor at the lake bottom. The transparency of the lake is not measured until now.

L176AAËŸ Small water temperature gradient is explained by cold air temperature. This is strange. Air temperature change is due to latent heat from the surface or advection. Enough radiation could increase the water temperature even the air temperature is cold with weak winds.

Response: We will discuss it in more detailed in the revision.

L175-180 Those are speculations, not results.

Response: We will discuss it in more detailed in the revision.

L138 "input data were averaged at weekly interval. Does heat budget screened by the wind direction by instantaneous data then averaged?

Response: Wind speed and direction is not used during the calculation of energy budget and lake evaporation.

Units in Fig. 10 are mm/d, cm and it is m3/s in Table 3. Please unify them to capture accurate water balance.

Response: Thanks for pointing out this. We will unify them in the revision.

Title of 3.3 "Lake hydrometeorology" is vague.

Response: We will change it to 'Air temperature and humidity over the lake surface'

L215ãËŸAAËŸ "There was a 1.5 month lag between lake surface temperature and air temperature." Is not clear. Response: We will change it to 'between the maximum surface water temperature and maximum air temperature'.

I could not understand the meaning to show the Fig.6.

Response: We will consider it further to remove or leave it.

L230 "Downward shortwave radiation at Paiku Co had an annual average of 251.8W°um-2 (Fig. 7), which is slightly higher than the TP average due to its lower latitude (Yang et al., 2009)." What is the TP average? Effects of Indian monsoon is stronger in southern TP in general, and cloudy weather may reduce the insolation. Or, the observation represent local weather in the valley?

Response: Mean annual precipitation at Paiku Co is about 150-200 mm. So we agree that the observation represent local weather in the valley.

L230-237 Discussions are not clear due to mixture of seasonal change and annual average. Why the rainy season is not clear?

Response: As we have shown in Fig. 7, the rainy season is not clear because the precipitation at Paiku Co is low.

L17, L248" "a deep lake". Many discussion attribute the characteristics to the depth of lake without examination. Manly lakes over the TP are shallower than the target lake? Please review that how the depth of lake over the TP characterize the lake temperature condition.

Response: As we have shown in section 4.3, seasonal changes in lake evaporation is related to the mean lake water depth and the lake heat storage. The mean water depths of the lakes have been mentioned in Section 4.3. Deep lake can store more energy in spring and summer and release more energy to the overlying atmosphere, which can have dramatic impact on the season pattern of lake evaporation.
* * *
[Figure]

Fig.1 Comparison of water temperature (ºC)between Bulaqu river (red line) and Paik Co (blue)

**Fig. 1.** Comparison of water temperature between Bulaqu river (red line) and Paik Co (blue)

---

## Referee Comment (RC2) · Anonymous Referee #2 · 5 Aug 2020

The manuscript "Thermal regime, energy budget and lake evaporation at Paiku Co, a deep alpine lake in the central Himalays" use in-situ measurements to analyze the energy budget components and obtain the evaporation amounts of Lake Paiku Co. As lake measurements are very limited on the Tibetan Plateau and most of the measurements are in the central or east parts of the Tibetan Plateau, the manuscript shows significance in describing clearly the thermal regime, energy budget and lake evaporation of a western lake on the Tibetan Plateau by in situ measurements. The structure of the manuscript is well-organized; the analysis of the processes are observation-based and reasonable; I consider the manuscript to be appropriate to be published in the HESS journal after a minor revision. The detailed comments are given as belows:

[Figure]

(1)In line 25-26ïïjŇthe last sentence in abstract seems has not clear connection with the other contents, I suggest to revise the sentence to keep it coherent with previous contents. (2)In line 120-125, in equation (2), Ra is the downward longwave radiation to lake, while in equation (3) Ra is rewritten as the longwave radiation from lake. Here, in equation (3) and line 125, I think it should be Rw. (3)In line 129-130, as daily averaged water temperature is used, in addition to the surface mixing by wind and convection, Here I suggest to add information that "there exists surface warming during the day and surface cooling at night for high elevation lakes, thus the two uncertainties by surface warming and cooling can cancel each other at a temporal resolution of daily." (4)In line 161, I suggest to use "period" instead of "time" here; in line 262, it should be "in low values" rather than "in low value"; Figure 3 caption, "at different depths" rather than "at different depth"; Figure 4, a unit of (OC) should be added for the colorbar.

––––––––––––––––––––––––––

---

## Referee Comment (RC3) · Anonymous Referee #3 · 18 Aug 2020

General comments: This study reports lake evaporation in the Tibetan Plateau and explains its seasonal variation through energy storage change within lake water. Since the Tibetan Plateau has been one of the least studied areas, lake evaporation study in this region is welcomed and worth publication in HESS. However, I find there are substantial problems in this study. That is the accuracy of the evaporation in their study. I pointed out this issue in the review at the time of their previous submission. Unfortunately, they failed to solve this problem and the manuscript was rejected for publication. The authors have added new data at the lake center to compare their measurements on the shoreline. This is good. On the other hand, their treatment of the comparison is not enough to convince readers that their evaporation estimates are

accurate and reliable. Details are listed below.

Major comments. The authors gave error estimates of evaporation in Section 4.2. They selected (1) net radiation and (2) water temperature differences between the northern and southern basins as the error sources. I think they should add other relevant error sources. They include, first, the use of air temperature and humidity on the shoreline instead of those above lake water. They compare the measurements on the shore and on the lake in Fig. 11 and conclude they are "very similar. . ... data from the shoreline. . . can be used to represent the general condition of the whole lake. . .". This is new information and should be used to evaluate error from using onshore measurements. What they should do is to determine the RMS error of temperature and humidity measurements on the shore and used them to estimate errors of Bowen ration and sensible and latent heat fluxes. This can be added to the final error evaluation. The second error source they should consider is the use of water temperature instead of surface temperature. They claim that "the daily average between them is very similar. . .". But no supporting evidence is shown. In fact, previous studies do indicate a difference between the two even for mean values for day or longer. The authors should accept this and add this as one of the error sources for the evaporation estimates. The third error source is the error in the energy storage estimation. To estimate energy storage, spatial mean water temperature profile, spatial mean water level, water level-volume relation, water level-surface area relation are needed. Since they are all based on some kind of measurement, there are always errors (measurement errors as well as sampling errors). They should be considered.

[Minor comments] Introduction. - The originality of the study: It is not clear what the original contribution of this study is. Authors claim that previous studies do not provide evaporation throughout a year. But in their study also, evaporation was not determined during the winter period. So it is not quite new. Please make it clear what is missing in previous studies and why their studies are needed based on a comprehensive review of previous studies. Also, these points should be reflected in discussion and conclusions. - Importance of TP lakes. The authors explain the abundance of lakes in TP. Then authors should add relevance of these lakes for TP (or even for larger areas). - Eddy correlation method. The authors mention that it is not suitable for long-term measurements. I believe this statement was correct perhaps 20 years ago. But it is easy to see the results of long-term measurements based on the eddy correlation method in the literature as well as datasets on the flux net sites. - Direct measurements of lake evaporation: I do not think the Bowen ratio method is in the category of direct measurements. It measures energy balance and evaporation is obtained only indirectly as one of the residuals of the energy balance equation. It relies on the similarity between temperature and humidity profiles.

- L96-97. "...therefore the meteorological condition over the lake surface can be recorded". This cannot be true without evidence. I made the same argument in previous reviews so please read it again. What I would suggest is to acknowledge that it is not the location where measurements should be made, but the measurements were used as a proxy of the above-lake measurements, and validity of this proxy will be discussed in section...(see major comment)

- L106 "weekly averaged radiation..."; What are the possible errors to apply the Bowen ratio method with weekly averaged data? The Bowen ratio equation (4) was derived from two profile equations of H and LE. Profiles equations derived by applying the similarity theory are valid for the steady-state condition under certain stability. So we generally apply them for 30-min to hourly mean values. For practical purposes, we apply them for daily data assuming neutral stability but strictly speaking, this is not valid since profile equations are not linear and therefore simple time averaging does not yield valid equations for the given averaging period.

- L114-115. "the influence of river discharge...can be neglected" You cannot say this without supporting evidence. The authors should give and compare lake storage and river discharge.

- L114-115. "…therefore…we do not consider …G.." There is no mention of the reason why G can be neglected.

- L153-155, "…reduction in wind speed (data not shown)"; As I recall from the reply of authors to the reviewers' comments in the previous version, authors do not have wind speed data. Then the statement heat of "data not shown" is misleading. When we see this statement, we tend to believe that there were data and authors checked them to validate what is written in the manuscript even though they are not shown in the manuscript with a figure or a table. If you do not have data, then you should not mention wind influence as if it was based on data. There are similar statements on wind speed here and there in the manuscript. Authors need to remove them or change expressions. Alternatively, authors could rely on wind speed from reanalysis data. However, the reliability of any reanalysis data set should be established first (perhaps by referring to previous studies) for the study area before they can use the reanalysis data.

- L179 "..but also the bottom water"; I do not understand what authors want to claim.

- L200-201; "....Lower temperature gradient caused stronger water convection....."; I do not understand the logic in this part. I assume water convection is stronger when the vertical gradient is larger.

- L204-205 "(Fig.1)"; Fig. 1 shows the locations of water level loggers but authors are talking about water temperature. In Fig.1, there are also the locations of water temperature measurements. This is confusing.

- L208-209 "....large errors can result if only water temperature data collected at the shoreline are used to calculate lake heat storage and energy budget."; Similarly, errors can result if only water temperature data collected at the center of the lake are used. The authors should acknowledge this possibility to make analysis accordingly (see major comment).

[Figure]

- L223 " Indian summer monsoon precipitation"; there is no mention of heat advection due to precipitation in the application of the Bowen ratio method. All energy sources that are used for the turbulent neat fluxes should be considered and mentioned.

- L257-259 " Negative value...., indicating the lake water absorbed energy from the overlying atmosphere. Positive value...., indicating the lake water released energy to the overlying atmosphere." These statements are not correct. The sign of the Bowen ratio simply indicates whether the fluxes are in the same direction (positive), or different direction (negative).

- L274-276 " Lake evaporation between middle January and April is not determined... "; Why not? The authors do give latent heat flux for this period. If it is not certain whether the lake surface is covered with solid water or liquid water, then authors could give two values of evaporation. One in the case of the ice surface, and another one in the case of liquid water surface. The true evaporation is somewhere in between. This can be used together with the evaporation estimate obtained by assuming it is the same as water level change in L290-291.

- L288 ".....lake ice can effectively prohibit evaporation."; Is this true? How about sublimation? Is the latent heat flux on ice-covered lake zero? The authors could add references to support their statement.

- L290-291 "Assuming lake evaporation between January and April is equal to lake level decrease ..."; the Authors should provide an error estimate of evaporation based on this assumption. Errors due to lake level measurements, mean lake water level estimation, water level-volume relation, water level-lake surface area relation, etc.

- L293 "20.4%"; the Authors should explain how this ratio was derived.

- L299 "We set up a platform in the southern centre of Paiku Co"; The location should also be shown in Fig.1

- L302 "....fluctuated very similarly between..."; This is a subjective statement. In fact, I

do not quite agree that they are VERY similar. A better presentation would be the determination of the difference between the two measurements and its error propagation into Bowen ratio and flux estimates (see major comment).

- L303 "...can be used to represent the general condition of the whole lake..."; This statement is based on a superficial analysis. It should be based on the error propagation analysis mentioned above (see also major comment).

- L318-319 "Although there is some spatial difference, the similar seasonal patterns of energy budget and lake evaporation at different sites indicate that our results are reliable."; Just like L208-209, authors should acknowledge the difference and estimate the magnitude of the error due to spatial difference, rather than ignoring the difference by simply saying "reliable". In fact, authors do give error estimates in L329-344. Thus the statement of "reliable" is not quite consistent with the error estimates.

- L320, Section 4.2. Here authors should give all possible error sources and their likely magnitude, and use them to give a total error in their evaporation estimates. They should include, among others, the error due to the use of temperature and humidity measurements at the shoreline. Also, when they talk about annual evaporation, there are possible errors due to the assumption in L290-291 (see major comment).

- L326 "very similar seasonal fluctuations (R2=0.55....with standard deviation of 23.9 WÂům-2.)"; With R^2=0.55, I do not think it is VERY similar. Why the standard deviation? ãĂĂRMS error is a more appropriate indicator of the similarity.

- L326-327 " Assuming approximately 70% of the net radiation was consumed by lake evaporation (Lazhu et al., 2016)........ ∼74.5 mm per year "; This percentage is from a different lake. Why not use estimates for Paiku Co. given in Table 2? The authors should explain how 74.5 mm/year was derived.

- L345- " Uncertainty of lake evaporation in this study was also validated by comparing lake level changes"; Just like the case in L290-291, authors should provide error

estimates for the evaporation estimates based on lake level measurements. Possible advection due to precipitation should be addressed as commented above for L223.

- L352 "As shown in Table 3, runoff at the three large rivers can contribute to lake level increase by 0.7∼1.6"; Runoff values in Table 3 are for a short period. Can you use them to estimate monthly runoff?

- L355 "To further explore the impact of lake heat storage on the seasonal pattern of lake evaporation...."; Authors should summarize at the end of section 4.3, what kind of new findings were obtained on the impact of heat storage to lake evaporation from their measurements/analyses and comparison with previous studies. The phase shift of lake evaporation due to lake heat storage is in a way common knowledge. We would like to hear something new here. How about differences among the TP lakes? For example, in the introduction, the authors mention the difference of lake size change between the interior TP and southern TP. Any new findings on this point? Also try to make clear the relation between the statements made in the introduction (e.g., L58-67) and those in this section. You do not have to say similar things in different parts of the manuscript.

- Section 4.3; In addition to comparison with other lakes in TP, authors may want to address the difference of the TP lakes in comparison with other lakes in the world. What are special about the TP lakes? Are there similar lakes in other parts of the world? What are the controlling factors to make them similar/different?

- L359 "2011-2012", L362 "2013-2015"; Evaporation estimates for these periods are continuous even during the winter season? Clarify this in the manuscript since there is a statement in the introduction saying "lake evaporation throughout the year is not typically investigated".

- Fig.1; Add a scale, latitude/longitude to the lower right figure of the panel a.

- Fig.5 caption; "depth of 0 m"; should it be 0.4 m?

- Fig. 9; Change the solid line into a dotted line when there are missing values for an extended period.

- Fig. 10; what are the spikes in the evaporation values? They are weekly values so that they look strange. The authors should explain this in the main text (perhaps in connection with error estimates).

- Fig. 11; explain in the main text what the averaging period of the plotted data is.

- Table 1; add information on GMX600.

Below are my comments made to the authors' reply of the previous version. Since many of the points were not reflected in the current version, I cite them again here. _______________________________________________________________ A. 2. L76-84. The use of temperature and humidity measured at this location and by this instrument for the purpose 175 of calculating Bowen ratio (Bo) is questionable. ..... Reply: Fig.2 is replaced by the Figure below, which shows more detailed information about the installation of the instrument. We also address the location of the instrument in more detailed in the revision (line 94-96). We agree that instrument should be installed in a right place. Paiku Co is a deep lake and has steep shoreline. It is 190 difficult to install the instrument in the lake center. The logger was installed in an outcrop ∼2 m above the lake surface at the north part of the lake. The instrument is under a rock where there is a hole facing the lake. This site is very close to the lake and we believe that it is an ideal place to install the instrument. The meteorological condition over the lake surface can be well recorded

I am not convinced at all without supporting evidence that this is an "ideal place to install the instrument" and that the "meteorological condition over the lake surface can be well recorded". Since measurement location does not satisfy what the theory requires, it seems to me that the only way to convince readers is to show that their measurements are not very different from those on lake surfaces, and the minor difference does not propagate into the Bowen ratio estimation too much.

B - The sensor specification states the accuracy of _0.35_C for temperature and _2.5% for RH (from10% to 90%). 205 They are not particularly high. The accuracy of the water temperature sensor is _0.2_C. What would be the resulting accuracy of Bo and fluxes? The final possible error of the estimated fluxes would be due to (1) plus (2). Reply: It is true that the instrument we used in this study is designed for indoor use. We selected this instrument for measuring air temperature and humidity because it is cheap and easy to install. In fact, the instrument is installed just under a big stole where there is good ventilation, so the meteorological condition over the lake 210 surface can be well recorded. The accuracy of air temperature and humidity is also addressed in the revision (line 92-93). The accuracy of Bowen ratio is estimated in the revision (line 313-318). 'The accuracy of Bowen ratio depends on the accuracy of temperature and water vapor differences between lake surface and the overlying atmosphere. The HOBO instrument has an accuracy of 0.35 oC for air temperature and 2.5% of relative humidity. The HOBO water temperature sensor has an accuracy of 0.2 oC. Therefore, the accuracy of temperature difference between lake surface and the overlying atmosphere is estimated to be 0.4 oC ($=\sqrt{0.35^2+0.2^2}$). The water vapor difference between lake surface and the overlying atmosphere is averaged to be 0.57 kPa between June 2015 and May 2016. Therefore, the error of Bowen ratio is estimated to be 0.03, according to equation (4) in the main text.'

They are an estimation based on sensor specifications. What happened to the radiation effect on the measurements?

C 4. L103-106. Authors assumed Ts=Tw "because surface water can be mixed quickly by wind in the afternoon" and used Tw for their flux estimation. Please show the data to validate this statement. If no data are available, authors may want to add an argument that a small difference between Ts and Tw does not produce large 260 estimation errors of Bo and fluxes. In general, Ts is not equal to Tw even under windy conditions (see., e.g., Prats et al., Earth Syst. Sci Data, 10, 727-743, 2018). Reply: It is true that Ts is not equal to Tw. In this study, we do not measure the surface water temperature

and lake water temperature at 0.4~0.8 m is used to represent the surface water temperature. However, there is small difference between them and this difference does not produce large estimation error of Bo and heat fluxes.

How do you know that "this difference does not produce large estimation error of Bo and heat fluxes"? I would like to see the evidence.

D Reply: Thanks for pointing out this. The equation should be 'S=ΣðÌŚŘðÌŚď×ðÌlJŇðÌŚď×ΔðÌŚĽðÌŚŰ×ΔðÌŚĞðÌŚŰ72.8ðÌŚŰ=0ÌŘťðÌŚŹ'. Here ðÌŚŘðÌŚď is the specific heat of water (J kg-1 K-1), ðÌlJŇðÌŚď is water density (kg m-3), ΔðÌŚĽðÌŚŰ is the lake volume at certain depth, and ΔðÌŚĞðÌŚŰ is water temperature change at the same depth, ðÌŘťðÌŚŹ is lake area (m2). Changes in lake heat storage are calculated at an interval of 5 m and therefore there are 13 layers in vertical direction. Lake volume is acquired according to the 5 m isobaths. Lake water temperature at each layer is taken as the average value between the top and bottom lay. We do not estimate the accuracy of lake heat storage in this study because both the lake bathymetry and lake water temperature are all in-situ measurement and the error can be neglected (line 140-147).

I do not think "the error can be neglected" "because the lake bathymetry and lake water temperature are all in-situ measurements." If they are based on measurements, there are always errors in the measurements. Sensor errors, sampling errors, etc, etc...

E 7. L170-174. "water circulation"; this is an interesting point. Are there any supporting data for the presence of such circulation? Reply: We discuss this in more detailed in the revision (line 208-214). 'This contrasting pattern of water temperature at the bottom layer occurred during the late summer or early autumn when the vertical temperature gradient started to decrease. As shown in Fig.3, both the start and end of lake stratification were about half a month earlier in the southern basin relative to the northern basin. However, deeper water convection occurred earlier in the northern basin relative to the southern basin during this period (Fig. 3) due to relatively lower vertical temperature gradient in the northern basin. Lower temperature gradient caused deeper water convection in the northern basin compared with the southern basin during the late summer and early autumn

The authors do not directly reply to my question. Perhaps authors do reply to it indirectly, but and I do not understand the logic of the authors' reply.

F 13. L333-335. "In-situ observations of runoff at the three main rivers indicate that the surface runoff had weak impact on lake level changes.....(Table 3)"; Discharge values in Table 3 are only for short durations. Are those periods during baseflow? What would happen in case of rainfall-runoff events, or snow melting discharge? Reply: Runoff measurement was mainly conducted in late May or early October when the water level is already low. Besides runoff measurement in the three rivers, water level is also records by using HOBO water level loggers. We found that this discharge can approximately represent the average state in spring and autumn. Fieldwork in early April 2018 shows that there was almost no surface runoff between January and March.

The authors do not directly reply to my question. Perhaps the authors' reply is an indirect manner, but and I do not understand the logic of the authors' reply.

Specific comments: 1. Bowen ratio As I mentioned above, the authors' reply has not convinced me that their measurements can be used to estimate the Bowen ratio above the lake. Since it appears that they do not have evidence that their temperature and humidity measurements are equivalent to those over the lake, what I could suggest is to stop using the name of the "Bowen ratio method". Instead, they could introduce a new method (or a new name), a kind of empirical Bowen ratio method, or a relaxed version of the Bowen ratio method. In this method, a new variable Bo' is defined similarly to the Bowen ratio Bo but air temperature and humidity over the lake surface are replaced with that those over the land surface. Ideally, the validity of this empirical method should be studied first with an independent method. But alternatively, authors could do it indirectly through the comparison of evaporation with water level decrease as they have done in

their study. The only difference is whether they refer their method as the Bowen ratio method or not; I simply do not think it is appropriate to call it Bowen ratio based on their measurement. By the way, this type of empirical methods have been proposed and applied to lake evaporation estimation. For example, Harbeck (1962)'s empirical mass transfer formula for evaporation is E=N u (es-ea). N is the mass transfer coefficient, u is wind speed, es is at the water surface, but ea is over the land surface.

2. Wind speed According to the authors' reply L226-226, wind speed in the study area is not available. Yet, in the main text, there are repeated mentions of wind regime in the study area (e.g., L155, L158, L167, and L168)

- L117 "groundwater"; there is no discussion on groundwater and yet authors came to the conclusion here. - L269-271, "Negative value Bowen ratio indicates the lake water absorbs energy from the overlying atmosphere, and positive value indicates the lake water releases energy to the overlying atmosphere."; this is not quite true. Negative value only means the sin of H and LE are different. Depending on the relative absolute magnitude of H and LE, energy can either be absorbed or released by the lake. |For example, if H|>|LE| and H>0>LE, then H+LE>0 and the lake water does not absorb energy from the overlying atmosphere. - L283-284; "Sensible heat flux was negative between April and July with an average value of -5.6 WÅům-2 (Fig. 9b), which was mainly due to the negative temperature difference between surface lake water and the overlying atmosphere"; it is not "mainly". The negative temperature difference is the ONLY reason for negative H. - Fig.1 (C); what the difference between the squares and circles?

---

## Author Comment (AC2) · 6 Sep 2020

General comments

Reply: Thanks a lot for the constructive and detailed comments about our submission. They are important for further improving the manuscript. We will consider these comments and revise the manuscript carefully. A general reply to your comments has been made as the following:

About the uncertainty of lake evaporation

Reply: We will evaluate the uncertainty of lake evaporation at Paiku Co in the following

aspects, net radiation, lake meteorological data (air temperature and humidity) in the shoreline, surface water temperature, and changes in lake heat storage. Uncertainty of net radiation has been evaluated in the submission. Uncertainty of air temperature and humidity, lake surface temperature, changes in lake heat storage will be evaluated in the revision. The error of air temperature and humidity the in the shoreline will evaluated by comparing with those in the lake center. We set up a platform at the water depth of ∼19 m in southern Paiku Co in September 2019, and we only acquired one month's hydro-meteorological data from the lake center. Generally, both air temperature and humidity in the shoreline are very close and fluctuated similarly with those in the lake center during the observing period with correlation coefficient of 0.9 (R2=0.81), indicating that meteorological data in the shoreline can be used to calculate lake evaporation. RMS error of air temperature is estimated to 0.91 oC between center and shoreline during the observing period (23 September to 25 October 2019). RMS error of air temperature is estimated to 0.069 kPa between shoreline and center. For lake water temperature, we agree that there is some difference between lake skin temperature and body temperature. Because it is still difficult to acquire long-term and continuous lake skin temperature in the lake center in such a deep lake like Paiku Co, we have to use lake water temperature at the depth of 0.4-0.8 m. In order to check the difference of lake skin temperature and body temperature, nighttime and daytime MODIS (MYD11A2) lake surface temperature in 2016 and 2017 is retrieved and compared with in-situ measurement of lake surface temperature. MODIS lake surface temperature is usually considered to reflect instantaneous lake skin temperature. Result shows that that lake skin temperature is higher than lake body temperature during daytime, and lower than lake body temperature during nighttime. In spring and summer when the lake water gets warm, the skin temperature derived from MODIS data is about 1.2 oC higher than lake body temperature. In spring and winter when the lake water gets cool, the skin temperature derived from MODIS data is about only 0.05 oC higher than lake body temperature. Therefore, the difference of 0.6 oC between MODIS LST and in-situ observation data during a whole year is used to estimate the uncertainty of lake

evaporation in the revision. Uncertainty in lake heat storage is mainly determined by errors in lake water storage and lake water temperature profile. Uncertainty of lake water storage may result from measured water depths, interpolation algorithms, volume calculation methods, etc. The depth sounder measured the water depth at an accuracy of 1%, and the maximum water depth of Paiku Co is 70 m. Assuming the uncertainty of the water depth is 1 m, the uncertainty of lake volume at Paiku Co is estimated to be 2.5%, according to the average water depth of $\sim$40m. Using method of Qiao et al. (2018), uncertainty of lake water storage estimation at Paiku Co is estimated to 6% by comparing the reconstructed lake level and ICESat and CryoSat-2 satellite altimetry data between 2003 and 2018. Lake water temperature profile was measured by HOBO logger at an interval of 5-10 m with an accuracy of $\pm$0.2 oC. The spatial difference of lake water temperature is investigated by comparing water temperature profile at the northern and southern centers of Paiku Co. The error of lake evaporation caused by lake water temperature difference on the northern and southern centers of Paiku Co is estimated to 35 mm.

Introduction. - The originality of the study:

Reply: The originality of our study mainly includes: 1) Lake evaporation at Paiku Co during the whole ice free period is investigated. In many previous studies, lake evaporation during the late autumn and early winter is not well studied because it is difficult to install and maintain measurement platform due to the harsh natural conditions and the influence of lake ice; 2) Changes in lake heat storage at Paiku Co are quantified and its impact on seasonal changes in lake evaporation is addressed. Changes in lake heat storage are not well studied for most lakes with eddy covariance method on the Tibetan Plateau, so its impact on seasonal changes in lake evaporation is not clearly addressed; 3) How lake evaporation affects seasonal lake level changes is still unclear. As introduced in the text, there is different magnitude of lake level seasonality, but the main causes for this remains unclear due to lack of comprehensive observation of lake water budget.

- L96-97. ": : :therefore the meteorological condition over the lake surface can be recorded". This cannot be true without evidence. I made the same argument in previous reviews so please read it again. What I would suggest is to acknowledge that it is not the location where measurements should be made, but the measurements were used as a proxy of the above-lake measurements, and validity of this proxy will be discussed in section: : :(see major comment)

Reply: Thanks for the suggestions. We will address this in the revision according to your suggestion. We agree that it is not the location where measurements should be made and the measurement was used as a proxy of above lake measurements in the revision. Uncertainty of air temperature and humidity has been evaluated (please see reply above)

- L106 "weekly averaged radiation..."; What are the possible errors to apply the Bowen ratio method with weekly averaged data? The Bowen ratio equation (4) was derived from two profile equations of H and LE. Profiles equations derived by applying the similarity theory are valid for the steady-state condition under certain stability. So we generally apply them for 30-min to hourly mean values. For practical purposes, we apply them for daily data assuming neutral stability but strictly speaking, this is not valid since profile equations are not linear and therefore simple time averaging does not yield valid equations for the given averaging period.

Reply: In this study, daily Bowen ratio is calculated. Weekly averaged radiation was used to calculate lake evaporation in order to reduce the error caused by regional difference. Weekly evaporation is calculated according to the weekly Bowen ratio.

- L114-115. "the influence of river discharge: : :can be neglected" You cannot say this without supporting evidence. The authors should give and compare lake storage and river discharge.

Reply: The influence of river discharge on lake energy budget will be discussed in the revision.

- L114-115. ": : :therefore: : :we do not consider : : :G.." There is no mention of the reason why G can be neglected.

Reply: We do not considered G in this study because there is no data available about G.

- L153-155, ": : :reduction in wind speed (data not shown)"; As I recall from the reply of authors to the reviewers' comments in the previous version, authors do not have wind speed data. Then the statement of "data not shown" is misleading. When we see this statement, we tend to believe that there were data and authors checked them to validate what is written in the manuscript even though they are not shown in the manuscript with a figure or a table. If you do not have data, then you should not mention wind influence as if it was based on data. There are similar statements on wind speed here and there in the manuscript. Authors need to remove them or change expressions. Alternatively, authors could rely on wind speed from reanalysis data. However, the reliability of any reanalysis data set should be established first (perhaps by referring to previous studies) for the study area before they can use the reanalysis data.

Reply: Here wind speed at Qomolangma station is used to compare with lake surface temperature changes. Qomolangma station is located at the northern slope of Mount Everest, about 150 km east of Paiku Co. If this data is just used as a simple comparison, this is no problem because it does not need high accuracy. But if wind speed at Qomolangma station is used to calculate lake evaporation, maybe it has too large spatial difference.

- L179 "..but also the bottom water"; I do not understand what authors want to claim.

Reply: What we want to address is that lake heat storage in summer is mainly determined by water temperature changes in the upper layer because the lake water is stratified, but in other seasons, the lake is fully mixed, lake heat storage is determined not only by the upper water, but also the bottom layer.

- L200-201; "....Lower temperature gradient caused stronger water convection....."; I do not understand the logic in this part. I assume water convection is stronger when the vertical gradient is larger.

Reply: As far as my knowledge, water convection is weaker when the vertical temperature gradient is larger because of its high stability. For example, water convection is weaker in August when the temperature vertical gradient is large, because the lake water is stratified. Similarly, water convection is strong in winter when the temperature vertical gradient is weak because the lake water is mixed.

- L204-205 "(Fig.1)"; Fig. 1 shows the locations of water level loggers but authors are talking about water temperature. In Fig.1, there are also the locations of water temperature measurements. This is confusing.

Reply: Please note that HOBO water level logger can not only record water level changes, but also water temperature changes.

- L208-209 "....large errors can result if only water temperature data collected at the shoreline are used to calculate lake heat storage and energy budget."; Similarly, errors can result if only water temperature data collected at the center of the lake are used. The authors should acknowledge this possibility to make analysis accordingly (see major comment).

Reply: Generally, lake water temperature at the lake center can stand for the average thermal condition, but lake water temperature at the shoreline is more regional.

- L274-276 " Lake evaporation between middle January and April is not determined... "; Why not? The authors do give latent heat flux for this period. If it is not certain whether the lake surface is covered with solid water or liquid water, then authors could give two values of evaporation. One in the case of the ice surface, and another one in the case of liquid water surface. The true evaporation is somewhere in between. This can be used together with the evaporation estimate obtained by assuming it is the same as

water level change in L290-291.

Reply: We will address this paragraph in more detail. We do not determine lake evaporation between middle January and April because the lake surface is sometimes covered by ice. Energy budget over the lake surface during ice covered season is different from that without lake ice. So we gave a rough estimation according to lake level change during this period when there is almost no river discharge and little precipitation.

- L288 ".....lake ice can effectively prohibit evaporation."; Is this true? How about sublimation? Is the latent heat flux on ice-covered lake zero? The authors could add references to support their statement.

Reply: We agree that lake water can still get lost through sublimation. We will mention it in the revision.

- L290-291 "Assuming lake evaporation between January and April is equal to lake level decrease ..."; the Authors should provide an error estimate of evaporation based on this assumption. Errors due to lake level measurements, mean lake water level estimation, water level-volume relation, water level-lake surface area relation, etc.

Reply: Thanks for the suggestion. We will give uncertainty of lake level variations in winter season between 2013 and 2019.

- L326-327 " Assuming approximately 70% of the net radiation was consumed by lake evaporation (Lazhu et al., 2016)........ 74.5 mm per year "; This percentage is from a different lake. Why not use estimates for Paiku Co. given in Table 2? The authors should explain how 74.5 mm/year was derived.

Reply: Thanks for the good suggestion. We will calculate the uncertainty according to estimation at Paiku Co in this study.

- L345- " Uncertainty of lake evaporation in this study was also validated by comparing lake level changes"; Just like the case in L290-291, authors should provide error

estimates for the evaporation estimates based on lake level measurements. Possible advection due to precipitation should be addressed as commented above for L223.

Reply: Thanks for the good suggestion. We will give uncertainty of lake evaporation in winter season. Error of lake level changes will be estimated and taken as uncertainty of lake evaporation in winter season.

- L352 "As shown in Table 3, runoff at the three large rivers can contribute to lake level increase by 0.71.6"; Runoff values in Table 3 are for a short period. Can you use them to estimate monthly runoff?

Reply: Monthly lake runoff at Paiku Co can be estimated based on this measurement, but with large uncertainty. Therefore, we just use daily runoff in this paper.

- L355 "To further explore the impact of lake heat storage on the seasonal pattern of lake evaporation...."; Authors should summarize at the end of section 4.3, what kind of new findings were obtained on the impact of heat storage to lake evaporation from their measurements/analyses and comparison with previous studies. The phase shift of lake evaporation due to lake heat storage is in a way common knowledge. We would like to hear something new here. How about differences among the TP lakes? For example, in the introduction, the authors mention the difference of lake size change between the interior TP and southern TP. Any new findings on this point? Also try to make clear the relation between the statements made in the introduction (e.g., L58-67) and those in this section. You do not have to say similar things in different parts of the manuscript.

Reply: Changes in lake heat storage at Paiku Co are quantified and its impact on seasonal changes in lake evaporation is addressed in this study. Changes in lake heat storage are not well studied for most lakes with eddy covariance method on the Tibetan Plateau, so its impact on seasonal changes in lake evaporation is not clearly addressed. We will further summarize the new findings in the revision.

- Section 4.3; In addition to comparison with other lakes in TP, authors may want to address the difference of the TP lakes in comparison with other lakes in the world. What are special about the TP lakes? Are there similar lakes in other parts of the world? What are the controlling factors to make them similar/different?

Reply: Beside lakes on the TP, we will check the water level changes of endorheic lakes in other regions of the world, and further compare them with lakes on the TP.

- L359 "2011-2012", L362 "2013-2015"; Evaporation estimates for these periods are continuous even during the winter season? Clarify this in the manuscript since there is a statement in the introduction saying "lake evaporation throughout the year is not typically investigated".

Reply: We will clarify this in the revision. As we have addressed in the introduction, lake evaporation derived from eddy covariance during winter season is not investigated at most of the previous studies. Lake water temperature profile and changes in lake heat storage are also not investigated. An except is Qinghai Lake, where water temperature at only the upper 3 m was investigated, but the whole water temperature profile is still unclear.

Below are my comments made to the authors' reply of the previous version. Since many of the points were not reflected in the current version, I cite them again here.

A. About the representativeness of meteorological data

Reply: We have added the meteorological data in the lake center and further compared these data with those in the shoreline. We agree that hydro-meteorological data at the shoreline does not completely satisfy the standard of Bowen ratio calculation, but at present it is difficult to install platform in the center in such a deep lake. We set up a platform at the water depth of ~19 m in southern Paiku Co in September 2019, and had attempted to get real hydro-meteorological data in the lake center. Unfortunately the platform was destroyed by lake ice and AWS station was lost in winter 2019. We

only acquired one month's hydro-meteorological data from the lake surface. After comparison, we found air temperature and humidity in the lake center varied similarly with those at the shoreline.

B - They are an estimation based on sensor specifications. What happened to the radiation effect on the measurements?

Reply: The instrument was installed just under a big stole where there is good ventilation. Radiation can not affect the equipment.

C 4. L103-106. About lake surface water temperature. Please show the data to validate this statement.

Reply: Please see the reply to the major comments. In situ measurement has validated by MODIS lake surface temperature data.

F 13. L333-335. About In-situ observations of runoff at the three main rivers. Perhaps the authors' reply is an indirect manner, but and I do not understand the logic of the authors' reply.

Reply: I think our reply is already very clear. 'Besides runoff measurement in the three rivers, water level is also records by using HOBO water level loggers. We found that this discharge can approximately represent the average state in spring and autumn. Fieldwork in early April 2018 shows that there was almost no surface runoff between January and March.'

Specific comments: 1. Bowen ratio As I mentioned above, the authors' reply has not convinced me that their measurements can be used to estimate the Bowen ratio above the lake. Since it appears that they do not have evidence that their temperature and humidity measurements are equivalent to those over the lake, what I could suggest is to stop using the name of the "Bowen ratio method". Instead, they could introduce a new method (or a new name), a kind of empirical Bowen ratio method, or a relaxed version of the Bowen ratio method. In this method, a new variable Bo' is defined similarly to the

Bowen ratio Bo but air temperature and humidity over the lake surface are replaced with that those over the land surface. Ideally, the validity of this empirical method should be studied first with an independent method. But alternatively, authors could do it indirectly through the comparison of evaporation with water level decrease as they have done in their study. The only difference is whether they refer their method as the Bowen ratio method or not; I simply do not think it is appropriate to call it Bowen ratio based on their measurement. By the way, this type of empirical methods have been proposed and applied to lake evaporation estimation. For example, Harbeck (1962)'s empirical mass transfer formula for evaporation is E=N u (es-ea). N is the mass transfer coefficient, u is wind speed, es is at the water surface, but ea is over the land surface.

Reply: About the hydro-meteorological data used in this study, we agree that it does not completely satisfy the standard of Bowen ratio calculation. We will evaluate the uncertainty of all the data used in this study and found the data we used in this study is generally suitable to study lake energy budget, evaporation and its impact on seasonal lake level changes.

Besides lake evaporation, our study investigates the thermal regime and energy budget of Paiku Co. We give a detailed description of changes in lake heat storage through in situ measurement of water temperature profile and energy budget at Paiku Co is investigated accordingly. As far as my knowledge, this is the first time to give such a detailed investigation of energy budget on the TP. Although the reviewer does not agree that energy budget is not a direct method to investigate lake evaporation, eddy covariance method can exactly not know the changes in lake heat storage of the whole lake and its impact on the seasonal pattern of lake evaporation.

---

## Author Comment (AC4) · 14 Sep 2020

(1) In line 25-26ïijNĔĞ the last sentence in abstract seems has not clear connection with the other contents, I suggest to revise the sentence to keep it coherent with previous contents.

Reply: Thanks for the suggestion. We will revise this sentence to keep it coherent with the main purpose of the paper.

(2) In line 120-125, in equation (2), Ra is the downward longwave radiation to lake, while in equation (3) Ra is rewritten as the longwave radiation from lake. Here, in

equation (3) and line 125, I think it should be Rw.

Reply: Thanks for pointing out this error. We will change Ra to Rw in Equation (3).

(3) In line 129-130, as daily averaged water temperature is used, in addition to the surface mixing by wind and convection, Here I suggest to add information that "there exists surface warming during the day and surface cooling at night for high elevation lakes, thus the two uncertainties by surface warming and cooling can cancel each other at a temporal resolution of daily."

Reply: Thanks for the good suggestion. We will add this information in the revision.

(4) In line 161, I suggest to use "period" instead of "time" here; in line 262, it should be "in low values" rather than "in low value"; Figure 3 caption, "at different depths" rather than "at different depth"; Figure 4, a unit of (OC) should be added for the color bar.

Reply: Thanks for pointing out these errors. We will revise them in the revision.

---

## Author Response (AR1)

**Response to reviewer #1**

*This paper reported the seasonal changes of lake water profile, lake levels, surface heat budget, and evaporations by three years in-situ observation data. They showed very interesting characteristics representing lake environment in southern edge of the TP, that gives us the hints to understand basic processes of heat/water budget of mountain lakes under Indian monsoon climate. As authors introduced in the introduction, lakes on the TP are changing. It is very important to reveal that how the global environment change could modify the lake environment through land-atmosphere interaction. The contents showed basic timelines of observed data with estimated heat budget and evaporation amount, and natures could be easily captured by figures.*

**Response:** We are very grateful to these comments and we believe that these comments are very helpful to improve the paper. We have revised the paper according to these comments carefully. The main responses to these comments are shown in the following:

*However, many key mechanisms are discussed by speculations without in-depth examination/comparisons to previous studies in the TP. This is because the study did not set clear objectives. Therefore, the title is also uncoordinated. For instance, do authors concern about the lake area (level) changes of Paiku Co? Figure 10 shows that lake level show small seasonal variation (within 1m), but do you think this is critical? Or, authors investigated large evaporation rate instead of previous studies? Readers can not understand how the Fig. 10 differs from other lake or even from ground in the TP. If the HESS request level of paper as scientific article instead of "report", I would like to suggest that paper needs fundamental revisions with clear objectives and results based on additional in-depth analysis.*

**Response:** Substantial revisions, including both structure of the paper and Figures, have been made according to your comments in order to make the paper look more like a scientific paper instead of 'report'. The main changes include 1): remove sections 3.2 and 3.3 in the previous version; 2) move section 4.4 in the previous version to section 3.4 in the revision; 3) uncertainty estimation of lake evaporation according to hydrometeorology in the lake center; 4) combine Fig. 6-9 into Fig. 4 and Fig. 5 in the revision; 5) add precipitation and runoff in Fig. 6; 6) add a new Fig. 7 in the revision to show different amplitude of lake level changes.

We also try to address the main objective of this study more clearly in the revision (Line 60-65), that is to quantify lake evaporation during the entire ice free period using energy budget method and investigate its effects on seasonal lake level changes. Until now, lake evaporation during the late autumn and early winter is not

typically investigated on the TP because it is difficult to install and maintain measurement platform due to the harsh natural conditions and the influence of lake ice. As a result, how lake evaporation affects seasonal lake level changes remains unclear due to lack of comprehensive observation of lake water budget.

We have added one paragraph in the introduction about the different amplitude of endorheic lake level seasonality on the TP (Line 42-51). 'Compared with numerous studies of inter-annual to decadal lake changes, seasonal lake level changes and the associated hydrological processes on the Tibetan Plateau (TP) are still less understood. Phan et al. (2012) showed that seasonal lake level variations in the southern TP are much larger than that in the northern and western TP. In-situ observations gave more details of seasonal lake level variations (Lei et al., 2017). One striking feature is the different amplitude of seasonal water level variations, that is, deep lakes usually exhibited considerably greater lake level variations than shallow lakes. For example, Zhari Namco and Nam Co, two large and deep lakes on the central TP (Wang et al., 2009, 2010), exhibited significant water level increase by 0.3~0.6 m during the summer monsoon season and a similar magnitude of lake level reduction by 0.3~0.5 m during post-monsoon season between 2010 and 2014. For the two nearby small and shallow lakes, Dawa Co and Bam Co, although there was a similar pattern of lake seasonality, the amplitude of seasonal lake level variations was considerably smaller than the two large and deep lakes (Lei et al., 2017).' The main causes for the different amplitude of lake level changes are still not investigated in previous studies, which is the main topic of our study.

We change the title of the paper to: 'Contrasting hydrological and thermal intensities determine seasonal lake level variations–A case study at Paiku Co in the central Himalayas'

Seasonal variations of lake level and lake evaporation are shown in a new figure (Fig. 6) to show the contrasting hydrological and thermal intensities of Paiku Co.

Previous studies about lake evaporation on the TP have been reviewed and compared in Section 4.2 in the revision.

*For the lake dynamics by means of hydrometeorology, following points need to be examined. 1) Water temperature profiles were almost homogeneity during Oct.- June (non-monsoon season), and author explained by "fully mixing" without any analysis. Please proof it physically using surface wind speed and variability conditions and water mixing theory. It is curious that such mixing occurred suddenly. In the central TP, large diurnal wind changes are found in winter due to the coupling of upper strong STJ and boundary layer development. Any relation to the seasonal change of atmospheric circulation?*

**Response:** Lake water temperature profile can be taken as a proxy of lake water mixing. In summer, the lake water is stratified due to the dramatic temperature gradient between surface and bottom. In autumn, the lake stratification is weakened gradually due to the decreased temperature gradient and intensified wind speed. From early October to later October, lake mixing was deepened gradually due to decreased temperature gradient and increase wind speed. Since the late October, the lake water is totally mixed as indicated by the disappearance of vertical temperature gradient. As we have addressed in the main text, the lake mixing is mainly forced by wind disturbance and water convection. This is also the common feature of lake water circulation, which has been addressed in many publications and books (i.e. Wetzel, 2001).

Clear Lake stratification at Paiku Co occurred in late June or early July, which corresponded to a significant reduction in wind speed. The lake stratification broke down in late October, which corresponded well to significantly increased wind speed. So lake water mixing may be related to seasonal changes in atmospheric circulation.

[Figure]

Comparison of lake water temperature and average wind speed. Daily wind speed at Qomolangma station is used.

*2) Seasonal change of water level should be explained by seasonal change of water budget, including precipitation, river runoff/inflow and surface water inflow. Even there are lack of areal in-situ measurements, some parameters could be estimated by previous studies or literature. This also links to Av calculation as mentioned in 3). I could not see precipitation*

*records, but the Rn sequence demonstrated that rain season is not clear compare to southern Himalayas and central plateau.*

*If the impact of monsoon is small with fair/non-freeze weather, location of the lake may represent local dry climate behind the Himalayas where lee-side subsidence prevails, and that would characterize evaporation rate at Paiku Co.*

**Response:** We agree that seasonal lake level change should be explained by lake water budget, including precipitation, runoff etc. We have added a new Fig. 6 in the revision. Lake level, precipitation and runoff data are included (Fig. 6). Notably, we do not focus on all components of lake water budget in this study. The main purpose of this study is to quantify lake evaporation and its impact on seasonal lake level changes.

We agree that the impact of monsoon precipitation is small in this dry area and location of lake represent local dry climate behind the Himalayas where subsidence prevails. We have added this point in the site description in the revision (Line 80-83).

*3) To consider the heat budget of the lake, especially for the condition of thermocline, advection of cold (snow/glacier-melt) water associated with river/surface inflow need to be considered. This paper only compares the heat budget at water surface, and conclude the evaporation as a ley parameter to affect lake level seasonality. Is there no effects of glacier melt water (they are illustrated in Fig. 1a) or monsoon precipitation inflow to establish lake temperature profile and lake level seasonality? Diurnal change of river level according to the glacier melt is observed by previous studies. There are some indication at the bottom temperature of northern point in Fig. 3b. At L115, please proof that Av can be ignored. Authors should not avoid those issues to analyze if they focus on the water cycle and environmental changes on the TP as introduced in Chapter. 1.*

**Response:** We have added one paragraph in Section 2.3 to evaluate the impact of Av on the lake energy budget (Line117-123). Av can be estimated according to total river discharge (Fig. 1) and the water temperature difference between river and lake (data not shown). Lake water temperature was almost same to river water temperature between April and June, but 2-4$^{o}$C higher between July and December. As a deep lake, total river discharge to Paiku Co was about 800-900 mm water equivalent to lake level and accounted for 2-2.5% of total lake water storage. The river discharge can accumulatively decrease lake water temperature by ~0.1 $^{o}$C in a year. Therefore, as a deep lake, the influence of river discharge and precipitation on the total lake heat storage at Paiku Co is very small and can be neglected. Therefore, we do not consider the influence of Av on the lake energy budget in this study.

[Figure]

Comparison of water temperature between Bulaqu river (red line) and Paik Co (blue)

110 We do not deny the contribution of precipitation and runoff in summer, there are very important for the rapid lake level increase in summer monsoon season. What we want to address in this study is the effect of high evaporation in autumn and early winter. Although solar radiation is already low in post monsoon season, high lake evaporation can lead to rapid lake level decrease due to the low runoff and precipitation during this period. Contribution of runoff to lake level increase in spring and summer is estimated in Table 3.

115

*Minor comments:*

*There are no previous studies in Paiku Co. ? Need reviews.*

**Response:** We give a review about previous studies at Paiku Co (Line 73-83). We did not do this because we have reviewed it in a previous publication (Lei et al., 2018).

120 Lei, Y., Yao, T., Yang, K., et al.: An integrated investigation of lake storage and water level changes in the Paiku Co basin, central Himalayas, J. Hydrol., 562, 599–608, 2018.

*Water temperature sensors in the upper profile are shaded? Or, how deep the insolation can penetrate the water at the target lake?*

125 **Response:** We do not think water temperature sensors in the upper profile are shaded. As we have illustrated in section 2.2, the first water temperature sensor were fixed below buoy at the water depth of ~0.5 m. The other sensors were fixed on the rope which was tied to an anchor at the lake bottom. The transparency of the lake is not measured until now.

130 *L176Aâ Small water temperature gradient is explained by cold air temperature. This is strange. Air temperature change is due to latent heat from the surface or advection. Enough radiation could increase the water temperature even the air temperature is cold with weak winds.*

**Response:** We attribute it the relatively lower lake surface temperature in the revision.

135 *L175-180 Those are speculations, not results.*

**Response:** We have deleted this part in the revision.

*L138 "input data were averaged at weekly interval. Does heat budget screened by the wind direction by instantaneous data then averaged?*

140 **Response:** Wind speed and direction is not used during the calculation of energy budget and lake evaporation.

*Units in Fig. 10 are mm/d, cm and it is m3/s in Table 3. Please unify them to capture accurate water balance.*

**Response:** Thanks for pointing out this. We have unified the units (mm) in Fig. 6 in the revision.

145 *Title of 3.3 "Lake hydrometeorology" is vague.*

**Response:** We have merged it to section 3.2.

*L215ã˘Aâ "There was a 1.5 month lag between lake surface temperature and air temperature." Is not clear.*

**Response:** We have deleted this part in the revision.

150

*I could not understand the meaning to show the Fig.6.*

**Response:** We have combined Fig. 6-9 into Fig. 4-5 in the revision.

*L230 "Downward shortwave radiation at Paiku Co had an annual average of 251.8W¡um-2 (Fig. 7), which is slightly*
155 *higher than the TP average due to its lower latitude (Yang et al., 2009)." What is the TP average? Effects of Indian monsoon is stronger in southern TP in general, and cloudy weather may reduce the insolation. Or, the observation represent local weather in the valley?*

**Response:** According to Yang et al (2009), downward shortwave radiation on the TP varies in a range of $150\sim290W/m^2$, which is slightly lower than that at Paiku Co ($190\sim290W/m^2$). Although effect of Indian monsoon
160 is stronger in the southern TP in general, mean annual precipitation at Paiku Co is only about 150-200 mm. So we

agree that the observation represents the local dry climate behind the Himalayas where subsidence prevails (Line 82-83).

*L230-237 Discussions are not clear due to mixture of seasonal change and annual average. Why the rainy season is not clear?*

**Response:** As we have shown in Fig. 6, the rainy season is not clear because the precipitation at Paiku Co is low.

*L17, L248,, "a deep lake". Many discussion attribute the characteristics to the depth of lake without examination. Manly lakes over the TP are shallower than the target lake? Please review that how the depth of lake over the TP characterize the lake temperature condition.*

**Response:** As we have shown in section 4.2, seasonal changes in lake evaporation are significantly related to the mean lake water depth and the lake heat storage. The mean water depths of the lakes have been mentioned in section 4.2. Deep lake can store more energy in spring and summer and release more energy to the overlying atmosphere, which affects the season pattern of lake evaporation.

195

200

**Response to reviewer #2**

*The manuscript "Thermal regime, energy budget and lake evaporation at Paiku Co, a deep alpine lake in the central Himalays" use in-situ measurements to analyze the energy budget components and obtain the evaporation amounts of Lake Paiku Co. As lake measurements are very limited on the Tibetan Plateau and most of the measurements are in the central or east parts of the Tibetan Plateau, the manuscript shows significance in describing clearly the thermal regime, energy budget and lake evaporation of a western lake on the Tibetan Plateau by in situ measurements. The structure of the manuscript is well-organized; the analysis of the processes are observation based and reasonable; I consider the manuscript to be appropriate to be published in the HESS journal after a minor revision. The detailed comments are given as belows:*

Reply: Thanks for the positive comments. We have revised the manuscript according to your comments.

215

*(1) In line 25-26ïj̈Nˇ the last sentence in abstract seems has not clear connection with the other contents, I suggest to revise the sentence to keep it coherent with previous contents.*

Reply: Thanks for the suggestion. We have revised this sentence to keep it coherent with the main purpose of the paper.

220  *(2) In line 120-125, in equation (2), Ra is the downward longwave radiation to lake, while in equation (3) Ra is rewritten as the longwave radiation from lake. Here, in equation (3) and line 125, I think it should be Rw.*

Reply: Thanks for pointing out this error. We will change $R_a$ to $R_w$ in Equation (3).

*(3) In line 129-130, as daily averaged water temperature is used, in addition to the surface mixing by wind and convection,*
225  *Here I suggest to add information that "there exists surface warming during the day and surface cooling at night for high elevation lakes, thus the two uncertainties by surface warming and cooling can cancel each other at a temporal resolution of daily."*

Reply: Thanks for the good suggestion. We have added this information in the revision (Line 134-136).

230 *(4) In line 161, I suggest to use "period" instead of "time" here; in line 262, it should be "in low values" rather than "in low value"; Figure 3 caption, "at different depths" rather than "at different depth"; Figure 4, a unit of (OC) should be added for the color bar.*

Reply: Thanks for pointing out these errors. We have revised them in the revision.

235

240

**Response to reviewer #3**

*Major comments:*

*Since the Tibetan Plateau has been one of the least studied areas, lake evaporation study in this region is welcomed and*
245 *worth publication in HESS. However, I find there are substantial problems in this study. That is the accuracy of the evaporation in their study. I pointed out this issue in the review at the time of their previous submission. Unfortunately, they failed to solve this problem and the manuscript was rejected for publication. The authors have added new data at the lake center to compare their measurements on the shoreline. This is good. On the other hand, their treatment of the comparison is not enough to convince readers that their evaporation estimates are accurate and reliable.*

250 Reply: Thanks a lot for your constructive and detailed comments about our submission. We have evaluated the uncertainty of lake evaporation at Paiku Co from the following aspects, net radiation, lake meteorological data (air temperature and humidity) in the shoreline, surface water temperature, and changes in lake heat storage.

*The authors gave error estimates of evaporation in Section 4.2. They selected (1) net radiation and (2) water temperature*
255 *differences between the northern and southern basins as the error sources. I think they should add other relevant error sources. They include, first, the use of air temperature and humidity on the shoreline instead of those above lake water. They compare the measurements on the shore and on the lake in Fig. 11 and conclude they are "very similar: : :.. data from the shoreline: : : can be used to represent the general condition of the whole lake: : :". This is new information and should be used to evaluate error from using onshore measurements. What they should do is to determine the RMS error of temperature*
260 *and humidity measurements on the shore and used them to estimate errors of Bowen ration and sensible and latent heat fluxes. This can be added to the final error evaluation.*

Reply: Thanks for the suggestion. We evaluate the error of air temperature and humidity the in the shoreline by comparing with those in the lake center. Generally, both air temperature and humidity in the shoreline are very close and fluctuated

similarly with those in the lake center during the observing period, indicating that meteorological data in the shoreline can be
used to calculate lake evaporation. RMS error of air temperature is estimated to 0.91 $^o$C between center and shoreline during
the observing period (23 September to 25 October 2019). RMS error of water vapor pressure is estimated to 0.069 kPa
between shoreline and center.

*The second error source they should consider is the use of water temperature instead of surface temperature. They claim*
*that "the daily average between them is very similar: : :". But no supporting evidence is shown. In fact, previous studies do*
*indicate a difference between the two even for mean values for day or longer. The authors should accept this and add this as*
*one of the error sources for the evaporation estimates.*

Reply: We agree that there is some difference between lake skin temperature and body temperature. Because it is still
difficult to acquire long-term and continuous lake skin temperature in the lake center in such a deep lake like Paiku Co, we
have to use lake water temperature at the depth of 0.4-0.8 m. To check the difference of lake skin temperature and body
temperature, nighttime and daytime MODIS (MYD11A2) lake surface temperature in 2016 and 2017 is retrieved and
compared with in-situ measurement of lake surface temperature. Generally, MODIS lake surface temperature is considered
to reflect instantaneous lake skin temperature. Result shows that that lake skin temperature is higher than lake body
temperature during daytime, and lower than lake body temperature during nighttime. The daily average water temperature
used in this study is close to lake skin temperature because the lake water can be well mixed by the strong wind in the
afternoon at Paiku Co. In spring and summer when the lake water gets warm, the skin temperature derived from MODIS data
is about 1.2 $^o$C higher than lake body temperature. In spring and winter when the lake water gets cool, the skin temperature
derived from MODIS data is about 0.05 $^o$C higher than lake body temperature. Therefore, the difference of 0.6 $^o$C between
MODIS LST and in-situ observation data during a whole year is used to estimate the uncertainty of lake evaporation in the
revision.

*The third error source is the error in the energy storage estimation. To estimate energy storage, spatial mean water*
*temperature profile, spatial mean water level, water level-volume relation, water level-surface area relation are needed.*
*Since they are all based on some kind of measurement, there are always errors (measurement errors as well as sampling*
*errors). They should be considered.*

Reply: Thanks for the suggestions. Uncertainty in lake heat storage is mainly determined by errors in lake water storage and
lake water temperature profile. Uncertainty of lake heat storage can significantly affect the seasonal distribution of lake
evaporation. Uncertainty of lake water storage may result from measured water depths, interpolation algorithms, volume
calculation methods, etc. The depth sounder measured the water depth at an accuracy of 1%, and the maximum water depth
of Paiku Co is 70 m. Assuming the uncertainty of the water depth is 1 m, the uncertainty of lake volume at Paiku Co is
estimated to be 2.5%, according to the average water depth of ~40m. Using method of Qiao et al. (2018), uncertainty of lake

water storage estimation at Paiku Co is estimated to 6% by comparing the reconstructed lake level and ICESat and CryoSat-2 satellite altimetry data between 2003 and 2018.

300 Qiao, B., Zhu, L., Wang, J., Ju, J., Ma, Q., Huang, L., Chen, H,, Liu, C., Xu, T.: Estimation of lake water storage and changes based on bathymetric data and altimetry data and the association with climate change in the central Tibetan Plateau. Journal of Hydrology, 578, 124052, 2019.

Minor comments:

305 *Introduction. - The originality of the study: It is not clear what the original contribution of this study is. Authors claim that previous studies do not provide evaporation throughout a year. But in their study also, evaporation was not determined during the winter period. So it is not quite new. Please make it clear what is missing in previous studies and why their studies are needed based on a comprehensive review of previous studies. Also, these points should be reflected in discussion and conclusions*

310 Reply: Thanks for the suggestion. The originality of our study mainly includes: 1) Lake evaporation at Paiku Co during the whole ice free period is investigated. In many previous studies, lake evaporation during the late autumn and early winter is not well studied because it is difficult to install and maintain measurement platform due to the harsh natural conditions and the influence of lake ice; 2) Changes in lake heat storage at Paiku Co are quantified and its impact on seasonal changes in lake evaporation is addressed. Changes in lake heat storage are not well studied for most lakes with eddy covariance method

315 on the Tibetan Plateau, so its impact on seasonal changes in lake evaporation is not clearly addressed; 3) How lake evaporation affects seasonal lake level changes is investigated in this study. As introduced in the text, there is different magnitude of lake level seasonality, but the main causes for this remains unclear due to lack of comprehensive observation of lake water budget.

320 *Importance of TP lakes. The authors explain the abundance of lakes in TP. Then authors should add relevance of these lakes for TP (or even for larger areas).*
Reply: The importance of TP lakes has been addressed in the revision (Line 34-41).

*Eddy correlation method. The authors mention that it is not suitable for long-term measurements. I believe this statement*
325 *was correct perhaps 20 years ago. But it is easy to see the results of long-term measurements based on the eddy correlation method in the literature as well as datasets on the flux net sites.*
Reply: We have deleted this paragraph in the revision.

*Direct measurements of lake evaporation: I do not think the Bowen ratio method is in the category of direct measurements.*
330   *It measures energy balance and evaporation is obtained only indirectly as one of the residuals of the energy balance equation. It relies on the similarity between temperature and humidity profiles.*

Reply: We have deleted this paragraph in the revision.

*- L96-97. ": : :therefore the meteorological condition over the lake surface can be recorded". This cannot be true without*
335   *evidence. I made the same argument in previous reviews so please read it again. What I would suggest is to acknowledge that it is not the location where measurements should be made, but the measurements were used as a proxy of the above-lake measurements, and validity of this proxy will be discussed in section: : :(see major comment)*

Reply: Thanks for the suggestions. We have addressed this in the revision according to your suggestion. We agree that it is not the location where measurements should be made and the measurement was used as a proxy of above lake measurements
340   in the revision (Line 99-101).

*- L106 "weekly averaged radiation..."; What are the possible errors to apply the Bowen ratio method with weekly averaged data? The Bowen ratio equation (4) was derived from two profile equations of H and LE. Profiles equations derived by applying the similarity theory are valid for the steady-state condition under certain stability. So we generally apply them for*
345   *30-min to hourly mean values. For practical purposes, we apply them for daily data assuming neutral stability but strictly speaking, this is not valid since profile equations are not linear and therefore simple time averaging does not yield valid equations for the given averaging period.*

Reply: In this study, daily Bowen ratio is calculated. Weekly averaged radiation was used to calculate lake evaporation in order to reduce the error caused by regional difference. Weekly evaporation is calculated according to the weekly Bowen
350   ratio.

*- L114-115. "the influence of river discharge: : :can be neglected" You cannot say this without supporting evidence. The authors should give and compare lake storage and river discharge.*

Reply: The influence of river discharge on lake energy budget has been discussed in the revision (Line 118-124). Av can be
355   estimated according to total river discharge (Fig. 1) and the water temperature difference between river and lake (data not shown). Lake water temperature was almost same to river water temperature between April and June, but 2-4$^{\circ}$C higher between July and December. As a deep lake, total river discharge to Paiku Co was about 800-900 mm water equivalent to lake level and accounted for 2-2.5% of total lake water storage. The river discharge can accumulatively decrease lake water temperature by ~0.1 $^{\circ}$C in a year. Therefore, as a deep lake, the influence
360   of river discharge on the total lake heat storage at Paiku Co is very small and can be neglected. Therefore, we do not consider the influence of Av on the lake energy budget in this study.

[Figure]

Fig. S2. Comparison of water temperature between Bulaqu river (red line) and Paik Co (blue)

*- L114-115. ": : :therefore: : :we do not consider : : :G.." There is no mention of the reason why G can be neglected.*

Reply: We do not considered the heat transfer between lake water and bottom sediment in this study because there is no data available.

*- L153-155, ": : :reduction in wind speed (data not shown)"; As I recall from the reply of authors to the reviewers' comments in the previous version, authors do not have wind speed data. Then the statement of "data not shown" is misleading. When we see this statement, we tend to believe that there were data and authors checked them to validate what is written in the manuscript even though they are not shown in the manuscript with a figure or a table. If you do not have data, then you should not mention wind influence as if it was based on data. There are similar statements on wind speed here and there in the manuscript. Authors need to remove them or change expressions. Alternatively, authors could rely on wind speed from reanalysis data. However, the reliability of any reanalysis data set should be established first (perhaps by referring to previous studies) for the study area before they can use the reanalysis data.*

Reply: Here wind speed at Qomolangma station is used to compare with lake surface temperature changes. Qomolangma station is located at the northern slope of Mount Everest, about 150 km east of Paiku Co. If this data is just used as a simple comparison, this is no problem because it does not need high accuracy. But if wind speed at Qomolangma station is used to calculate lake evaporation, maybe it has too large spatial difference.

[Figure]

Comparison of lake water temperature and average wind speed. Daily wind speed at Qomolangma station is used.

*- L179 "..but also the bottom water"; I do not understand what authors want to claim.*

Reply: We have deleted this sentence in the revision.

*- L200-201; "....Lower temperature gradient caused stronger water convection....."; I do not understand the logic in this part. I assume water convection is stronger when the vertical gradient is larger.*

Reply: According to my knowledge, water convection is weaker when the temperature vertical gradient is larger. For example, water convection is weaker in August when the temperature vertical gradient is large, because the lake water is stratified. Similarly, water convection is strong in winter when the temperature vertical gradient is weak because the lake water is mixed.

*- L204-205 "(Fig.1)"; Fig. 1 shows the locations of water level loggers but authors are talking about water temperature. In Fig.1, there are also the locations of water temperature measurements. This is confusing.*

Reply: We have revised the caption of Fig. 1. HOBO water level logger can not only record water level changes, but also water temperature changes.

*- L208-209 "....large errors can result if only water temperature data collected at the shoreline are used to calculate lake*

400 *heat storage and energy budget."; Similarly, errors can result if only water temperature data collected at the center of the lake are used. The authors should acknowledge this possibility to make analysis accordingly (see major comment).*

Reply: We have deleted this paragraph in the revision.

*- L223 " Indian summer monsoon precipitation"; there is no mention of heat advection due to precipitation in the application*

405 *of the Bowen ratio method. All energy sources that are used for the turbulent neat fluxes should be considered and mentioned.*

Reply: The influence of heat advection due to precipitation on lake energy budget has been mentioned in the revision (Line 118-124).

410 *- L257-259 " Negative value...., indicating the lake water absorbed energy from the overlying atmosphere. Positive value...., indicating the lake water released energy to the overlying atmosphere." These statements are not correct. The sign of the Bowen ratio simply indicates whether the fluxes are in the same direction (positive), or different direction (negative).*

Reply: We have deleted this sentence in the revision.

415 *- L274-276 " Lake evaporation between middle January and April is not determined... "; Why not? The authors do give latent heat flux for this period. If it is not certain whether the lake surface is covered with solid water or liquid water, then authors could give two values of evaporation. One in the case of the ice surface, and another one in the case of liquid water surface. The true evaporation is somewhere in between. This can be used together with the evaporation estimate obtained by assuming it is the same as water level change in L290-291.*

420 Reply: We have addressed these sentences in more detail (Line 220-222). We do not determine lake evaporation between January and April because the lake surface is sometimes covered by ice. Energy budget over the lake surface during ice covered season is different from that without lake ice.

*- L288 ".....lake ice can effectively prohibit evaporation."; Is this true? How about sublimation? Is the latent heat flux on ice-*

425 *covered lake zero? The authors could add references to support their statement.*

Reply: We agree that lake water can still get lost through sublimation. We have revised this sentence in the revision (Line 256-257).

*- L290-291 "Assuming lake evaporation between January and April is equal to lake level decrease ..."; the Authors should*

430 *provide an error estimate of evaporation based on this assumption. Errors due to lake level measurements, mean lake water level estimation, water level-volume relation, water level-lake surface area relation, etc.*

Reply: Thanks for the good suggestion. We do not consider the lake evaporation between January and April in the revision.

*- L293 "20.4%"; the Authors should explain how this ratio was derived.*

435   Reply: We have deleted this sentence in the revision.

*- L299 "We set up a platform in the southern centre of Paiku Co"; The location should also be shown in Fig.1*

Reply: We have shown the location of the platform in Fig. 1 in the revision.

440   *- L302 "....fluctuated very similarly between..."; This is a subjective statement. In fact, I do not quite agree that they are VERY similar. A better presentation would be the determination of the difference between the two measurements and its error propagation into Bowen ratio and flux estimates (see major comment).*

Reply: Thanks for the suggestion. We have quantified the difference of the two data (Line 278-280).

445   *- L303 "...can be used to represent the general condition of the whole lake..."; This statement is based on a superficial analysis. It should be based on the error propagation analysis mentioned above (see also major comment).*

Reply: We have quantified the difference of the two data (Line 278-280).

*- L318-319 "Although there is some spatial difference, the similar seasonal patterns of energy budget and lake evaporation*

450   *at different sites indicate that our results are reliable."; Just like L208-209, authors should acknowledge the difference and estimate the magnitude of the error due to spatial difference, rather than ignoring the difference by simply saying "reliable". In fact, authors do give error estimates in L329-344. Thus the statement of "reliable" is not quite consistent with the error estimates.*

Reply: We have deleted this paragraph in the revision.

455

*- L320, Section 4.2. Here authors should give all possible error sources and their likely magnitude, and use them to give a total error in their evaporation estimates. They should include, among others, the error due to the use of temperature and humidity measurements at the shoreline. Also, when they talk about annual evaporation, there are possible errors due to the assumption in L290-291 (see major comment).*

460   Reply: Please see the reply to the major comment.

*- L326 "very similar seasonal fluctuations (R2=0.55....with standard deviation of 23.9 W·um-2.)"; With $R^2$=0.55, I do not think it is VERY similar. Why the standard deviation? ã˘AA˘RMS error is a more appropriate indicator of the similarity.*

Reply: We have quantified the difference between the two dataset. RMS error between the two data is estimated.

465

*- L326-327 " Assuming approximately 70% of the net radiation was consumed by lake evaporation (Lazhu et al., 2016).........*
*74.5 mm per year "; This percentage is from a different lake. Why not use estimates for Paiku Co. given in Table 2? The*
*authors should explain how 74.5 mm/year was derived.*

Reply: We have deleted this sentence in the revision.

470

*- L345- " Uncertainty of lake evaporation in this study was also validated by comparing lake level changes"; Just like the*
*case in L290-291, authors should provide error estimates for the evaporation estimates based on lake level measurements.*
*Possible advection due to precipitation should be addressed as commented above for L223.*

Reply: Thanks for the good suggestion. Lake level changes can be measured accurately (<1 cm), so error estimated is not
475    considered in this study. Precipitation is also not considered because it is usually very low in winter season.

*- L352 "As shown in Table 3, runoff at the three large rivers can contribute to lake level increase by 0.71.6"; Runoff values*
*in Table 3 are for a short period. Can you use them to estimate monthly runoff?*

Reply: Monthly lake runoff at Paiku Co can be estimated based on this measurement, but with large uncertainty. Therefore,
480    we just use daily runoff in this paper.

*- L355 "To further explore the impact of lake heat storage on the seasonal pattern of lake evaporation...."; Authors should*
*summarize at the end of section 4.3, what kind of new findings were obtained on the impact of heat storage to lake*
*evaporation from their measurements/analyses and comparison with previous studies. The phase shift of lake evaporation*
485    *due to lake heat storage is in a way common knowledge. We would like to hear something new here. How about differences*
*among the TP lakes? For example, in the introduction, the authors mention the difference of lake size change between the*
*interior TP and southern TP. Any new findings on this point? Also try to make clear the relation between the statements*
*made in the introduction (e.g., L58-67) and those in this section. You do not have to say similar things in different parts of*
*the manuscript.*

490    Reply: We have added a new section 3.4, Fig.6 and Fig.7 in the revision. The main findings are summarized in section 3.4
(Line 233-237). 'Fig.6a shows that positive lake water budget at Paiku Co mainly occurred during the summer monsoon
season (Jul. and Aug.), while negative water budget mainly occurred during the post monsoon season (Oct. to Dec.).
Precipitation and lake inflow are mainly concentrated during the summer monsoon season, meanwhile lake evaporation is
relatively low (Fig. 6b, c), which together leads to the rapid lake level increase (40~60 cm). During the post monsoon season,
495    precipitation and lake inflow are already very low, meanwhile lake evaporation is in its high value, which leads to rapid lake
level decrease (~40 cm). Contrasting hydrological and thermal intensities play an important role in the seasonal lake
level variations of Paiku Co.'

500

Reply: In this study, we mainly focus on lakes on the TP (section 3.4). We add Fig.7 to show this phenomenon at other lakes

on the TP.

505  *- L359 "2011-2012", L362 "2013-2015"; Evaporation estimates for these periods are continuous even during the winter*
*season? Clarify this in the manuscript since there is a statement in the introduction saying "lake evaporation throughout the*
*year is not typically investigated".*

Reply: We have clarified this in the revision. As we have addressed in the introduction, lake evaporation derived from eddy

covariance during winter season is not investigated at most of the previous studies. Lake water temperature profile and

510  changes in lake heat storage are also not investigated. An except is Qinghai Lake, where water temperature at only the upper

3 m was investigated, but the whole water temperature profile is still unclear.

*- Fig.1; Add a scale, latitude/longitude to the lower right figure of the panel a.*

Reply: We have revised Fig.1 according to the suggestion.

515

*- Fig.5 caption; "depth of 0 m"; should it be 0.4 m?*

Reply: We have deleted this figure in the revision.

*- Fig. 9; Change the solid line into a dotted line when there are missing values for an extended period.*

520  Reply: Thanks, we have revised this Figure.

*- Fig. 10; what are the spikes in the evaporation values? They are weekly values so that they look strange. The authors*
*should explain this in the main text (perhaps in connection with error estimates).*

Reply: The spikes in the evaporation values may come from changes in lake heat storage. Changes in lake storage mainly

525  depend on lake water temperature change, which is measured at an interval of 5-10m. So, large water temperature changes

can cause large changes in lake heat storage and spikes of lake evaporation. If lake water temperature on vertical profile is

measured more densely, changes in lake heat storage can be quantified more accurately.

*- Fig. 11; explain in the main text what the averaging period of the plotted data is.*

530  Reply: We have addressed this in Fig.8 in the revision.

*- Table 1; add information on GMX600.*

Reply: Thanks for the suggestion. We have added information about GMX600 in Tab.1 in the revision.

535

540

545

550

555

560

[revised manuscript text omitted]
. Meanwhile, energy budget over the lake surface is seldom investigated due to the lack of in situ measurements of lake thermal structure on the TP. However, lake evaporation throughout the year is not typically investigated because it is difficult to install and maintain measurement platform due to the harsh natural conditions on the TP and the influence of lake ice during the late autumn and early winter. As a result As a result, how lake evaporation affects seasonal lake level changes variations on the TP remains unclear due to the lack of comprehensive observations of lake water budget on the TP.

[revised manuscript text omitted]

[Figure]

**Figure 8: A comparison of air temperature and water vapour pressure between shoreline and lake centre.**

---

## Referee Report (RR1)

The manuscript describe clearly the hydrometeorology conditions of a high-elevation lake on the west part of Tibetan Plateau, which is important for lake evaporation research of high-elevation lakes on the Tibetan Plateau. I think the revised form can be accepted after several minor corrections as follows:

(1) In Abstract (Line 16), the sentence read a little strange and the structure of the sentence should be revised.

(2) Line 21 and line 354, it should be "5 months"

(3) In line 134, "Ra" is not defined in the revised manuscript.

(4) In line 211, I could understand that Bowen ratio is used to allocate sensible heat flux and latent heat flux following energy budget equation. But I think "Bowen ratio method" appear here is inappropriate.

(5) In line 303, "Runoff are shown in Tab. 3." "in" should be added.

(6) Line 348, the sentence is suggested to be "the bottom water reached to its highest value two months later"

(7) In line 349, the sentence is suggested to be "The thermocline depth formed between 15 m and 25 m".

---

## Referee Report (RR2)

The submitted manuscript of "Contrasting hydrological and thermal intensities determine seasonal lake-level variations A case study at Paiku Co on the southern Tibetan Plateau" presented comprehensive hydrometeorological observations of a high-elevation large lake on the southwest part of Tibetan Plateau, where the lake process measurements are very limited and the results of lake processes in this area are also rare. Under this background, the long-term hydrometeorological measurements in Lake Paiku Co show high significance for our understanding on high-elevation lake processes of the Tibetan Plateau. And the detailed analysis of lake evaporation through Bowen ratio based energy budget method show also reasonable results. As the most uncertainties raised by meteorological measurements have been discussed in section 3.6 of the resubmitted manuscript, I consider the work can push forward our understanding of the lake processes and ware resources evaluation on the high-elevation lakes of the Tibetan Plateau. I suggest the manuscript to be "**accepted subject to minor revisions**", and the following two comments should be addressed before publication.

1, For the discussion on the relationship between lake evaporation and lake-level variation, the **sub-surface water inflow and outflow** present the highest uncertainties, but are lacking and considered to be not important in the manuscript. But this may not be the case. From my own experience, the subsurface flows are very important for lake-level variations and should be considered with a great attention. Considering the importance of sub surface flows, the related contents can not appear as results in "abstract" and "conclusions". As most of the contents focus on the Paiku Co, the last sentence in abstract should be revised to exclude the information on "deep and shallow lakes and the southern and northern lakes".

2, in section 3.2, "the main components of energy budget over the lake surface, including solar radiation,······ from the lake body", generally all the three variables of "solar radiation, atmospheric longwave radiation to the lake and upward longwave radiation from the lake body" belongs to the radiation budget, rather than energy budget in references. Thus, I suggest to revise "energy budget" to "radiation budget" in this sentence.

With these revisions, I think the contents can be published for public with a detailed description of the hydrometeorological study in Lake Paiku Co.

---

## Author Response (AR2)

**Editor' comments**

I have now received comments from three referees and after consulting them, I would like to ask you to consider the following issues:

5   1)   provide precise description of used measurements (temperature and humidity), their errors and time period of averaging used for the estimation of lake evaporation,

Response: We are very grateful to the reviewers' comments. We have made detailed description about used measurement (temperature and humidity) in 'Data acquisition' part (Line 102-107). 'Two loggers were installed in an outcrop ~2 m above the lake surface. One is located in the north shoreline of Paiku Co, the other is located in the central shoreline of Paiku Co
10   (Fig. 1). The instruments were under large rock where there was a hole facing the lake. The monitoring site was ventilated and therefore the meteorological condition over the lake surface can be well recorded. Air temperature and relative humidity were recorded at an interval of 1 hour and daily-averaged values were used in this study. The air temperature and humidity measurements in the shoreline were further validated by a simple AWS (GMX 600) in the Paiku Co's southern center (Section 4).'

15   Their uncertainties are estimated in section 3.6 (Line 363-371). 'To validate its representativeness, we set up a platform in the southern centre of Paiku Co in September 2019 (water depth: 19 m; least distance from shoreline: 2 km) and a simple AWS station (GMX600) was installed on the platform. Meteorological data between 22 September and 26 October were acquired. We made a comparison of meteorological data between shoreline and lake centre. Results show that both air temperature and relative humidity fluctuated similarly between the shoreline and lake centre (Fig. 9), indicating the
20   meteorological data from the shoreline of Paiku Co can be used to represent the general condition of the whole lake at least during the observed period. The RMSE of daily air temperature and water vapour pressure in the shoreline was estimated to be 0.91 $^{\circ}$C and 0.069 kPa. The uncertainty of Bowen ratio was estimated to be 0.05 during this period by using those in the lake centre as true value, which corresponds to 5.6 W m$^{-2}$ of latent heat flux.'

About the time period, changes in lake heat storage, sensible and latent heat fluxes and lake evaporation used in this study
25   were calculated at weekly interval. All the other components used are daily values. We have clarified this in several places in the main text (Line 96, 106, 113, 159-160, 168-170).

2)   provide (detailed) justification to estimate the above-the-lake Bowen ratio using the on-shore measurements,

Response: Air temperature, humidity and Bowen ratio using the on-shore measurements are validated by using measurement
30   in the lake center (Fig. 9). The RMSE of daily air temperature and water vapour pressure in the shoreline was estimated to be 0.91 $^{\circ}$C and 0.069 kPa by using those in the lake centre as true value. The uncertainty of Bowen ratio is estimated to be 0.05 during this period. We added the comparison of Bowen ratio between lake center and shoreline in Figure 9c.

3)   provide estimates of error propagation to the Bowen ratio and latent/sensible and fluxes.

35    Response: The accuracy of Bowen ratio and heat fluxes are estimated by error propagation in the revision (section 3.6). The uncertainty of weekly Bowen ratio is estimated to be 0.05. The uncertainties of latent heat flux and lake evaporation are estimated to be 17.2 $Wm^{-2}$ and 0.6 mm/day. The uncertainty of total lake evaporation amount is estimated to be 146 mm during ice-free period (May to Dec).

50

55

60

65

**Reviewer 1**

70

Major comment

1) As section 3.4 with Fig. 6 would be the most important parts, in-depth explanation with discussions are expected with re-structure chapters. If you want to divide Result and Discussion, the Discussion part should contain comprehensive issues mentioned in the Result. As the 4.1 treats some kind of execution for observational accuracy, beside 4.2 compared with other

75 lakes overlapping to 3.4, I would like to recommend, delete "4 Discussion", "5 Conclusion" may change to "5 Summary and discussion" if possible where the problems or future challenges such as 4.1 should be covered there, and 4.2 should move to Chapter. 3. (if the article HESS format accept.)

Response: We are very grateful to the constructive comments. According to your suggestions, we re-organized the structure of the paper in the revision. We have combined 'Results' and 'Discussion' sections into 'Result and discussion' (Section 3)

80 in the revision. We believe that it would be too long if 'Conclusion' part becomes 'Summary and discussion'. More explanation about the impact of lake evaporation on seasonal lake level variations on the TP is given in Section 3.5 (Line 321-336).

2) Data and units are very important for papers based on the observational confidence, and many of them are missing. There

85 are also uncertainties in the label and amount calculation. For instance, Fig.4, 5 are all daily value? What is the original observation interval to calculate heat/water budget? How did you filter errors/missing with correction or interpolation to born Fig. 4-6 ? In Fig.6, is this evaporation/precipitation rate? If so, they should be mm/???.

Response: Thanks for pointing out these errors. In this study, changes in lake heat storage, sensible and latent heat fluxes and lake evaporation were calculated weekly. All the other components used are daily values. For original air temperature and

90 humidity in the shoreline and lake water temperature profile in the lake center, hourly values were monitored and daily values were used. We have clarified this in the main text (Line 96, 106, 113, 159-160, 168-170).

We have checked the unit of Fig. 4-6. The unit of lake evaporation should be mm/day. We have also addressed the units in their captions of Figures 4, 5.

95 3) Total river discharge was about 800-900 mm (per what??), and Av was not considered (L128). On the other hand, at L226, total evaporation for M-D was 975 mm and that is a matter of concern. They are the same order. By means of heat budget, yes that heat advection and heat needed for evaporation/condensation is quite different. But, still I can not understand the logic explaining the lake water level changes only by the evaporation term (river discharge data is missing in post-monsoon at Fig. 6b). Problem would be that there is no precise explanation about the water budget depending on the season including

100 precipitation amount at around L125. Also, G is ignored (L128) but calculated at L151. Very strange.

Response: The unit of total river discharge was mm/year equivalent to lake level. The heat transfer between lake water and bottom sediment should be expressed as 'G' and changes in lake heat storage should be expressed as 'S'. We have revised them in the revision (Line 126, 162).

Total river discharge is close to total evaporation which is about 800-1000 mm equivalent to lake level. The amount of total river discharge accounts for 2-2.5% of the total lake water storage (mean depth 42 m). In-situ observation shows that lake water temperature is 2-6 $^{o}$C higher in summer than river water. We can see that the river discharge can accumulatively decrease lake water temperature by ~0.1 $^{o}$C in summer, which corresponds to 2.1 W $m^{-2}$ of heat flux between July and September and 0.07 mm/day of lake evaporation. Therefore, the impact of river discharge and precipitation on the energy budget at Paiku Co is relatively small and can be neglected. We have given a detailed description about this in the revision (Line 130-137).

4) Seasonal variation of wind speed, as shown in the reply sheet, is valuable to explain the characteristics of lake temperature stratification. Why don't you add the wind speed variation on the Fig. 2 ?

Response: Thanks for the suggestion. We have added a comparison with the wind speed variations in Fig. S2 in the revision.

Other comments

L16    lake evaporation was "estimated" ?

Response: We have revised the sentence as 'lake evaporation and its effects on seasonal lake level variations were investigated at Paiku Co on the southern TP based on in-situ observations of lake thermal structure and hydrometeorology (2015-2018)'.

L18 results showed ,, (results should be passed sentence)

Response: We have revised it.

L23 It is better to show lake evaporation amount for pre-monsoon, monsoon, and pos-monsoon, respectively, to emphasis their seasonality.

Response: In the revision, we have shown the amount of lake evaporation during the pre-monsoon, and pos-monsoon season as you suggest (Line 18-19).

L25 Better to unify "summer monsoon" and "monsoon".

Response: Thanks for the suggestion. We use 'monsoon season' in the revision.

L27 "shallow lake" ,, again, please explain which level is shallow and which level is deep ? Also, lake depth should be added at L49,51.

135

Response: Shallow or deep lake in this study is not defined in this study. What we want to address is that deep lakes have larger seasonal lake level fluctuations than relatively shallow lakes due to the different heat storage and seasonal pattern of lake evaporation.

140 L83 Paiku Co tends to lower the lake level. So, how this evidence relates to this study's results? Better to be discussed in the last chapter.

Response: We have previously showed that the main cause of lake level decrease was mainly related to decrease in precipitation during the past decades (Lei et al., 2018). In this study, we mainly focus on the impact of lake evaporation on 145 different amplitude of seasonal lake level changes on the TP.

L119 Just my impression that G should be $\Delta S$ and S should be G? Because, G came from Ground, and S is Storage.
Response: Thanks for the suggestion. We have revised the G and S in the revision (Line 127).

150 Fig. 4 Spell out SVD,LWD,LWU
Response: We have spelled out them in caption of the figure.

Please check the order of reference again.
Response: We have carefully checked the order of the references.
155

160

165

**Reviewer 2**

170

(1) In Abstract (Line 16), the sentence read a little strange and the structure of the sentence should be revised.

Response: Thanks for the positive comment very much. We have revised the sentence (Line 15). ' lake evaporation at Paiku Co on the southern TP and its effect on seasonal lake level variations were investigated'.

175 (2) Line 21 and line 354, it should be "5 months"

Response: We have revised it.

(3) In line 134, "Ra" is not defined in the revised manuscript.

Response: Thanks for pointing out the error. 'Ra' should be 'Rl'.

180

(4) In line 211, I could understand that Bowen ratio is used to allocate sensible heat flux and latent heat flux following energy budget equation. But I think "Bowen ratio method" appear here is inappropriate.

Response: We agree that this sentence is inappropriate there and have deleted it in the revision.

185 (5) In line 303, "Runoff are shown in Tab. 3." "in" should be added.

Response: We have revised this sentence.

(6) Line 348, the sentence is suggested to be "the bottom water reached to its highest value two months later"

Response: Thanks for pointing out this error. We have revised it.

190

(7) In line 349, the sentence is suggested to be "The thermocline depth formed between 15 m and 25 m".

Response: We have revised this sentence in the revision, "The thermocline formed at the depth of 15~25 m".

195

200

**Reviewer 3**

Main issue:

It has been the accuracy of the estimated evaporation in their study. They used on-the-shore measurements of temperature and humidity with an instrument designed for indoor use without ventilation/radiation shield (except for an instrument container box) to estimate the above-the-lake Bowen ratio. Since this is not where and how they should be measured, it is the responsibility of the authors that their measurements produce evapotranspiration with sufficient accuracy to allow them to claim their findings are meaningful against errors. My suggestion of the previous review is (1) to identify all the error sources and estimate each error magnitude, and then (2) to estimate error propagation to the Bowen ratio and latent/sensible and fluxes. Authors did (1) but did not go on to carry out (2). They do not explain why they did not in their responses (except perhaps in L284 saying "the difference of 0.6℃ .....is used to estimate the uncertainty of lake evaporation" but this estimation is not shown in the main text. Authors should respond (item-by-item) to every comment made by reviewers. If they do not agree, that is fine. But they need to explain why with supporting evidence. Please also note that some of the error estimates are for the whole year. But what is really needed is the seasonal changes of errors since they focus on seasonal changes. We need to know if the magnitude of the seasonal variation of evaporation shown by the authors is larger than the error bars. Note also that they do give their error estimate of evapotranspiration based on water balance. However, this is a rather crude estimate (see minor comment on L302-310 given below) and should be supplemented with the micrometeorological error estimates.

Response: Many thanks for the very detailed and constructive comments so many times, which are really very important for improving the manuscript. We have further revised manuscript according to these comments. Although the error estimation based on lake water balance is regarded as crude, we believe that it is a robust method, so we keep it in the revision. We further estimated the uncertainties of Bowen ratio and lake evaporation by using error propagation in the revision in section 3.6. The uncertainty of lake evaporation is estimated to be 0.6 mm/day and the total error of lake evaporation amount is 146 mm during the ice free period. From Fig. 6, we can see that the seasonal variations of lake evaporation in this study are reliable compared with its uncertainty.

We gave a more detailed description about the air temperature and humidity measurement in section 2.2 (Line 92-98). The uncertainty of air temperature and humidity in the shoreline was validated by data in the lake center in Section 3.6 (Line 365-373). 'To validate its representativeness, we set up a platform in the southern centre of Paiku Co in September 2019 (water depth: 19 m; least distance from shoreline: 2 km) and a simple AWS station (GMX600) was installed on the platform. Meteorological data between 22 September and 26 October were acquired. We made a comparison of meteorological data between shoreline and lake centre. Results show that both air temperature and relative humidity fluctuated similarly between the shoreline and lake centre (Fig. 9), indicating the meteorological data from the shoreline of Paiku Co can be used to represent the general condition of the whole lake at least during the observed period. The RMSE of daily air temperature and

235  water vapour pressure in the shoreline was estimated to be 0.91 °C and 0.069 kPa. The error of weekly Bowen ratio was estimated to be 0.02 during this period by using those in the lake centre as true value.'

Here we further make an estimation of seasonal changes in Bowen ratio uncertainty. According to Taylor expansion, uncertainty of Bowen ratio can be expressed as:

$$\delta B = 3.57 \times 10^{-2} \times \sqrt{\left(\frac{\Delta T_a}{T_s - T_a}\right)^2 + \left(\frac{\Delta e_a}{e_s - e_a}\right)^2}$$

Here $\delta T_a = 0.91°C$, $\delta e_a = 0.069$ kPa are used according to measurement in the center from 22 September to 26 October
240  2019. The results show that uncertainty of Bowen ration is 0.02 in pre-monsoon season, 0.02 in monsoon season and 0.008 in post-monsoon season. The uncertainty of seasonal latent heat flux is estimated to 2.7 W m$^{-2}$ in pre-monsoon, 1.2 W m$^{-2}$ in monsoon season and 1.2 W m$^{-2}$ post monsoon season. We can see that seasonal changes in Bowen ration uncertainty and latent heat flux uncertainty are relatively small compared with their annual average. Therefore, uncertainties of Bowen ratio and evaporation during the entire ice-free period are estimated and used in this study.

245

Specific comments:

Title: "in the central Himalayas"; in their manuscript, they mainly explain lake evaporation and water level in the Tibetan Plateau, and not in the central Himalayas. Since not all HESS readers are familiar with the geography over there, authors should mention the relation between the two, or they should only use the Tibetan Plateau throughout the manuscript.
250  Response: Thanks for the suggestion. We use Tibetan Plateau throughout the manuscript instead of Himalayas in the revision.

L53-54 "The main causes ....still unclear"; aren't there any theories or hypotheses presented in previous studies? Since this will be discussed as the main point of this study in section 4.2, previously proposed ideas should be thoroughly reviewed and explained here.
255  Response: As far as I know, Phan et al (2012) found this phenomenon for the first time based on ICESat satellite altimetry data, but the main causes are still not investigated until now.

L56, L321 "direct measurement... energy budget method"; in my previous review, I made a comment on this saying that Bowen ratio or energy balance method is not generally considered a direct method. In the response of the authors, they
260  simply say, "We have deleted this paragraph."(their response, L333). I take it they have agreed. Then they should revise the same point throughout their manuscript. If they do not agree, they should explain why.
Response: We agree that energy budget method is an indirect method of lake evaporation. We re-organized the sentence as 'Both the eddy covariance system and energy budget method are effective ways to determine lake evaporation (Line 55-56)'.

265 L63-66 "...not typically investigated through eddy covariance system because....difficult ... measurement platform"; if it is the platform issue, then the same applies to the Bowen ratio system, doesn't it?

Response: We agree that both methods have same issue. In this study, we try to avoid this issue by installing the water temperature loggers below lake ice and installing air temperature and humidity loggers in the shoreline. Although this method has its shortcoming, it is more suitable for long-term monitoring of lake evaporation.

270

L70 "ice-free period"; they do report evaporation and latent heat flux during the ice-covered period in Figs.5-6.

Response: Thanks for pointing out this mistake. Lake evaporation based on energy budget method during the ice-covered period may not be reliable because lake ice can significantly affect the energy balance over the lake surface. Therefore, we do not show it in Figures 7 in the revision.

275

L82 "highest shoreline"; is this paleo-shoreline?

Response: We have revised it.

L93 "water temperature profiles were"; add sensors after profile to write "water temperature profile sensors were".

280 Response: We have revised this sentence as 'Two water temperature profiles were set up in Paiku Co's southern …' (Line 92-93).

L96 "daily-averaged values"; I assume they are used only for thermal regime analysis of the lake. For the application of the energy balance method, weekly-averaged values are used, aren't they? Please clarify in the main text.

285 Response: We have clarified this in the main text in several places (Line 96, 106, 113, 159-160, 168-170). In this study, changes in lake heat storage, sensible and latent heat fluxes and lake evaporation were calculated at weekly interval. All the other components used are daily values.

L100 "over the lake"; this does not express the exact location of their measurement. It should be something like "above the shoreline".

290 Response: We have changed 'over the lake' to 'above the shoreline'.

L103 "accuracy"; mention that it is the accuracy of sensors without considering the error arising from lack of ventilation and a radiation shield.

295 Response: We agree that there may be some errors due to ventilation and radiation shield. We have further evaluated the accuracy of air temperature and humidity by using data from the lake center (Line 365-373).

L103 "Fig.2" this is the wrong figure number.

Response: Thanks for pointing out this error. We have deleted this figure number.

300

L128 "very small and can be neglected"; they make this statement based on the estimated annual lake water temperature decrease (about 0.1 ℃). However, it should be based on the estimation of seasonal impact on the storage in the energy unit. The impact should be different depending on the season (e.g., the 2-4 ℃ higher water flows into the lake from July to December), and even a small temperature change could produce a large storage change since water volume is quite large.

305 Response: We have further addressed this in the revision (Line 130-137). Lake water temperature was almost same to river water temperature between April and June, but 2-4$^o$C higher between July and December (Fig. S3). As a deep lake, total river discharge to Paiku Co was about 800-900 mm water equivalent to lake level and accounted for 2-2.5% of total lake water storage. The river discharge can accumulatively decrease lake water temperature by ~0.1 $^o$C in a year, which corresponds to 2.1 W•m-2 of heat flux error between July and September and 0.07 mm/day of evaporation uncertainty.

310 Therefore, the influence of river discharge and precipitation on the total lake heat storage at Paiku Co is relatively small and can be neglected.

L128 "we do not consider...G"; they do consider G with Eq. (5). Perhaps this is precipitation? But then no discussion is made on the impact of rainfall advection. It should be made if G should be replaced with precipitation. If G represents "the heat transfer between the lake water and bottom sediment" (as mentioned in their response, L366-367), they still need to explain in the text why it was ignored. Note that the reason "because there is no data available" (their response, L366-367) cannot be a valid reason by itself; the reason has to be, first, scientific (and then perhaps practical).

Response: Thanks for pointing out this. In Eq. (5), G should be S (changes in lake heat storage). In fact, the heat transfer between the lake water and bottom sediment (G) is not considered. We explain it in the main text (Line 129-136)

320 'Meanwhile, the heat transfer between lake water and bottom sediment (G) is also neglected because groundwater discharge is usually much less than surface runoff and geothermal is not detected at the lake bottom of Paiku Co'.

L134 "Ra"; is this Rl?

Response: Thanks for pointing out this error. We have revised it in the revision.

325

L141-142 "the two uncertainties...... cancel each other ...daily"; I do not think this is always true, and whether or not they cancel out depends on the specific condition of the diurnal changes of wind speed, solar radiation, etc. For example, Wilson et al. (2013, doi:10.1002/jgrd.50786) and Sugita et al. (2020, doi:10.1029/2020WR027173) report the cases where daytime and nighttime values do not cancel out. I am sure some reports indicate mutual cancelation. It is condition-specific.

330 Response: We agree that lake water temperature at the depth of 0.4-0.8 m is different from the lake 'skin' temperature. In this study, the difference between skin temperature and bulk temperature are investigated by comparing with MODIS LST data (Line 346-355). 'In spring and summer when the lake water gets warm, the skin temperature derived from MODIS data

is about 1.2 °C higher than lake body temperature. In autumn and winter when the lake water gets cool, the skin temperature derived from MODIS data is about 0.05 °C higher than lake body temperature. Therefore, the mean difference between lake surface temperature and in-situ observation is estimated to be 0.6 °C for a whole year.'

L151 Eq. (5); in this equation, the summation is from i=0 to 72.8. This should be from 0 to 13? What would you do when the water level changes? Layer number remains the same, but the representative depth of each level changes?

Response: I think it should be 72.8 because it is an integral of water depth. The maximum water depth at Paiku Co is 72.8 m. 13 layer is used in calculation. The water level changes can surely affect the depth of water temperature loggers, but it is very small relative to the large water depth. Generally, Paiku Co's lake level fluctuates in a range of 0.4-0.5 m, which is only about 1% of the mean water depth. So, we believed that the seasonal water level changes have only minor impact on the changes in lake heat storage.

L166 "reduction in wind speed (data not shown)" and L175 "significant increased wind speed.(data not shown)"; I commented on this in the previous review report. The authors' response explains that they obtained this result of "reduction" based on the Qomolangma station measurements (their response, L377-380). Then they need to explain this fact in the main text, perhaps in section 2.2 around L112. Also, they need to explain how similar the wind speeds are between the Qomolangma station and Paiku Co to make their statement valid. By the way, wind direction should also be an important factor, particularly for the estimation of measurement errors with the sensors on the shore. When wind direction is from the lake to the shoreline, this is a better situation while the opposite condition would produce a larger error.

Response: We agree that different wind directions have different representativeness of lake conditions. If we estimate hourly or daily lake evaporation, we should consider wind direction. In our case, we estimate weekly-mean evaporation error, so wind direction is not considered.

The climate of the TP is mainly dominated by the westerlies in winter, and mainly controlled by Indian summer monsoon summer. Since both the Qomolangma station and Paiku Co are located in the northern slope of the Himalayas, the climate, altitude and topography are all similar, so we believe that seasonal changes in wind speed are similar between the two places. We have added a new Figure (Figure S2) about the comparison with wind speed in the revision and given an explanation about this (Line 178-179). 'The occurrence of thermal stratification corresponded to a significant reduction in wind speed. Generally, average wind speed was relatively low between July and the middle of October, but high in other months (Fig. S2)'.

L186 "lakes on the TP are considerably lower than those in other parts of the world (Livingstone,......)"; two references are not available in the reference list, so I cannot check what "other parts of the world" means. Valid comparisons should perhaps be made between lakes in the TP with lakes of similar size and depth, and in the same latitude but in lower altitude.

Response: We checked the reference list and found that the two references are in the reference list. We selected some lakes in other parts of the world with similar area and depth, information about these lakes are given in the main text, for example area, depth and location (Line 197-208).

370 L187 "due to the low lake surface temperature"; please explain the logic here. I would think that when summer surface temperature is low, the temperature in a deeper layer should also become low. Therefore gradient may not be affected much by this fact.

Response: We gave further explanation about this in the main text (Line 197-208).' The vertical temperature gradients of Paiku Co and other lakes on the TP are considerably lower compared with those in other parts of the world, for example

375 Lake Qiaodaohu (area: 580 km$^2$, maximum depth: 108 m) in east China (Zhang et al., 2015), Lake Zurich (area: 65 km$^2$, maximum depth: 136 m) on the Swiss Plateau (Livingstone, 2003) and Lake Simcoe (area: 580 km$^2$, maximum depth: >40 m) in Canada (Stainsby et al., 2011). This may be mainly related to the high elevation of Paiku Co. Yang et al (2010) showed that although downward shortwave radiation received by the TP is considerably higher than the surrounding lower elevation region, the downward longwave radiation is significantly lower. Due to the elevation effect, the highest air temperature at

380 Paiku Co is only ~12 $^o$C in summer. The lake surface temperature of Paiku Co (13 $^o$C) is considerably lower in summer compared with lakes in other parts of the world (e.g. Lake Qiaodaohu: ~32 $^o$C, Lake Zurich: ~22 $^o$C and Lake Simcoe: ~22 $^o$C), while the bottom water temperature (7 $^o$C) does not show much difference with these lakes (e.g. Lake Qiaodaohu: ~10 $^o$C, Lake Zurich: ~5 $^o$C, and Lake Simcoe: ~4 $^o$C)'.

385 L190 "3.2 Energy budget"; with this title, the sensible and latent heat fluxes should be treated here. Currently, it is in "3.3 Lake evaporation" section.

Response: We agree. In the revision, we have moved the sensible and latent heat fluxes to 'energy budget' section (section 2.2, line 230-239).

390 L196-198; are these numbers in the parentheses the averages during the specified period? Please indicate what it is in the main text.

Response: Yes, there numbers are the averages during the specified period. To make it easier to understand, we revised the two sentences (Line 216-219). 'The net radiation over Paiku Co varied seasonally in a range of 19.0~212.1 W m$^{-2}$, with an average value of 125.8 W m$^{-2}$. Relatively high net radiation occurred from April to August, with the highest value of 212.1

395 W m$^{-2}$ in June. Relatively low net radiation occurred from October to February, with the lowest value of 19.7 W m$^{-2}$ in December.'

L222 "during the ice-free season is shown in Fig.6b"; Fig.6(a), rather than 6(b), does show evaporation during the lake ice periods.

400 Response: Thanks for pointing out this. We have corrected it.

L226-228 "lake evaporation ....not determined because ....covered by the lake ice..."; since it is reported in Fig.6(a), why not? The presentation has to be consistent throughout the manuscript.

Response: We believe that lake evaporation estimated during the ice covered season is not reliable because lake ice can
405 significantly affect the energy balance over the lake surface. So the lake evaporation between January and April is not shown in the revision.

L240 "Fig. 6b,c)"; they should be 6a and 6b?

Response: Thanks for pointing out this error, we have corrected them.
410

L290-291 "MODIS derived lake surface temperature"; add information/reference of this product. Is this an instantaneous value? Or is it some kind of time averages? How many data points were used to derive the difference? And spatial resolution? Is it small enough to match with in-situ measurements, without the land surfaces in the field-of-view?

Response: Thanks for the suggestion. We have added some information about MODIS product (line 348-350). For MODIS
415 Terra and Aqua land surface temperature products, two instantaneous observations were collected every day (Terra: approximately 10:30 and 22:30 local time, Aqua: approximately 13:30 and 01:30 local time). Aqua MODIS 8-day land surface temperature products (MYD11A2 V006) were used to determine the difference of lake surface temperature and bulk temperature. The product was produced with spatial resolution of about 1 km and the accuracy is estimated to be 1 K under clear sky conditions. The MODIS 8-day data is the averaged lake surface temperature of daily MODIS product over eight
420 days (Wan Z., 2013).

Wan, Z.: Collection-6 MODIS land surface temperature products users' guide. ICESS, University of California, Santa Barbara. 2013.

425 L295 "whole year"; estimation should be made in shorter time intervals to allow seasonal changes of errors of the Bowen ratio/evaporation (see major comment).

Response: Please see the response to the main issue.

L302-310; is it correct that mean evaporation and mean lake level decrease were determined for May (one month) and Oct-
430 Dec (3 months)? Is it also correct that river runoff contribution was determined once in spring and once in Autumn for three years (so the total number of measurements are 6)? Is it reasonable to compare the 6-day measurements with one- to three-month averages? Please clarify and explain them in the main text. Then why only October discharge contribution was considered? Why not the spring contribution? Is the stated contribution of 1.2 mm/d (L307) correct? A simple average would

produce 1.0 mm/d. Also when you use water level, you need to consider the error magnitude of the water storage estimates
(L300).

Response: For the first question, error of lake evaporation during pre-monsoon and post monsoon is estimated by comparing with lake level decrease because precipitation and runoff are already low during these periods and lake level decrease is mainly due to evaporation. So when the runoff is determined, it is reasonable to give uncertainty of lake evaporation according to lake level decrease. Runoff becomes smaller from October to December, so when the runoff in early October is determined, the largest runoff can be estimated. Therefore, we may give the largest error of lake evaporation estimation using this method.

For the second and third question, runoff was determined in early October and June every year, so we can give the largest runoff in dry seasons. Therefore, our result can give the largest error of lake evaporation estimation during the study period.

For the last three questions, the lake level decrease (1.8 mm/day) is close to lake evaporation (1.8 mm/day) in pre-monsoon season, indicating that error of lake evaporation estimation is low. In post-monsoon season, the large discrepancy has to be partly due to precipitation and runoff. We have revised the error of lake evaporation to be 144 mm during the entire ice free period.

L303 "Runoff measurements"; add an explanation on how, when, and at what intervals the measurements were made in the method section.

Response: We have added information about runoff and lake level measurements in the method section (Line 116-123). 'As an important part of lake water budget, runoff was measured at three main rivers, Daqu, Bulaqu and Barixiongqu (Fig. 1). The water level of the three rivers was recorded automatically at 1 hour interval by using the HOBO water level logger (U20-001-01). River runoff was measured during the field expedition in spring (late May/early June) and autumn (late September/early October) using a LS1206B propeller-driven current meter (Nanjing Institute of Hydrological Automatization). Runoff was measured at least twice a day, including the largest runoff in the afternoon and lowest runoff in the morning. Meanwhile, lake level was monitored at 1 hour interval in the littoral zone of north Paiku Co (Lei et al., 2018). Daily lake level changes were used to compare with the seasonal pattern of lake evaporation in this study.'

L311 "4.2 Comparison with other lakes on the TP"; authors discuss the differences between deep and shallow lakes here. This is the issue explained in the introduction section as "unclear" so that it is good. Then how about another issue (difference of the range of seasonal lake level variation between southern and northern Tibetan Plateau) mentioned in the introduction section L45-46. Any new findings on that in this study?

Response: Thanks for the suggestion. We gave further explanation about it in the revision (Line 321-336). Our result may have further implication for the different amplitudes of seasonal lake level changes on the TP. Based on ICESat satellite altimetry data, Phan et al (2012) showed that there is larger amplitude of seasonal lake level changes on the southern TP than that on the northern TP. However, the main causes have not been investigated until now. Different seasonal patterns of lake

evaporation between the southern and northern TP may partly explain this. Generally, it is much colder on the northern TP than the southern TP due to the higher elevation and latitude (Maussion et al., 2014). Lakes on the northern TP usually freeze

470 up earlier and break up later relative to the southern TP (Kropacek et al., 2013), which results in longer ice cover duration on the northern TP (159-209 days on average) relative to the southern TP (126 days days). Longer ice cover duration on the northern TP may considerably reduce lake evaporation in post-monsoon season (Wang et al., 2020). Meanwhile, lakes on the southern TP are usually larger and deeper than those on the northern TP (e.g. Wang et al., 2009, 2010), which indicates that it can store more energy in spring and summer, and release it to the overlying atmosphere in autumn and early winter. For

475 endorheic lakes, relatively higher lake evaporation on the southern TP in post-monsoon season may lead to larger lake level decrease compared with those on the northern TP. Therefore, the combination of the lake ice phenology and seasonal pattern of lake evaporation may lead to the different amplitudes of lake level change between the southern and northern TP. Note that other factors including lake salinity and solar radiation may also have impact on the spatial difference of lake evaporation on the TP. More studies are still needed to quantify the impact of these factors on the total precipitation amount

480 and its seasonal distribution.

Table 1; the listed values of the accuracy of GMX600 are not accuracy but resolution. I cannot find the accuracy of "5%" in the specification of CNR4.
Response: Thanks for your detailed comments. We have revised these values in Table. 1.

485

Fig.6; change the unit of mm in the y-axis title to mm/month, mm/week, or mm/day.
Response: Thanks for pointing out the error. We have changed the unit of the y-axis.

Fig.7; add units to the y-axis. There are two grey rectangular shapes; add an explanation to each.
490 Response: We have added unit of the y-axis and added explanation of the grey rectangles.

Authors' response L348-350 "daily Bowen ratio is calculated...."; this should be explained in the main text around L145-150. As it is, the description is "all the input data were averaged at the weekly interval before lake evaporation was calculated" (L149-150), and therefore we come to believe that the Bowen ratio was calculated from weekly average temperature and
495 humidity.
Response: In this study, daily Bowen ratio is calculated and weekly averaged Bowen ratio is used to calculate lake evaporation.

Authors' response L407 "... heat advection due to precipitation... has been mentioned in the revision"; in the main text, it
500 simply says "the influence of ...precipitation is very small and can be neglected" (L127-128) without supporting evidence. Add evidence, please.

Response: Precipitation in the study area is about 150-200 mm, which is very small relative to large lake water depth and can only have minor impact on lake energy budget. So we do not consider its impact in this study.

505 Authors' response L530 "We have addressed... in Fig.8...."; I do not see it.

Response: Thanks, we have added the time period of the data used in the caption of Figure 9.

[revised manuscript text omitted]

[Figure]

1365 **Figure 9: A comparison of air temperature, water vapour pressure and Bowen ratio between central shoreline and lake centre.**

---

## Author Response (AR3)

**Reviwer 1#**

*The authors revised their manuscript by adding error propagation analysis. Unfortunately, their analysis is not complete.*
*Many of the points which were pointed out in previous reviews were not considered. As a result, I suspect that the magnitude*
*of error is underestimated. Just like the last review, I am in the opinion that this manuscript should be rejected at this point*
*after repeated reviews without successful revision.*

Response: Response: Thanks a lot for your constructive comments. We have revised the manuscript according to your comments. Energy budget method needs a lot of in-situ measurements, however, it is difficult to conduct some measurements due to the harsh natural condition and remoteness. We have to use some similar measurements instead. For example, we use lake surface temperature at the depth of ~0.4m to represent lake skin temperature. We used hydro-meteorological data in the shoreline to represent that in the lake center. We believe that more and more accurate measurements will be done at Paiku Co in the future.

*1, As pointed out before, the final error estimate should not be made on an annual basis since they are comparing seasonal*
*variations. Errors should be determined each season (month?) based on as much as possible of seasonal (monthly)*
*means/errors.*

Response: We have checked the four main factors that can cause uncertainty of lake evaporation, including solar radiation, changes in lake heat storage, lake surface temperature and meteorological data (temperature and humidity). For the uncertainties of solar radiation, changes in lake heat storage and meteorological data, we do not find there is considerable seasonal difference. For lake surface temperature, we agree that the uncertainty has seasonal difference. During the pre-monsoon season, the surface water is heated by solar radiation in the daytime and there may exist temperature gradient for the surface water. During the post-monsoon season, the surface water gets cool and can mix with the bottom water easily by wind and convection, so there is very small temperature gradient. However, the temperature gradient during the pre-monsoon season may not last long because the lake water can be mixed well by the strong wind in the afternoon. So, the seasonal difference between lake skin temperature and surface temperature is small on daily timescale. Therefore, uncertainties of Bowen ratio and evaporation during the entire ice-free period are estimated in this study.

*2, In the equation to estimate the error of Bowen ratio given in L238-239 of their reply, the errors arising from inaccuracy of*
*Ts and es are missing. They should be included. The error of Ts was estimated and is explained in L414-420 of the authors'*
*response. The authors compared 8-day average temperatures. However, since Bowen ratio was calculated daily, the Ts*
*error should also be estimated on daily basis.*

Response: MODIS 8-day data is used to estimate uncertainties of Ts because MODIS daily data is easily affected by cloud cover. MODIS 8-day data is 8 day averaged data after poor quality data has been removed. We agree that uncertainty estimation of Tw is very coarse, but no other higher quality data can be used. Therefore, when estimating the error of Bowen

35 ratio, we do not consider the errors arising from inaccuracy of Ts and es. As we have pointed out, the lake water can be mixed well by the strong wind in the afternoon in this high elevation area. The temperature difference may not be as large as that in low elevation area on daily scale.

*3. The error equation is given for the Bowen ratio in the authors' response so that it was easy to check their estimates.*
40 *However, for other components, methods and equations are not explained. I think the details of error estimates should be provided in their manuscript (perhaps as supplementary material).*
Response: We gave a detailed description about the uncertainty estimation in the revision so that we believe that there is need to show them in the supplementary material. Uncertainty of upward long-wave radiation and Bowen ration is estimated by error propagation (Line349-351, 370-372). Uncertainty of solar radiation is estimated by comparing Hamawari-8 satellite
45 data at Paiku Co and Qomolangma Station (Line 338-339). Uncertainty of changes in lake storage is estimated by comparing lake temperature changes at the northern and southern Paiku Co (Line 359-360). Meanwhile, the main results are already shown in the supplementary information (Fig S4, S5).

*4. In L 344- 347 of their main text, they state "The uncertainty in the atmospheric longwave radiation is not estimated, the*
50 *variations of solar radiation and atmospheric longwave radiation are usually opposite at a site, so their total uncertainty should not exceed the individual uncertainty". I am not familiar with the radiation regime up there, but isn't it unusual that solar radiation and atmospheric longwave radiation change in the opposite direction? When solar radiation increases and air temperature becomes higher with some phase shift, then longwave radiation should also increase. How come it is in opposite direction?*
55 Response: At a site, the dominant factor controlling downward solar radiation and longwave radiation is cloudiness. Clouds reduce solar radiation because of clouds' extinction but enhance longwave radiation because the emissivity for atmospheric longwave radiation is nearly 1 for cloudy sky but less than 0.8 for clear sky in Tibet due to low vapor pressure (see Yang et al., 2010). Therefore, when cloudiness increases, solar radiation decreases and longwave radiation increases, and vice versa.
Yang, K., He, J., Tang, W., Qin, J., and Cheng, C.: On downward shortwave and longwave radiations over high altitude
60 regions: Observation and modeling in the Tibetan Plateau, Agric. For. Meteorol., 150(1), 38-46, doi:10.1016/j.agrformet.2009.08.004, 2010

*5. The error estimate of the energy storage. As pointed out from the first review, there is another error source: water volume estimate that comes from bathymetry error and water level measurement error. The authors considered the difference*
65 *between the two lake center temperature profiles. How about likely differences of temperature between at lake centers and near the shorelines?*

Response: Lake volume was estimated in previous study. We take it as a constant value in this study (13 layers in total). Season lake level changes (~0.5 m) are far less than the average lake water depth, so its impact on the energy budget can be neglected.

70

*6. Advection. The authors conclude "Therefore, the impact of river discharge and precipitation on the energy budget at Paiku Co is relatively small and can be neglected." However, the magnitude of river advection is of a similar size to others, and thus the error arising from the omission of the advection term should be included in the error analysis. Note that they did not show evidence to claim the advection due to precipitation can be neglected. They need to show that if they want to say the above statement (this was made in the authors' response in L503-504; this should be mentioned in the main text).*

75

Response: Paiku Co is a deep alpine lake with an average depth of 42 m. The near balanced state in recent years indicates that the total water input (runoff, precipitation and groundwater) should be close to annual evaporation, which is about 1000mm equivalent to lake level. This is far less than the mean water depth, so energy advection by runoff and precipitation have very limited impact on energy budget of Paiku Co, so we do not consider these factors in this study.

80

*7. Errors in the water balance estimation. "lake evaporation (1.7 mm/day) is similar with the rate of lake-level decrease (1.8 mm/day). " What can be concluded from this? Do not forget about river discharge (0.62-0.89 mm/d) and precipitation (small but not zero in June according to Fig.7). " This difference (0.6 mm/day)....... is very close to the uncertainty of lake evaporation estimated by error propagation. " Do not forget about error of water balance estimation. I think explanations given in the authors' response (L436-447) should also be mentioned in the main text.*

85

Response: Runoff during the pre-monsoon season was measured in the late May or early June, so this represents the largest runoff during the pre-monsoon period. During the pre-monsoon season, lake evaporation (1.7 mm/day) is close to the rate of lake-level decrease (1.8 mm/day), which may indicate that the estimation of lake evaporation is accurate because runoff can be much less than the measured values earlier this time. The difference (0.6 mm/day) between lake level changes and evaporation during post-monsoon season is taken as uncertainty of lake evaporation for the entire ice-free period.

90

Minor comments to the manuscript

*L151-152 "There exists surface warming during the day and surface cooling at night for high elevation lakes. However, the daily difference between them is small during most time of a year because surface water can be mixed quickly by water convection or strong wind in the afternoon and the two uncertainties by surface warming and cooling can cancel each other at daily timescale." As pointed out before, these sentences are not appropriate without supporting evidence. In the previous review, I made the same comment. In L330 of the authors' response they write "We agree", and still the same sentences appear in the revised manuscript. I do not understand why.*

95

Response: We agree that there is some difference between lake skin temperature and surface water temperature. However, we believe that the difference between them is small on daily scale in this high elevation area because surface water can be

100

well mixed by water convection or strong wind in the afternoon. The two uncertainties by surface warming and cooling can cancel each other at daily timescale.

*Minor comments to the responses*

*L414-420. Not only in the response, but also in the main text, the product's name should be mentioned.*

Response: We have added the product's name in the revision (Line 348).

*L339-144. "I think it should be 72.8 because it is an integral of water depth." If the equation represents an integral, yes, it should be 72.8. But the equation is not an integral but a summation. Then the number of times you add should be an integer (and not a decimal).*

Response: We accepted this suggestion and revised the equation in the revision (Line 164).

*L366-369 "We checked...found the two references are in the reference list"; actually Zhang et al. (2015) was not in the list and is still not in the list. Stainsby et al. (2011) was indeed on the list but in the wrong place (it is still in the wrong place). Livingston (2003) was ok, but it is at a strange place now. I would recommend checking the entire list and correspondence with the main text citation.*

Response: The three references are indeed in the reference list. We are sorry for the wrong orders of them. We have checked the entire list in the revision.

 **Reviwer 2#**

*Most of the explanation to reviewers' comments were done in the contents, but they are still scattering and additionally. I would like to suggest brush up the story, especially for following matters.*

*Major comments*

140 *1) According to the abstract, one of highlights of this study is that evaporations from the lake were estimated with their differences depending on pre-monsoon, monsoon, and post-monsoon seasons. However, there is no clear explanation that why the defined months correspond to the actual seasonality of monsoon at Paiku Co. Monsoon variability can be identified by precipitation or wind condition at the lake, however, there is no analysis about the monsoon itself. Please add the explanation about in-situ monsoon seasonal changes to match the months. Also, "summer" is used in chapter 3.1 and*

145 *"Spring" and "Autumn" are used in Table 3. Please unify the term of season.*

Response: We added one sentence about the duration of Indian summer monsoon in the study area in the revision (Line Line290-291). 'As a monsoon dominated region, Indian summer monsoon around Paiku Co usually starts in the middle June and ends in the late September.'

'summer', 'spring', and 'autumn' have already been represented by 'monsoon season', 'pre-monsoon season' and 'post-

150 monsoon season' in the revision.

*2) My understanding is that heat budget at the lake surface was estimated by Bowen-ratio method where the ratio was determined by the temperature and humidity profile at the lake shore. But, downward radiation components were observed at another station 150 km far, and albedo was also used as constant value (0.07) based on a reference (I could not find it in*

155 *the reference, but it is not about the lake on TP). Upward L was estimated by the lake water, not the skin water temperature. Author need to explain those key concepts clearly at first in Chapter 2.3. Observing net radiation (Rn) is quite important for this method, and also important for results because comparison between the evaporation and Rn was discussed. I wonder the heat budget can be closed, if you use downward radiation components in such far station (Fig.S4 may help?) and also albedo do not change depending on the seasons or places. The paper discussed them in Chapter 3.6, but I think this is not a*

160 *matter of discussion but parts of method (to be discussed in Chapter 2) to proof that estimated values are accurate. Also, the estimation was done by daily or weekly scale. Also, again, please explain how the daily or weekly data are made from original data interval with treatment of missing data (see previous comment!). For instance, daily averaged Tw or Ta is used to obtain Bowen ratio in the formula (4)?*

Response: Radiation at Paiku Co is not measured so we have to use radiation data at Qomolangma station, which about 150

165 km east of Paiku Co. We compared the radiation data between the two places by using Hamawari-8 satellite data and found that daily solar radiation at the two sites exhibited similar seasonal variations with mean difference of 3.8 W $m^{-2}$. So it is reliable to use solar radiation at Qomolangma station to represent that at Paiku Co.

Albedo of lake surface was usually taken as constant value of 0.07 in most studies because lake surface is very flat and does not change much from lake to lake, which is different from land surface.

170 Upward L was estimated by the lake water, not the skin water temperature. We have addressed this in line 149-154. Because of the skin cooking or heating, the estimate should be slightly higher at night but slightly lower in the daytime, and thus the daily mean should be more accurate.

We believe that the 'uncertainty' part should be in discussion section because some of the data belongs to result part. If this part appear in the 'method' section, it may be difficult for authors to understand without reading the following parts.

175 About time scale of the main components in energy budget, we have addressed them clearly in each part of the data acquisition (Line 96-97, Line 106-107, Line 114-115, Line 161-162, Line170-172).

*3) I understood that deep lake could store the surface energy till the post monsoon. However, I still wonder that surface evaporation was significantly controlled by the heat storage to determine the lake-level, or not. I thought that lake*
180 *evaporation would be controlled by surface winds and dryness of atmospheric air mass. Namely, it is a matter of local climate. However, authors suspect it as lake depth matter, such as the importance of heat storage of deep lake (discussing it in Chapter 3.5 based on time lag story). As the Bowen ration is determined by the gradient (4), not by the absolute value of Tw, it is still uncertain that S control post-monsoon evaporation. Wind speed increase after monsoon (Fig.S2) that may be indicating exchanging wet monsoon air-mass to dry one which could mix the lake but also accelerate evaporation. Or, you*
185 *mean all lakes introduced in L306-313 exist under the same climate and surface-water budget condition? There is still a doubt to L404 "For deep lakes like Paiku Co, contrasting hydrological and thermal intensities determines the large amplitude of seasonal lake-level variations." It is better to consult with lake hydrologist.*

Response: We agree that lake evaporation is mainly controlled by surface winds and dryness of atmospheric air mass. In fact, this does not contradict with our conclusion. For deep lakes like Paiku Co, lake heat storage can affect lake evaporation
190 because lake water temperature is still high during the post monsoon. Although air temperature decreases considerably during the post-monsoon, there is still large water vapor difference between the lake surface and the overlying atmosphere. For shallow lakes, lake heat storage may only have very limited effect of lake evaporation because lake water temperature can decrease rapidly during post-monsoon season due to its small heat capacity.

For the observed lakes in Figure 8 (also L304-306), they are very close to each other. So they are located in the similar
195 climate condition and environment. However, there is considerably different amplitude of lake level seasonality.

Many studies have investigated the seasonal pattern of lake evaporation from deep lakes on the TP. For the first time we found that for deep lakes like Paiku Co, contrasting hydrological and thermal intensities determines the large amplitude of seasonal lake-level variations. We address this relationship in more detailed in the revision (Line 290-299).

200 *Individual comments*

*L26   Suddenly started about the "shallow lake". Why you start implications about deep and shallow matter here? Is this about alpine lake or in general? If there is no clear confidence that lake depth control the lake levels, it is better to delete this part in the abstract.*

Response: We have deleted this sentence in the revision and added another sentence about the implication of this study (Line26-28). 'This study further implies that lake evaporation may play an important role in the different amplitudes of seasonal lake-level variations between deep and shallow lakes and between the southern and northern TP'.

*Abstract; ±** is used. Is this a standard deviation?*

Response: ± means uncertainty of lake evaporation.

*L55-65 Better to cite examples of evaporation rate by previous studies here (++mm/day or season) to compare the your result in the abstract.*

Response: We have compared lake evaporation in this study with other studies in section 3.4 and Figure 6.

*L97 better to say "daily values were used to calculate*** or estimate +++".*

Response: Thanks for the suggestion. We have changed this sentence 'Water temperatures were recorded at an interval of 1 hour and daily values were used to investigate thermal structure of Paiku Co and estimate changes in lake heat storage in this study'.

*L114 "radiation at this station is used" for what?*

Response: We have changed this sentence to 'daily solar radiation and downward longwave radiation at this station were used to calculate the net radiation and energy budget over the lake surface.'(Line 114-115)

*L170-170 There is no information about missing data. Did you conduct perfect observation? Otherwise, better to explain how did you fill or interpolate missing values to get weekly/daily value.*

Response: Lake evaporation was not calculated during this period when there is no data available.

*L147 Tw is the same of "surface water temperature" in Fig. 5 ?*

Response: We have unified them to 'lake surface temperature' in the revision.

*L296 "These relationships illustrate how contrasting hydrological and thermal intensities played an important role in the large amplitude of seasonal lake-level variations at Paiku Co." is not clear.*

Response: We have rephrased this paragraph in the revision. (Line 290-299).

[revised manuscript text omitted]

---

## Author Response (AR4)

Editor comments:

*After consulting with three referees, I would like to recommend another wound of revision. Please consider carefully the comments provided by two of the three referees and submit a revision together with a point-by-point response to review comments.*

Response: Thanks a lot for dealing with our submission. We appreciate your and the two reviewers' constructive comments very much. We have revised the manuscript according to these comments and the two suggested papers. The main changes in the revision include 1): We addressed the fact that there is spatial difference of lake evaporation due to the horizonal difference of surface temperature and wind speed in section 3.6 (Line 337-344), however uncertainty of lake evaporation in this study is estimated according to measurements at a single point; 2): We gave more detailed description why the contrasting hydrological and thermal intensities determined seasonal lake level changes at deep lakes like Paiku Co (Line 294-303), 3): We addressed the characteristics of lake evaporation at Paiku Co in more detailed in the revision (Line 247-260). 4): We revised some small errors and modified some Figures (Fig. 4, 6).

[revised manuscript text omitted]

---

## Author Response (AR5)

**Reply to reviewer's comments**

*1. For the discussion on the relationship between lake evaporation and lake-level variation, the sub-surface water inflow and outflow present the highest uncertainties, but are lacking and considered to be not important in the manuscript. But*

*this may not be the case. From my own experience, the subsurface flows are very important for lake-level variations and should be considered with a great attention. Considering the importance of sub surface flows, the related contents can not appear as results in "abstract" and "conclusions". As most of the contents focus on the Paiku Co, the last sentence in abstract should be revised to exclude the information on "deep and shallow lakes and the southern and northern lakes"*

Reply: We appreciate the reviewer's constructive comments very much. At present, it is still difficult to accurately quantify the contribution of sub-surface inflow and outflow to lake level variations at Paiku Co. One solution is to investigate the lake water budget during the ice-covered season because both lake water input (snowfall, runoff) and output (sublimation) are very limited. If there is significant sub-surface inflow and outflow, the lake level should increase or decrease considerably. According to the in-situ lake level observations we have acquired between 2013 and 2020, we found that there is no significant lake level variations during the ice covered season. Taking 2013/2014 as an example, the lake level decreased slightly at a rate of 13 mm per month between mid-January and mid-April when the lake surface frozen up (Lei et al., 2018). During this period, surface runoff was still negligible because the melting of glacier and snow has not started yet. Lake sublimation is estimated to be ~40 mm per month on average according to the results at Nam Co (Wang et al., 2019). Based on lake water budget, we can conclude that sub-surface inflow to Paiku Co should be less than the rate of lake ice sublimation, which is much less than the seasonal lake level variations. Therefore, we do not consider the contribution of sub-surface inflow in this study. We have addressed this in line 137-145 in the revision.

About the abstract and conclusion, we agree with the reviewer's suggestion because most of the contents focus on Paiku Co. We have deleted the information on "deep and shallow lakes and the southern and northern lakes" in the abstract and conclusion.

Lei, Y., Yao, T., Yang, K., Bird B.W., Tian, L., Zhang, X., Wang W., Xiang Y., Dai, Y.F., Lazhu, Zhou, J., Wang, L.: An integrated investigation of lake storage and water level changes in the Paiku Co basin, central Himalayas, J. Hydrol., 562, 599–608.

Wang, B., Ma, Y., Wang, Y., Su, Z., Ma, W.: Significant differences exist in lake-atmosphere interactions and the evaporation rates of high-elevation small and large lakes, J. Hydrol., 573, 220–234, 2019.

*2. In section 3.2, ' the main components of energy budget over the lake surface, including solar radiation, ...f the lake body', generally all the three variables of ' solar radiation, atmospheric longwave radiation to the lake and upward longwave radiation from the lake body' belongs to the radiation budget, rather than energy budget in references. Thus, I*

*suggest to revise 'energy budget' to radiation budget' in this sentence.*

Reply: We agree with reviewer's suggestion and have revised this sentence in the revision.

[revised manuscript text omitted]